# OptiTree: Hierarchical Thoughts Generation with Tree Search for LLM Optimization Modeling

**Haoyang Liu[1],\* Jie Wang[1]†, Yuyang Cai[1], Xiongwei Han[2], Yufei Kuang[1], Jianye Hao[2,3]**

[1]MoE Key Laboratory of Brain-inspired Intelligent Perception and Cognition,
University of Science and Technology of China,
[2] Noah's Ark Lab, Huawei Technologies,  [3] Tianjin University

## Abstract

Optimization modeling is one of the most crucial but technical parts of operations research (OR). To automate the modeling process, existing works have leveraged large language models (LLMs), prompting them to break down tasks into steps for generating variables, constraints, and objectives. However, due to the highly complex mathematical structures inherent in OR problems, standard fixed-step decomposition often fails to achieve high performance. To address this challenge, we introduce OptiTree, a novel tree search approach designed to enhance modeling capabilities for complex problems through adaptive problem decomposition into simpler subproblems. Specifically, we develop a modeling tree that organizes a wide range of OR problems based on their hierarchical problem taxonomy and complexity, with each node representing a problem category and containing relevant high-level modeling thoughts. Given a problem to model, we recurrently search the tree to identify a series of simpler subproblems and synthesize the global modeling thoughts by adaptively integrating the hierarchical thoughts. Experiments show that OptiTree significantly improves the modeling accuracy compared to the state-of-the-art, achieving over 10% improvements on the challenging benchmarks. The code is released at `https://github.com/MIRALab-USTC/OptiTree/tree/main`.

## 1 Introduction

Optimization models are fundamental in operations research (OR) with a wide range of critical applications in route planning [33], production planning [4], economics [23], and so on. Typically, real-world OR problems are presented as natural language descriptions that must be manually converted into optimization models before developing solver codes (e.g., Gurobi [14] and Pyomo [6, 16]) to solve. However, the modeling process is highly technical and time-consuming, requiring extensive human expertise and domain knowledge [28]. Modeling experts often engage in thorough discussions with clients to fully grasp the problem scenarios and context. This is followed by a lengthy iterative process to refine and improve the models, enhancing their accuracy and efficiency.

To reduce the time and cost spent in optimization modeling, recent advances have leveraged large language models (LLMs) to automate this process, leveraging the rich domain knowledge learnt by LLMs [39, 1, 17]. Given an OR problem, the LLMs take a natural language description as input and generate both the optimization model and the corresponding solver code. Existing works on LLM-based optimization modeling include the prompt-based modeling methods [39, 1, 2] and the fine-tuned LLM modeling agents [17, 21]. The prompt-based methods typically break down modeling tasks into sequential steps for generating variables, constraints, and objectives [1, 2]. However, this

---

\*This work was done when Haoyang Liu interned at Huawei. E-mail: dgyoung@mail.ustc.edu.cn

†Corresponding author. Email: jiewangx@ustc.edu.cn

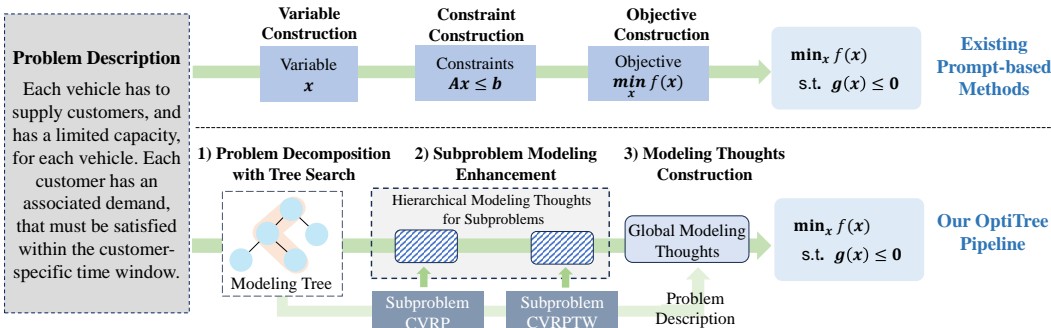

Figure 1: **The Upper**: Existing prompt-based methods construct variables, constraints and objectives sequentially. **The Lower**: Our OptiTree leverages tree search for subproblem decomposition and uses the modeling thoughts of subproblems to enhance the modeling process.

rigid decomposition does not account for problem complexity, making it particularly challenging for more complex problems and potentially leading to suboptimal modeling accuracy. Our analysis of a challenging dataset reveals that a substantial portion of errors arises from incorrect variable definitions, particularly in the hard modeling problems (please see Motivation 1 in Section 3).

Therefore, a natural question arises: how can we adaptively decompose a problem into simpler subproblems? For each OR problem, we introduce the concept of *modeling subproblem*, which is defined as parts of the original problem with reduced modeling complexity (please see Section 4.1 for a formal definition). For example, the standard vehicle routing problem (VRP) can serve as a subproblem for more complex variants such as the capacitated VRP (CVRP) and the VRP with time windows (VRPTW). From the perspective of OR, these subproblems share the same taxonomy as the original problems and exhibit similar modeling structures, but their lower complexity makes them easier to model. When modeling a VRP with time windows, we might begin by constructing a standard VRP model and then incrementally integrate the time window constraints. We have observed that these patterns of subproblem decomposition are prevalent across a wide range of complex problems. While most complex problems do not fit neatly into standard OR categories, they often encompass simpler standard OR subproblems (please see Motivation 2 in Section 3). For further understanding, we provide several examples in Appendix C.

In light of this, we propose a novel approach called OptiTree, designed to enhance modeling accuracy by adaptively decomposing complex problems into a series of simpler subproblems. The core of OptiTree is to *distill prevalent decomposition patterns from various problems and apply the most suitable ones for unseen complex problems*. Specifically, as illustrated in Figure 1, we develop a modeling tree that organizes a diverse range of OR problems according to their taxonomy and complexity. Each node in the modeling tree represents an OR problem, with problems of parent nodes corresponding to the subproblem of their children. We store relevant modeling thoughts in each tree node to enhance the modeling accuracy for each subproblem. Given a problem to solve, we 1) recurrently search for suitable subproblems to decompose, 2) retrieve high-level modeling thoughts for each subproblem, and 3) synthesize global modeling thoughts by adaptively integrating the hierarchical thoughts. To ensure the scalability and reliability of the modeling tree, we dynamically update and refine the subproblems and thoughts across a wide range of OR problems. The notable feature of OptiTree is that it transfers the search into the highly structured subproblem space, significantly reducing the search space while fully leveraging the modeling thoughts associated with the subproblems. Experiments demonstrate that our method achieves state-of-the-art performance, significantly improving modeling accuracy by over 10%.

## 2 Related Work

**LLM-based Optimization Modeling** LLM-based optimization modeling has emerged as a promising approach to reduce the time and expertise required during the modeling process [7, 22]. Existing methods in this field can be categorized into prompt-based and fine-tuned methods. Prompt-based methods involve the careful design of modeling prompts for pre-trained LLMs, such as GPT-4o and DeepSeek-V3. Notable works in this area include CoE [39] and OptiMUS [1], which utilize

multi-agent cooperation workflows to iteratively construct models and solver codes. Additionally, [2] employs Monte Carlo tree search to explore the space of variables, constraints, and objectives step by step, identifying the best components for model construction. In contrast, another line of research focuses on fine-tuning LLMs with extensive operations research and modeling knowledge. Examples include ORLM [17], Evo-Step [37], LLMOPT [21], OptMATH [25], and OptiBench [41, 35]. These methods typically generate large modeling datasets to train specialized modeling language models. In this paper, we focus on the prompt-based methods and aim to fully exploit the reasoning capabilities of pre-trained LLMs. Unlike existing approaches that decompose tasks into fixed steps for generating variables, constraints, and objectives, we propose an incremental modeling strategy that involves modeling simpler subproblems.

**LLM Reasoning and Retrieval-Augmented Generation** LLMs have shown promising performance across various reasoning tasks [43]. However, they are prone to generating false, misleading, or fabricated information—a phenomenon known as hallucination. Consequently, researchers have explored several methods to mitigate these hallucinations [18]. Early approaches, such as Chain-of-Thoughts (CoT) [36] and Tree-of-Thoughts (ToT) [42], break down complex problems and solve them step-by-step. The Buffer-of-Thoughts (BoT) [40] method stores a meta-buffer of modeling thoughts, instantiating relevant templates to construct reasoning processes. Recent advances in reasoning LLMs such as OpenAI-o1 [30] and DeepSeek-R1 [8] have gained significant popularity. Additionally, researchers have focused on leveraging external knowledge to further reduce hallucinations [44, 10, 13]. Retrieval-augmented generation (RAG) involves querying vector or text databases for relevant documents and integrating this retrieved information into the generation process, thereby producing more accurate responses [12, 3]. In our work, we search for relevant modeling thoughts for each subproblem and dynamically combine them to generate global modeling thoughts.

# 3 Motivated Observations

In this section, we present key observations that motivate our decomposition method. (1) The standard fixed-step decomposition approach often fails in complex problems, as accurately identifying variables remains a significant challenge. (2) Most complex OR problems contain subproblems of standard OR problems. (3) The performance of LLMs can benefit from the subproblem decomposition. The experiments in this part are conducted in the modeling dataset IndustryOR [17], which classifies problems into three difficulty levels: Easy, Medium, and Hard.

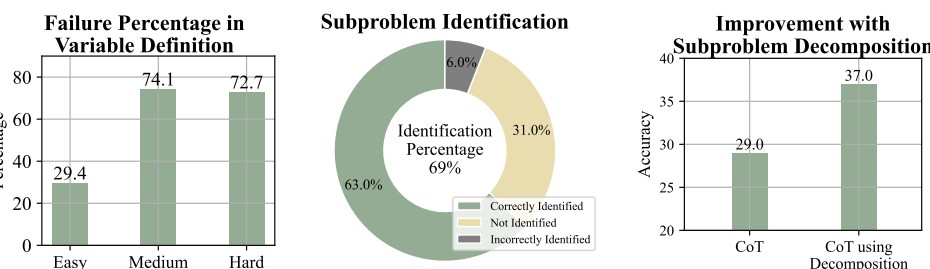

Figure 2: The observations. **Left**: We find most errors come from the incorrect variable definition in the Medium and Hard problems. **Middle**: We find the most complex problems have subproblems of standard OR problems, and LLMs are effective in identifying the subproblems. **Right**: We observe an improvement in modeling accuracy using subproblem decomposition.

**Motivation 1: The Drawbacks of Existing Methods** We evaluate CoT [36] on this dataset, breaking the problems down into steps involving variables, constraints, and objective generation. We find that a significant portion of errors arise from incorrect variables, particularly in Medium and Hard problems (over 70%, please see the left subfigure of Figure 2). This indicates that existing decomposition methods are not well-suited for complex problems.

**Motivation 2: Prevalent Subproblem Decomposition Patterns** We first collect the optimization models in the textbook [1] as a ground-truth model for the 50 standard problems. For a problem in IndustryOR to formulate, we use LLM to identify whether the problem contains subproblems

in the 50 standard OR problems. If a subproblem is identified, we augment the LLM's prompt for optimization modeling by integrating the relevant ground-truth model of the subproblem as hints. As shown in the middle subfigure of Figure 2, the LLM identifies subproblems for 69% of the problems. Upon manual verification, we find that over 63% of problems are associated with correct subproblems. These results underscore the prevalence of subproblem decomposition and demonstrate that LLMs are effective in identifying relevant subproblems with high identification accuracy.

**Motivation 3: LLMs Perform Better with Vanilla Subproblem Decomposition**    We leverage the set of 50 standard OR problems as subproblem candidates to guide the modeling process. For each problem in IndustryOR, we decompose the process into two steps. First, we use LLM to select a suitable subproblem from the candidates. If a relevant subproblem is identified, we provide the ground-truth models for that subproblem before proceeding to model the original problem based on it. We evaluated the performance of LLMs using this two-step approach and present the results in the right subfigure of Figure 2. Our findings indicate that modeling based on a subproblem reduces task complexity and has the potential to enhance performance.

# 4    OptiTree: Hierarchical Thought Generation with Tree Search

The overall framework of OptiTree is illustrated in Figure 3. The core is to organize modeling knowledge through a hierarchical tree structure that captures both prevalent decomposition patterns and thoughts for various problems (Section 4.2). We utilize tree search for efficient retrieval of subproblems along with their hierarchical modeling thoughts, adaptively integrating these thoughts to construct comprehensive global modeling thoughts (Section 4.3). To ensure scalability and reliability, we construct the modeling tree using a real-world modeling dataset, dynamically managing both subproblems and modeling thoughts. This enhances the tree's capacity to distill more decomposition patterns and thoughts, allowing it to generalize effectively to unseen complex problems (Section 4.4).

## 4.1    Subproblem Definition and Identification

Given an OR problem $\mathcal{P}$, we specialize the corresponding optimization model as follows,

$$\min_{\boldsymbol{x}} f(\boldsymbol{x}) \quad \text{s.t. } g_i(\boldsymbol{x}; \beta_i) \leq 0, \text{ for } i = 1, \cdots, N, \tag{1}$$

where $f$ denotes the objective function, $\boldsymbol{x}$ is the decision variables, $g_i$ represents the the $i^{th}$ constraint function, and $\beta_i$ is the parameters in the $i^{th}$ constraint. Suppose that we partition the variables into two groups $\boldsymbol{x} = (\tilde{\boldsymbol{x}}_1, \tilde{\boldsymbol{x}}_2)$, the objective and constraints has the decomposition $f(\boldsymbol{x}) = f_1(\tilde{\boldsymbol{x}}_1) + f_2(\tilde{\boldsymbol{x}}_2) + f_3(\tilde{\boldsymbol{x}}_1, \tilde{\boldsymbol{x}}_2)$, and $g_i(\boldsymbol{x}_i, \beta_i) = g_{i,1}(\tilde{\boldsymbol{x}}_1; \tilde{\beta}_{i,1}) + g_{i,2}(\tilde{\boldsymbol{x}}_2; \tilde{\beta}_{i,2}) + g_{i,3}(\tilde{\boldsymbol{x}}_1, \tilde{\boldsymbol{x}}_2; \tilde{\beta}_{i,3})$. We define another OR problem $\tilde{\mathcal{P}}$ as a subproblem of $\mathcal{P}$, if the optimization model takes the following form,

$$\min_{\tilde{\boldsymbol{x}}_1} f_1(\tilde{\boldsymbol{x}}_1) \quad \text{s.t. } g_{i_k,1}(\tilde{\boldsymbol{x}}_1; \beta_{i_k,1}) \leq 0, \text{ for some } i_k \in \{1, \cdots, N\}, \tag{2}$$

where $\beta_{i_k,j}$ is the parameter, and the optimization model (2) can be viewed as part of the original optimization model (1). However, we need to identify a subproblem through the natural language descriptions, as the ground-truth optimization models are unavailable in practice. Directly comparing two problem descriptions can lead to hallucination issues for LLMs, so we adopt a clear and comprehensible format. We first use a LLM to distill and summarize the problem description into a set of atomic high-level statements $\mathcal{C}_{\mathcal{P}} = \{c_1, c_2, \cdots, c_{n_{\mathcal{P}}}\}$, called statement thoughts, where each thought $c_i$ summarize a feature or requirement related to optimization modeling. We then use the LLM to identify a subproblem $\tilde{\mathcal{P}}$ of $\mathcal{P}$, if statement thoughts of $\tilde{\mathcal{P}}$ are semantically contained within those of $\mathcal{P}$ (denoted by $\mathcal{C}_{\tilde{\mathcal{P}}} \subseteq_{\mathcal{S}} \mathcal{C}_{\mathcal{P}}$). With slight abuse of notation, we write $\tilde{\mathcal{P}} \subseteq_{\mathcal{S}} \mathcal{P}$ to indicate that $\tilde{\mathcal{P}}$ is a subproblem of $\mathcal{P}$. Due to the limited space, please see Appendix J.1 and J.2 for examples of statement thoughts and detailed meta-prompts for subproblem identification.

Now we formulate the modeling process using subproblem decomposition as follows. We decompose the OR problem as a series of subproblems $[\mathcal{P}^{(1)}, \mathcal{P}^{(2)}, \cdots, \mathcal{P}^{(M)}]$, where $M$ is the subproblem number and $\mathcal{P}^{(1)} \subseteq_{\mathcal{S}} \mathcal{P}^{(2)} \subseteq_{\mathcal{S}} \cdots \subseteq_{\mathcal{S}} \mathcal{P}^{(M)}$. We then build the optimization model incrementally.

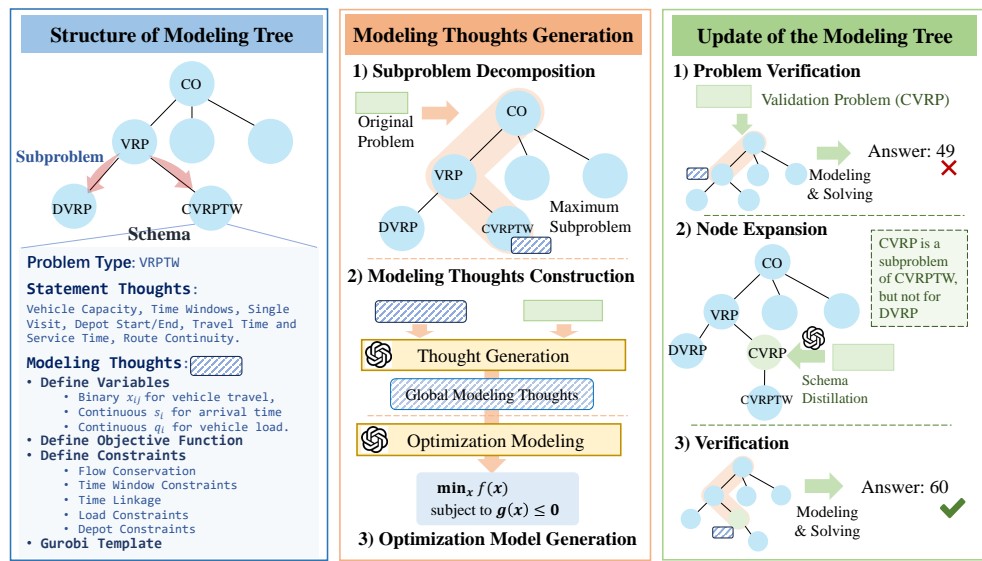

Figure 3: The overview of OptiTree. (1) OptiTree constructs a modeling tree for subproblem decomposition and stores the modeling thoughts for subproblems in the tree nodes. (2) OptiTree leverages tree search to identify the subproblems and construct modeling thoughts by integrating the subproblems' thoughts. (3) We update the modeling tree to ensure its scalability and reliability.

## 4.2 Modeling Tree for Problem Decomposition

We propose the concept of modeling tree for problem decomposition. We also utilize the modeling thoughts, distilled from the real-world modeling process for each subproblem, to effectively guide LLMs in accurately modeling these subproblems.

**Modeling Thoughts for OR Problems** Most complex OR problems involve conventional modeling techniques, such as variable definition and constraint formulation, which can be implicit and pose significant challenges for reasoning. Thus, we distill these modeling techniques into high-level modeling thoughts, which serve as concise, step-by-step guidelines for the modeling process. When faced with a similar optimization problem, we can utilize these informative thoughts to enhance accuracy. Specifically, we design a comprehensible schema to efficiently retrieve and apply the modeling thoughts (Please see the left of Figure 3 for the schema). Each schema contains three critical elements: (1) the name of problem category, (2) statement thoughts $\mathcal{C}_\mathcal{P}$ for problem identification, and (3) modeling thoughts $\mathcal{T}(\mathcal{P})$ to guide the modeling process. These modeling thoughts include concrete reasoning steps for variable definition, constraint formulation, objective formulation, and Gurobi code templates. Unlike static examples, these thoughts are flexible and can generalize to various complex problems with similar mathematical structures.

**Modeling Tree of OR Problems** While existing prompt-based and fine-tuned methods consider each OR problem individually, our method fully recognizes the relationships between different OR problems and organizes them into a taxonomy tree known as the modeling tree. In this structure, each node represents an OR problem along with its corresponding schema of modeling thoughts. The root node represents an abstract class of combinatorial optimization problems. Each parent node is a subproblem of its child nodes; that is, if the node $\mathcal{P}_i$ is a child of $\mathcal{P}_j$, then $\mathcal{P}_j \subseteq_\mathcal{S} \mathcal{P}_i$. Child nodes inherit fundamental constraints and variables from their parent nodes while adding specialized components. For example, the VRP branch might split into dynamic VRP (DVRP) and VRP with time windows. The top tier of the tree consists of simpler, foundational OR problems that are easier to model, while deeper levels feature increasingly complex variations. This structure organizes similar problems into the same branch and facilitates an efficient search for the subproblems. Please see the left of Figure 3 for the structure of the modeling tree. We formally define this structure as follows.

**Definition 4.1.** The modeling tree is called *subproblem order-preserving* if for any problem $\mathcal{P}_i$ that is the ancestor of problem $\mathcal{P}_j$, it holds that $\mathcal{P}_i \subseteq_\mathcal{S} \mathcal{P}_j$.

### 4.3 Modeling Thoughts Searching and Construction

**Tree Search for the Subproblems and Modeling Thoughts**   Given an OR problem $\mathcal{P}$ to model, we search the modeling tree for suitable subproblems for decomposition. We employ an LLM to extract the statement thoughts $\mathcal{C}_\mathcal{P}$ and compare $\mathcal{C}_\mathcal{P}$ with the statement thoughts at the tree node to identify relevant subproblems. Starting from the root node, we search for the first subproblem in the first tree level, i.e., $\mathcal{P}_1^{(0)}, \mathcal{P}_2^{(0)}, \cdots, \mathcal{P}_T^{(0)}$. We select the subproblem that best matches $\mathcal{P}$ based on the highest similarity,

$$\mathcal{P}^{(1)} = \underset{\mathcal{P}_t^{(0)}(t=1,\cdots,T)}{\arg\max} \ \mathrm{I}(\mathcal{P}_t^{(0)} \subseteq_\mathcal{S} \mathcal{P}) \cdot \mathrm{Sim}_{\mathrm{LLM}}(\mathcal{C}_{\mathcal{P}_t^{(0)}}, \mathcal{C}_\mathcal{P}), \tag{3}$$

where I is the indication function with $\mathrm{I}(a) = 1$ if the condition $a$ is true and $\mathrm{I}(a) = -\infty$ otherwise. $\mathrm{Sim}_{\mathrm{LLM}}$ represents the similarity score provided by the LLM, measuring the similarity between two problems. We select the child problem $\mathcal{P}^{(1)}$ and continue the search process on the corresponding tree node. If none of the children qualify as subproblems of $\mathcal{P}$ (with similarity score 0), we terminate the search. If the search halts at the root node, we do not provide modeling thoughts. Please see Appendix J.2 for meta-prompts for the tree search.

**Global Modeling Thought Construction**   We obtain a series of identified subproblems $\mathcal{P}^{(1)}, \mathcal{P}^{(2)}, \cdots, \mathcal{P}^{(M)}$ of increasing complexity. Notably, the modeling thoughts of the problem $\mathcal{P}^{(M)}$ encompass those of the preceding subproblems. Thus, we call $\mathcal{P}^{(M)}$ the *maximum subproblem*. We retrieve the corresponding modeling thoughts of $\mathcal{P}^{(M)}$, i.e., $\mathcal{T}(\mathcal{P}^{(M)})$. Subsequently, we combine these modeling thoughts with the problem description of $\mathcal{P}$ to synthesize the global modeling thoughts $\mathcal{T}(\mathcal{P})$, thereby enhancing the modeling process (for meta-prompts, please see Appendix J.3).

### 4.4 Construction of the Modeling Tree

**Tree Construction**   The tree construction and updating process is fully automated and does not require human curation. To ensure the problem coverage and reliability, we construct the modeling tree from scratch using a dataset that includes problem descriptions and their corresponding ground-truth optimization models. This allows us to distill decomposition patterns and modeling thoughts effectively. In practice, we choose parts of the OR-Instruct 3K dataset, which is the training dataset of ORLM [17] covering a wide range of OR problems. We sequentially add tree nodes to enhance the capacity of the modeling tree. For each problem in the dataset, we evaluate whether it should be integrated into the tree. First, we conduct a tree search for subproblems and identify the maximum subproblem $\mathcal{P}^{(M)}$, which has the greatest depth. We then construct global modeling thoughts using $\mathcal{T}(\mathcal{P}^{(M)})$ to guide the modeling process and solve the model for the final answer. If this final answer matches the ground-truth answer, no update to the tree is necessary, indicating that the modeling tree can successfully handle this type of problem. Conversely, if the answers differ, it suggests the emergence of new decomposition patterns or modeling thoughts, necessitating an update to the tree using the failed problem $\mathcal{P}$.

**Node Expansion**   We aim to expand the tree node with the problem $\mathcal{P}$ by following carefully designed rules to maintain the subproblem order-preserving structure. Due to the search process, $\mathcal{P}$ should be a child node of $\mathcal{P}^{(M)}$, as $\mathcal{P}^{(M)}$ is the maximum subproblem. We first distill the modeling schema (problem type, $\mathcal{C}_\mathcal{P}, \mathcal{T}(\mathcal{P})$) for the failed problem. Next, we verify the subproblem relationships between $\mathcal{P}$ and the children of $\mathcal{P}^{(M)}$, denoted as, $\mathcal{P}_1^{(M)}, \mathcal{P}_2^{(M)}, \cdots, \mathcal{P}_{T_M}^{(M)}$.

- If $\mathcal{P} \subseteq_\mathcal{S} \mathcal{P}_k^{(M)}$, we insert $\mathcal{P}$ as a child of $\mathcal{P}^{(M)}$ and as a parent of $\mathcal{P}_k^{(M)}$.

- Otherwise, we insert $\mathcal{P}$ as a child of $\mathcal{P}^{(M)}$ and as a sibling of $\mathcal{P}_k^{(M)}$.

**Proposition 4.2.** *The modeling tree remains subproblem order-preserving during the update process.*

Please refer to Appendix F for the proof. After adding a new node, we conduct the decomposition and modeling process again to verify the correctness of the new nodes, continuing this until the problem $\mathcal{P}$ can be correctly modeled.

Table 1: Comparison of modeling accuracy between our method and baselines across the benchmarks. We mark the best results in **bold** the underline the second-best results.

| Model | Method | NL4Opt | MAMO EasyLP | MAMO ComplexLP | ComplexOR | IndustryOR | OptiBench | OptMATH |
|---|---|---|---|---|---|---|---|---|
| | | | | Fine-tuned Method | | | | |
| | ORLM | 85.7 | 82.3 | 37.4 | 63.2 | 38.0 | 51.1 | 2.6 |
| | Evo-Step | 84.5 | 85.3 | 61.6 | - | 36.4 | - | - |
| | OptMATH | 95.9 | 89.9 | 54.1 | - | 31.0 | 66.1 | 34.7 |
| | LLMOPT | 93.0 | **97.0** | 68.0 | 72.7 | 46.0 | 66.4 | 40.0 |
| | | | | Prompt-based Methods | | | | |
| Reasoning LLMs | DeepSeek-R1 | 86.1 | 79.5 | 57.3 | 68.4 | 38.0 | 70.2 | 33.1 |
| | OpenAI-o1 | 87.1 | 87.6 | 54.5 | 73.6 | 40.0 | 71.5 | 34.9 |
| GPT-4o | Standard | 70.3 | 84.3 | 41.2 | 57.8 | 27.0 | 42.3 | 17.5 |
| | CoT | 71.6 | 84.8 | 42.3 | 57.8 | 29.0 | 42.0 | 20.5 |
| | CoE | 76.4 | 85.7 | 46.4 | 68.4 | 34.0 | 43.2 | 18.6 |
| | OptiMUS | 82.0 | 85.1 | 47.3 | 79.0 | 34.0 | 45.8 | 20.2 |
| | MCTS | 90.3 | 87.4 | 56.8 | 68.4 | 42.0 | 64.0 | 37.3 |
| | OptiTree | 96.2 | 95.6 | 81.0 | **84.2** | 48.0 | 71.9 | 45.8 |
| DeepSeek-V3 | Standard | 70.5 | 84.3 | 39.8 | 52.6 | 29.0 | 52.4 | 16.2 |
| | CoT | 74.0 | 82.9 | 40.7 | 52.6 | 35.0 | 53.1 | 21.1 |
| | CoE | 79.2 | 85.9 | 43.1 | 63.2 | 33.0 | 55.2 | 24.1 |
| | OptiMUS | 80.6 | 87.1 | 45.2 | 79.0 | 36.0 | 58.8 | 32.5 |
| | MCTS | 89.6 | 88.0 | 51.6 | 79.0 | 46.0 | 67.9 | 38.6 |
| | OptiTree | **98.3** | 96.9 | **81.5** | **84.2** | **54.0** | **74.7** | **52.4** |

# 5 Experiments

## 5.1 Experiment Setups

**Dataset** We consider seven modeling datasets to evaluate our method and the baselines. (1) NL4Opt [32] is from NL4Opt competition in NeurIPS 2022, composing 289 elementary-level linear programming problems. (2) MAMO EasyLP [20] contains 652 easy linear programming problems. (3) MAMO ComplexLP [20] consists of 211 more complex optimization problems. (4) ComplexOR [39] has 19 challenging problems derived from academic papers, textbooks, and real-world industry scenarios. (5) IndustryOR [17] contains 100 real-world problems from eight industries with different difficulty levels: Easy, Medium and Hard. Finally, (6) OptiBench [41] contains 605 problems, and (7) OptMATH [25] dataset has 166 challenging problems. The work OptMATH [25] proposed a data generation process for optimization modeling; OptMATH can be referred to both the testing dataset and the model trained on the training set.

**Baselines** OptiTree is a prompt-based model for LLM-based optimization modeling. We compare our method mainly with seven prompt-based baselines. The five prompt-based methods are as follows. (1) Standard is the direct output of pre-trained LLMs. (2) Chain-of-Thoughts (CoT) breaks the problem into reasoning steps. (3) Chain-of-Expert (CoE) [39] is a multi-agent workflow for modeling, with agents focusing on interpreting the problems, formulating the problems, writing and debugging the solver codes. (4) OptiMUS [1] is an improved multi-agent workflow with structured problem input. (5) MCTS [2] employs Monte Carlo tree search to search for variables, constraints and objective sequentially in different search depths. We also compare our method with (6) DeepSeek-R1 [8] and (7) OpenAI-o1 [30]. For completeness, we also report the results of fine-tuned OR LLMs. The fine-tuned models, (8) ORLM [17] (using LLaMA-3-8B as backbone), (9) Evo-Step [37] (using LLaMA-3-8B as backbone), (10) OptMATH [25] (using Qwen2.5-32B as backbone) and (11) LLMOPT [21] (using Qwen1.5-14B as backbone). Notice that the trained model of OptMATH and Evo-Step have not been released, the results of these two baselines follows those in the original papers [37] and [25].

**Metric and Implementation** Following existing works [39, 1, 17, 2, 21], we use solving accuracy to evaluate the performance, i.e., whether the optimal objective of the generated optimization models equals to ground-truth values. To demonstrate the generalization of our method, we conduct experiments on GPT-4o [29] and DeepSeek-V3 [9]. We construct the modeling tree using 400 randomly selected problems in the OR-Instruct dataset [17], which is part of the training dataset for ORLM and contains 3,000 problems. We run across problems in the dataset to update the modeling tree, where the statistical information of the tree can be found in Appendix B.

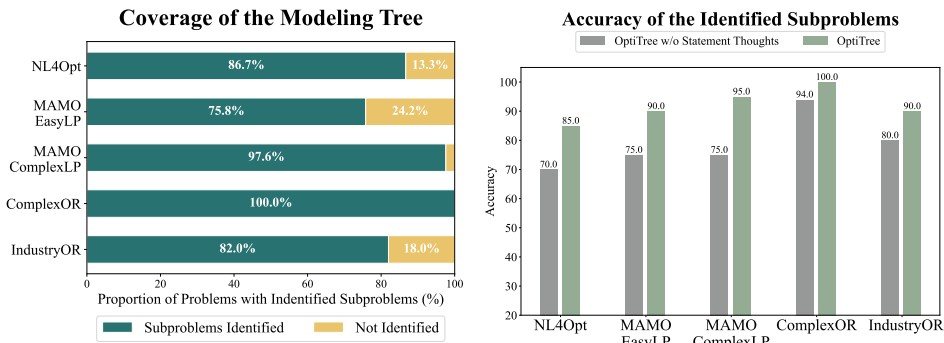

Figure 4: **Left**: OptiTree can identify subproblems for a wide range of problems, including the challenging datasets. **Right**: We manually check the accuracy of the subproblem decomposition.

## 5.2 Main Results

We evaluate OptiTree against the competitive baselines on five modeling datasets using the DeepSeek-V3 and GPT-4o models. The results, presented in Table 1, highlight three key findings. (1) **High Accuracy**. Our method significantly outperforms the baselines, achieving approximately a 10% improvement on the challenging datasets MAMO ComplexLP, ComplexOR, and IndustryOR. (2) **Strong Generalization**. Using only 400 problems, the constructed OptiTree has shown strong generalization ability across both easy and challenging benchmarks. (3) **Adaptation to Different LLMs**. Built on top of different LLMs, OptiTree consistently adapts well and exhibits impressive performance. Furthermore, it outperforms state-of-the-art reasoning LLMs, including DeepSeek-R1 and OpenAI-o1. OptiTree also outperforms the fine-tuned modeling LLMs. In Appendix A.1, we also report the proportion of codes generated by the methods that can be executed successfully.

## 5.3 Analysis on the Subproblem Decomposition

**Coverage: Can we find a subproblem in the search process?** We present the success rate of identifying subproblems, excluding the root node, in the left subfigure of Figure 4. The results demonstrate that OptiTree can successfully identify subproblems for an average of 88% of the problems. This observation suggests that, despite the diversity and complexity of OR problems, there are prevalent decomposition patterns and modeling techniques that generalize effectively across various problems. The modeling tree successfully captures these patterns.

**Reliability: Does the obtained subproblem correctly match the original problem?** We randomly sample twenty problems from each dataset (nineteen from ComplexOR) where LLMs can identify at least one subproblem. We manually verify whether the selected subproblems are suitable and relevant. We compare OptiTree with its variant, OptiTree w/o Statement Thoughts, which relies solely on problem descriptions for subproblem identification. The results in the right subfigure of Figure 4 demonstrate that OptiTree achieves a high accuracy in subproblem identification, while OptiTree w/o Statement Thoughts suffers from a severe drop in accuracy.

**Efficiency: What about the time cost of the searching process?** We analyze the average time cost for each problem, comparing OptiTree with other prompt-based baselines in Table 2. We (1) use tree search to find an appropriate problem and (2) perform optimization modeling, combining the modeling thoughts from the problem. The two parts of time (Tree search and modeling) refer to these two steps. We also provide the average time to solve all the problems across the benchmarks. For the baselines, the time is just the average inference time, and for OptiTree, this is the average of the one-time cost of the tree construction and inference time. **Specifically, the one-time cost across the benchmarks is under three hours**. The execution time for the solver code is under a second, which can be considered negligible. OptiTree exhibits the highest efficiency compared to baselines. While the search and modeling time increases for more complex problems (such as MAMO ComplexLP, ComplexOR, and IndustryOR), we find that the majority of the time is spent on modeling, indicating that the search process itself is efficient.

To understand why OptiTree is more efficient, we analyze on the following two factors. First, OptiTree explores a much smaller search space. OptiTree searches predefined candidate subproblems, requiring only the identification of the most suitable subproblem within a finite set, instead of the

Table 2: Comparison of the total time (seconds) of prompt-based methods.

| Method | NL4Opt | MAMO EasyLP | MAMO ComplexLP | ComplexOR | IndustryOR |
|---|---|---|---|---|---|
| CoE | 37.2 | 42.6 | 76.7 | 80.0 | 81.8 |
| OptiMUS | 26.4 | 22.5 | 53.2 | 68.3 | 57.8 |
| MCTS | 103.2 | 110.8 | 111.4 | 190.1 | 124.6 |
| OptiTree (Tree Search) | 8.9 | 4.9 | 4.2 | 11.2 | 8.4 |
| OptiTree (Modeling) | 5.0 | 4.4 | 9.1 | 19.8 | 11.5 |
| OptiTree (Inference) | 13.9 | 9.3 | 13.3 | 31.0 | 19.9 |

Table 3: Comparison of different search depths.

| Model | Method | NL4Opt | MAMO EasyLP | MAMO ComplexLP | ComplexOR | IndustryOR |
|---|---|---|---|---|---|---|
| GPT-4o | OptiTree (Depth=1) | 93.8 | 93.3 | 75.4 | 68.4 | 40.0 |
| | OptiTree (Depth=3) | 96.2 | 95.6 | 80.1 | 79.0 | 44.0 |
| | OptiTree | 96.2 | 95.6 | 81.0 | 84.2 | 48.0 |
| DeepSeek-V3 | OptiTree (Depth=1) | 97.2 | 95.3 | 75.4 | 68.4 | 48.0 |
| | OptiTree (Depth=3) | 97.9 | 95.7 | 79.6 | 79.0 | 50.0 |
| | OptiTree | 98.3 | 96.9 | 81.5 | 84.2 | 54.0 |

variable space or constraint space that can grow exponentially large for more complex problems. Second, OptiTree has fewer iterations in the workflow. While the multi-agent-based baselines often use a manager to automatically decide the agent calls, OptiTree avoids the useless agent calls with a more streamlined workflow.

## 5.4 Ablation Studies

**Impact of the Tree Search** We first investigate the impact of the tree search on modeling performance. The modeling tree organizes subproblems within its nodes. We compare this approach with a method that does not utilize a tree structure or tree search for problem decomposition, referred to as OptiTree w/o Tree Search. In this variant, LLMs directly select suitable subproblems from a pool at each step, rather than utilizing a structured modeling tree. The subproblem pool is also collected from the same dataset as the modeling tree. As shown in the left subfigure of Figure 5, OptiTree w/o Tree Search exhibits a decline in modeling accuracy. This degradation occurs because the modeling tree effectively reduces the search space and mitigates hallucinations for LLMs at each step.

**Impact of the Modeling Thoughts** We conduct ablation experiments on the impact of the modeling thoughts. We implement a variant of OptiTree, called OptiTree w/o Modeling Thoughts, which models the corresponding subproblems step by step without utilizing modeling thoughts. The results are presented in the right subfigure of Figure 5. We observe a significant performance drop in this method, particularly on the challenging datasets. This highlights the crucial role of modeling thoughts in guiding the modeling of subproblems.

**Impact of the Search Depth** To better understand the tree search process, we also conduct experiments on different depth limitations of the tree search process. The depth represents the number of decomposed subproblems in the modeling process, with greater depth allowing for more decomposition subproblems. OptiTree operates without depth restrictions, while we implement two variants—OptiTree (depth=1) and OptiTree (depth=3)-which limit the tree search depths to 1 or 3, respectively. The results in Table 3, demonstrate that a deeper search depth enables the identification of more suitable subproblems for decomposition, resulting in improved modeling performance. OptiTree (depth=3) still achieves promising results, underscoring the effectiveness of the tree search.

**Impact of the Statement Thoughts** The statement thoughts distilled from OR problems provide a clear and comprehensible format for subproblem identification. To investigate the effects, we implement a variant that disables statement thoughts for subproblem identification, instead relying on the problem descriptions, referred to as OptiTree w/o Statement Thoughts. As shown in Table 4, we observe a significant performance decline without statement thoughts, accompanied by an accuracy drop in subproblem matching in the right subfigure of Figure 4. This decline occurs because the statement thoughts summarize the modeling-related information to reduce hallucinations for LLMs.

## 6 Conclusion

This paper introduces a novel LLM-based optimization modeling method called OptiTree, which decomposes complex problems into a series of simpler subproblems to enhance modeling accuracy.

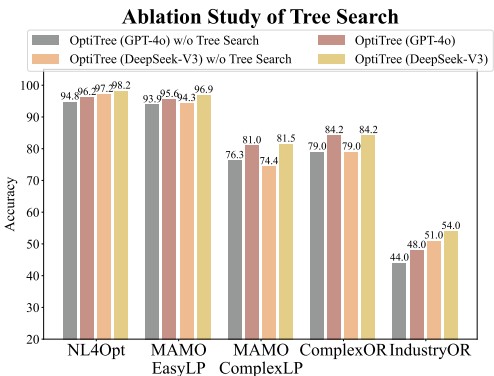
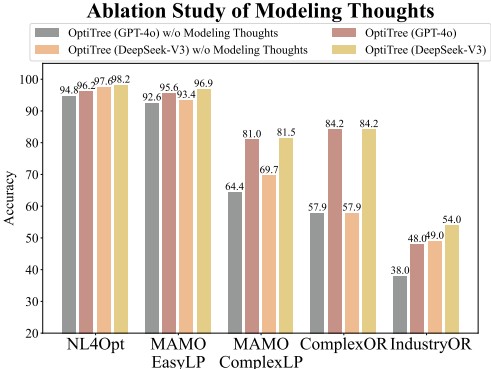

Figure 5: Ablation studies on the tree search (left) and modeling thoughts (right). The results demonstrate the critical roles of these two components.

Table 4: Ablation studies on the statement thoughts.

| Model | Method | NL4Opt | MAMO EasyLP | MAMO ComplexLP | ComplexOR | IndustryOR |
|---|---|---|---|---|---|---|
| GPT-4o | OptiTree w/o Statament Thoughts | 91.0 | 91.5 | 69.2 | 72.7 | 42.0 |
| | OptiTree | 96.2 | 95.6 | 81.0 | 84.2 | 48.0 |
| DeepSeek-V3 | OptiTree w/o Statament Thoughts | 87.1 | 92.6 | 75.3 | 68.4 | 46.0 |
| | OptiTree | 98.3 | 96.9 | 81.5 | 84.2 | 54.0 |

OptiTree dynamically updates a modeling tree to store decomposition patterns and modeling thoughts for these subproblems, employing a tree search to identify suitable subproblems at each step. Experimental results demonstrate the superiority of OptiTree, consistently outperforming baseline methods across several challenging modeling datasets.

## Acknowledgments

The authors would like to thank all the anonymous reviewers for their insightful comments and valuable suggestions. This work was supported by the National Key R&D Program of China under contract 2022ZD0119801, and the National Nature Science Foundations of China grants U23A20388, 62021001 and 624B1011.

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

## Table of Contents for Appendix

# A    Additional Experiments

## A.1    The Code Pass Rate for Each Dataset

We present the code pass rates of various methods across the benchmarks. Following the definition in [39], the code pass rate refers to the proportion of solver code that can be executed without encountering any errors. The results in Table 5 demonstrate that our method achieves the highest code pass rate across all datasets, demonstrating the high quality of the codes generated by our approach.

Table 5: Comparison of code pass rates between our method and baselines across the benchmarks.

| Model | Method | NL4Opt | MAMO EasyLP | MAMO ComplexLP | ComplexOR | IndustryOR |
|-------|--------|--------|-------------|----------------|-----------|------------|
| GPT-4o | Standard | 95.7 | 98.9 | 97.0 | 84.2 | 84.0 |
| | CoT | 99.6 | 99.7 | 97.5 | 84.2 | 91.0 |
| | CoE | 98.6 | 99.6 | 99.8 | 89.4 | 96.0 |
| | OptiMUS | 100.0 | 99.8 | 100.0 | 94.7 | 98.0 |
| | OptiTree | 100.0 | 100.0 | 100.0 | 100.0 | 98.0 |
| DeepSeek-V3 | Standard | 97.8 | 93.1 | 94.5 | 100.0 | 87.0 |
| | CoT | 95.2 | 98.0 | 97.5 | 84.2 | 86.0 |
| | CoE | 97.9 | 100.0 | 98.8 | 100.0 | 98.0 |
| | OptiMUS | 99.6 | 98.4 | 99.5 | 100.0 | 94.0 |
| | OptiTree | 100.0 | 100.0 | 100.0 | 100.0 | 99.0 |

## A.2    The Greatest Search Depth

We report the greatest search depth across different datasets. For the harder datasets, the search depth tends to increase, indicating that more complex problems feature more sophisticated subproblem structures.

Table 6: The greatest search depth

| | NL4Opt | MAMO EasyLP | MAMO ComplexLP | ComplexOR | IndustryOR |
|-------|--------|-------------|----------------|-----------|------------|
| Greatest Search Depth | 4 | 4 | 10 | 9 | 10 |

## A.3    More Evaluation Results

The objective metric is standard in the LLM-based optimization modeling literature [39, 1], and in subsequent works ORLM, LLMOPT, OptMATH, OptiBench. However, defining a robust "partial credit" metric is a significant challenge. Most existing benchmarks only contain labeled optimal value and **do not contain annotated ground-truth optimization model**, making the partial credit metric difficult to design. Even with ground-truth models, simple text-based or structural comparisons can be unreliable. A model may have minor syntactic differences from the ground truth but be semantically equivalent. Conversely, a model might be nearly identical yet contain a critical error, such as a misplaced inequality sign.

For further evaluation, we find that the training set in OptMATH has annotated models. We use 100 problems for evaluation. As our method and the baselines were not trained on this dataset, the comparison is safe and fair. We present the percentages of LLM-generated models that align with the ground-truth models based on statistical information, including the number of variables, binary variables, integer variables, constraints and objective values. Note that all the statistical information is just a reference, as the **mismatch of variables and constraints does not imply the model is incorrect**. Notably, OptiTree exhibits the highest matching ratio with the ground-truth model, underscoring its reliability.

Table 7: Evaluation of partial credit (in %)

| OptMATH | Var | Bin | Int | Cons | Obj |
|---|---|---|---|---|---|
| DeepSeek-R1 | 79.0 | 85.0 | 67.0 | 46.0 | 66.0 |
| OpenAI-o1 | 83.0 | 87.0 | **93.0** | 50.0 | 68.0 |
| ORLM | 43.0 | 52.0 | 44.0 | 35.0 | 37.0 |
| LLMOPT | 56.0 | 65.0 | 57.0 | 37.0 | 41.0 |
| CoE | 67.0 | 78.0 | 85.0 | 46.0 | 58.0 |
| OptiMUS | 84.0 | 79.0 | 76.0 | 43.0 | 61.0 |
| MCTS | 75.0 | 80.0 | 81.0 | 46.0 | 63.0 |
| Ours | **90.0** | **92.0** | 83.0 | **68.0** | **71.0** |

## A.4 Built OptiTree on More Models

We conduct experiments to build OptiTree on (1) stronger models, including GPT-o1 and DeepSeek-R1; (2) weaker models, including Qwen-2.5 7B and Llama3-8B. The results in Table 8 demonstrate that OptiTree can significantly improve the performance of the LLMs.

Table 8: Improvement on more models.

| | NL4Opt | MAMO EasyLP | MAMO ComplexLP | ComplexOR | IndustryOR |
|---|---|---|---|---|---|
| DeepSeek-R1 | 86.1 | 79.5 | 57.3 | 68.4 | 38.0 |
| Ours (DeepSeek-R1) | 97.9 | 96.9 | 82.5 | 84.2 | 57.0 |
| OpenAI-o1 | 87.1 | 87.6 | 54.5 | 73.6 | 40.0 |
| Ours (OpenAI-o1) | 96.5 | 95.1 | 83.9 | 84.2 | 53.0 |
| Qwen2.5 14B | 79.41 | 79.6 | 45.0 | 57.9 | 31.0 |
| Ours (Qwen2.5 14B) | 88.2 | 89.6 | 78.7 | 73.6 | 37.0 |
| Llama3.1-8B | 40.5 | 71.2 | 39.8 | 63.1 | 24.0 |
| Ours (Llama3.1-8B) | 55.0 | 75.9 | 64.4 | 73.7 | 28.0 |

## A.5 Constructing Modeling Tree with Different Datasets

We investigate the robustness of the datasets for tree construction. (1) For the first variant, we use 100 problems from MAMO EasyLP and 100 from ComplexLP, which we refer to as OptiTree (MAMO). (2) For the second variant, we use 100 problems from the OR-Instruct 3K dataset, representing only 25% of the problems used in our main experiments. We call this variant OptiTree (100 OR-Instruct). We evaluate performance on three challenging benchmarks: MAMO ComplexLP, ComplexOR, and IndustryOR. Unlike the OR-Instruct 3K dataset, the MAMO datasets do not include a ground-truth modeling process for tree construction; we rely solely on the final answers for this purpose. The results presented in Table 9 reveal two key findings. (1) Despite utilizing easier problems without a ground-truth modeling process, OptiTree (MAMO) achieves high performance, with only a slight drop compared to the original OptiTree. (2) OptiTree demonstrates data efficiency, as OptiTree (100 OR-Instruct) exhibits performance comparable to that of the full OptiTree. These findings suggest that common decomposition patterns and modeling techniques are widely applicable across OR problems, and the modeling tree effectively captures this general knowledge across different datasets.

Table 9: Constructing a modeling tree using different datasets.

| Model | Method | MAMO ComplexLP | ComplexOR | IndustryOR |
|---|---|---|---|---|
| GPT-4o | OptiTree (MAMO) | 84.4 | 79.0 | 44.0 |
| | OptiTree (100 OR-Instruct) | 81.0 | 84.2 | 46.0 |
| | OptiTree | 81.0 | 84.2 | 48.0 |
| DeepSeek-V3 | OptiTree (MAMO) | 83.9 | 79.0 | 46.0 |
| | OptiTree (100 OR-Instruct) | 81.5 | 84.2 | 54.0 |
| | OptiTree | 81.5 | 84.2 | 54.0 |

# B   Subproblem Identification Analysis

We analyze the structure of the modeling tree, where the statistical information of the tree is presented in Table 10. We also examine the problems represented within the modeling tree. We also analyze the subproblem matching distribution in different datasets. We find significantly different distributions of the sbuproblems on different datasets. The consistent superior performance of OptiTree across the datasets demonstrates the strong generalization ability across different problems from various scenarios and difficulty levels.

Table 10: Statistics of the modeling tree.

|                | DeepSeek-V3 | GPT-4o |
|----------------|-------------|--------|
| Average Degree | 2.42        | 2.74   |
| Depth          | 10          | 10     |
| Node Number    | 320         | 285    |

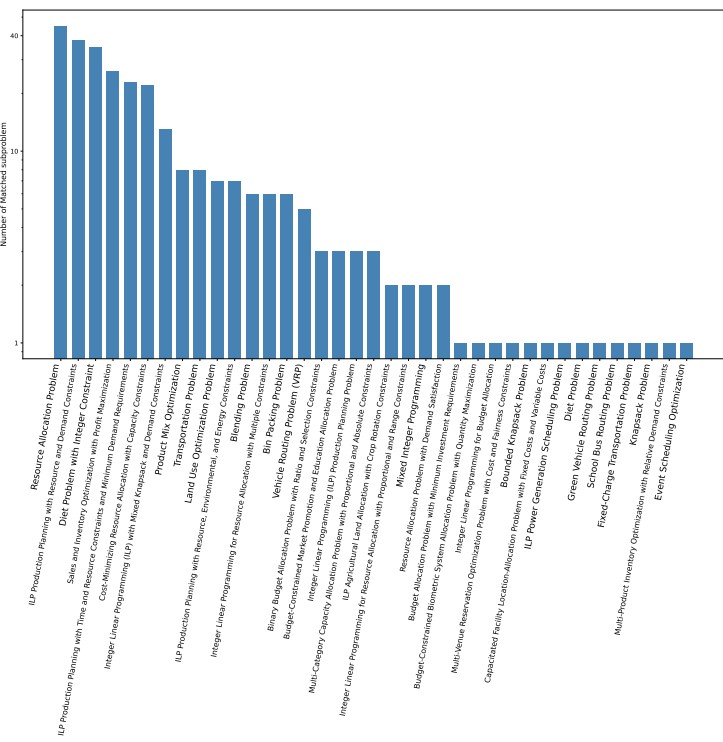

Figure 6: Subproblem distribution in NL4Opt.

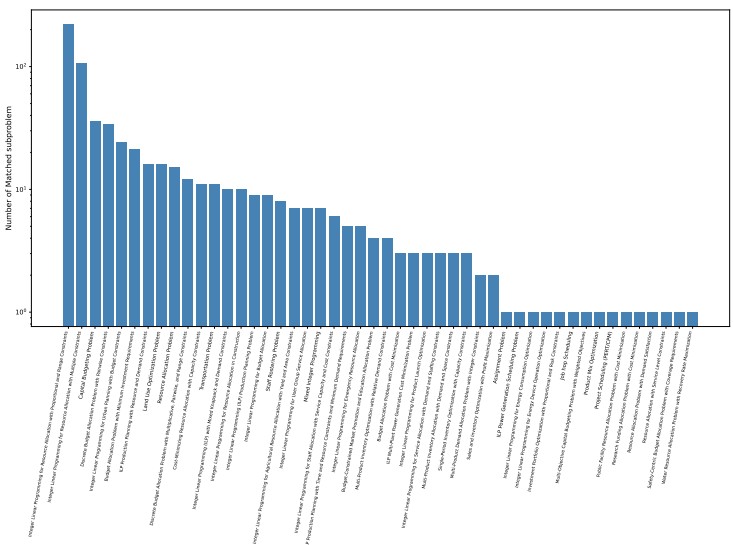

Figure 7: Subproblem distribution in MAMO EasyLP.

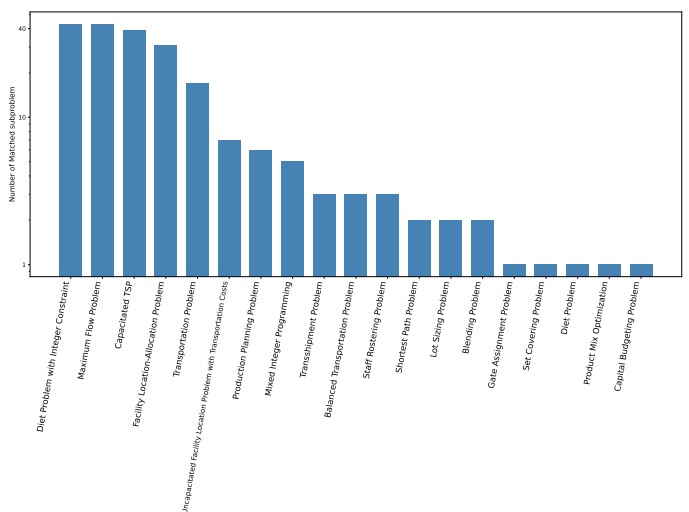

Figure 8: Subproblem distribution in MAMO ComplexLP.

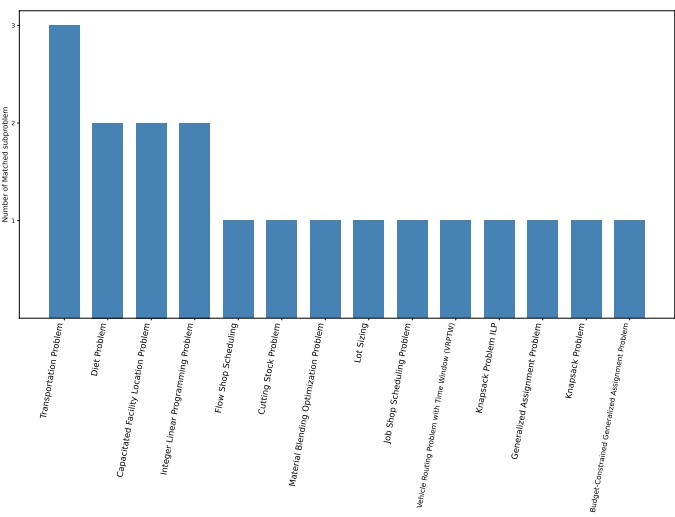

Figure 9: Subproblem distribution in ComplexOR.

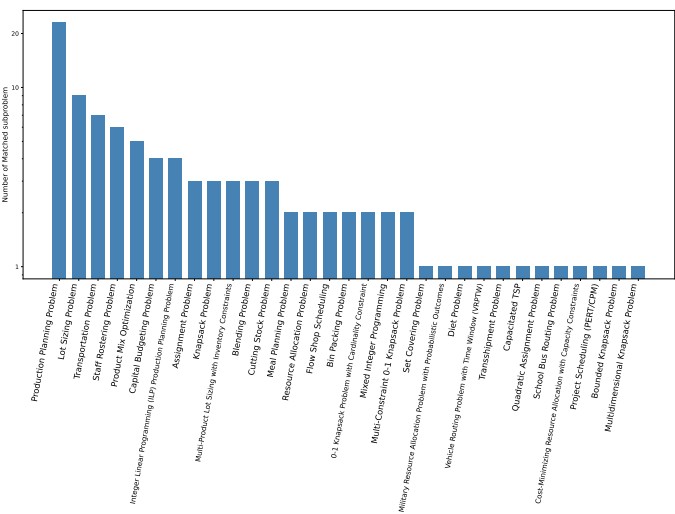

Figure 10: Subproblem distribution in IndustryOR.

# C Common Subproblem Decomposition Patterns

In combinatorial optimization, there are commonly subproblem decomposition patterns. Here, we provide observations and examples to illustrate these patterns. First, we can expand a subproblem into more complex problems using the following patterns.

- **Constraint Addition** introduces new constraints to model refined requirements.
- **Variable Expansion** incorporates new decision variables.
- **Dimensional Replication** replicates the problem structure across multiple items.

We provide the following examples to demonstrate the three decomposition patterns.

## C.1 Analysis on CVPR and CVRPTW

The Capacitated Vehicle Routing Problem (CVRP) aims to determine optimal routes for a fleet of vehicles to serve a set of customers from a central depot while minimizing total travel cost or distance. The basic CVRP formulation includes constraints ensuring each customer is visited exactly once, vehicles return to the depot, and the fleet size is respected. In contrast, the Capacitated Vehicle Routing Problem with Time Windows (CVRPTW) extends the CVRP by adding temporal constraints, requiring each customer to be served within a predefined time window. This introduces additional complexity, as routes must now satisfy both spatial routing constraints and temporal synchronization.

The optimization model for CVRP is as follows. $V$ is the set of nodes (depot 0 and customers), $c_{ij}$ is the travel cost from $i$ to $j$. $x_{ij}$ is the binary decision variable for route inclusion. $K$ is the fleet size.

$$\text{Minimize} \quad \sum_{i \in V} \sum_{j \in V} c_{ij} x_{ij}$$

$$\text{Subject to} \quad \sum_{j \in V} x_{ij} = 1, \quad \forall i \in V \setminus \{0\} \quad \text{(Customer visited once)}$$

$$\sum_{i \in V} x_{ij} = 1, \quad \forall j \in V \setminus \{0\} \quad \text{(Flow conservation)}$$

$$\sum_{j \in V} x_{0j} \leq K \quad \text{(Fleet size)}$$

$$\sum_{i \in S} \sum_{j \in S} x_{ij} \leq |S| - 1, \quad \forall S \subseteq V \setminus \{0\}, S \neq \emptyset \quad \text{(Subtour elimination)}$$

$$x_{ij} \in \{0, 1\}, \quad \forall i, j \in V$$

The optimization model for CVRPTW is as follows. $t_i$ is the service start time at node $i$. $[a_i, b_i]$ is the time window for node $i$. $s_i$ is the service duration at $i$. $d_{ij}$ is the travel time from $i$ to $j$. $M$ is the big-M number.

$$\text{Minimize} \quad \sum_{i \in V} \sum_{j \in V} c_{ij} x_{ij}$$

$$\text{Subject to} \quad \text{All VRP constraints}$$

$$a_i \leq t_i \leq b_i, \quad \forall i \in V \setminus \{0\} \quad \text{(Time window)}$$

$$t_j \geq t_i + s_i + d_{ij} - M(1 - x_{ij}), \quad \forall i, j \in V \setminus \{0\} \quad \text{(Time consistency)}$$

$$t_i \geq 0, \quad \forall i \in V$$

Notice that CVRP is a subproblem of CVRPTW. We identify the following decomposition patterns.

- Variable Expansion. The introduction of new continuous variables $t_i$ tracks service times, adding a temporal dimension to the original CVRP.

- Constraint Addition: CVRPTW introduces time windows $a_i \leq t_i \leq b_i$ and synchronization constraints (time consistency), which refine feasible solutions by eliminating temporally infeasible routes.

## C.2 Analysis on SCMCFP and MCNFP

The Single-Commodity Minimum-Cost Flow Problem (SCMCFP) optimizes the flow of a homogeneous good (e.g., water, electricity) through a network to meet supply/demand at nodes while minimizing costs, subject to arc capacity constraints. Its linear programming formulation tracks flow conservation and capacity limits for a single product type. In contrast, the Multi-Commodity Network Flow Optimization Problem (MCNFP) generalizes this by handling multiple distinct commodities (e.g., different goods in logistics, data streams in networks) sharing the same infrastructure. Each commodity has unique sources, sinks, and demands, coupled through shared arc capacity constraints.

The optimization model for SCMCFP is as follows. Suppose that $G = (V, E)$ is the network with nodes $V$ and directed arcs $E$. $b_i$ is the net supply $(> 0)$ or demand $(< 0)$ at node $i$. $c_{ij}$ is the cost per unit flow on arc $(i, j)$. $u_{ij}$ is the capacity of arc $(i, j)$. $x_{ij}$ is the flow decision variable on arc $(i, j)$.

$$\text{Minimize} \quad \sum_{(i,j)\in E} c_{ij} x_{ij}$$

$$\text{Subject to} \quad \sum_{j:(i,j)\in E} x_{ij} - \sum_{j:(j,i)\in E} x_{ji} = b_i, \quad \forall i \in V \quad \text{(Flow conservation)}$$

$$0 \le x_{ij} \le u_{ij}, \quad \forall (i,j) \in E \quad \text{(Capacity constraints)}$$

The optimization model for MCNFP is as follows. Suppose that $K$ is the set of commodities. $s_k, t_k$ is the source/sink nodes for commodity $k$. $d_k$ is the demand for commodity $k$. $c_{ij}^k$ is the commodity-specific arc cost. $x_{ij}^k$ is the flow of commodity $k$ on arc $(i, j)$.

$$\text{Minimize} \quad \sum_{k\in K} \sum_{(i,j)\in E} c_{ij}^k x_{ij}^k$$

$$\text{Subject to} \quad \sum_{j:(i,j)\in E} x_{ij}^k - \sum_{j:(j,i)\in E} x_{ji}^k = \begin{cases} d_k & \text{if } i = s_k \\ -d_k & \text{if } i = t_k \\ 0 & \text{otherwise} \end{cases}, \quad \forall i \in V, k \in K$$

$$\sum_{k\in K} x_{ij}^k \le u_{ij}, \quad \forall (i,j) \in E \quad \text{(Shared capacity)}$$

$$x_{ij}^k \ge 0, \quad \forall (i,j) \in E, k \in K$$

Notice that SCMCFP is a subproblem of MCNFP, where we identify the following pattern.

- Dimensional Replication: Adds commodity dimension $k \in K$ to flow variables.

# D    Algorithm of OptiTree

---

**Algorithm 1** Tree Search for Subproblem Decomposition

---

**Require:** Target problem $\mathcal{P}$, modeling tree $\mathbb{T}$
**Ensure:** Hierarchical modeling thoughts $\mathcal{T}(\mathcal{P})$, and the maximal subproblem $\tilde{\mathcal{P}}$

 1: **function** TREESEARCH($\mathcal{P}, \mathbb{T}$)
 2:      Initialize current node $\mathcal{N} \leftarrow$ The root of the modeling tree $\mathbb{T}$
 3:      Extract statement thoughts $\mathcal{C}_{\mathcal{P}}$
 4:      # Identified subproblems
 5:      **while** $\mathcal{N}$ has children **do**
 6:          Get the children of $\mathcal{N}$
 7:          # Prompt of Subproblem Identification in Appendix J.2
 8:          Identify the subproblem of $\mathcal{P}$ among the children with similarity scores
 9:          **if** $\mathcal{P}$ has a subproblem among the children **then**
10:             $\mathcal{N} \leftarrow$ Subproblem with the max similarity score
11:             The maximal subproblem $\tilde{\mathcal{P}} \leftarrow \mathcal{N}$
12:          **else**
13:             $\mathcal{N} \leftarrow$ None
14:          **end if**
15:      **end while**
16:      Synthesize $\mathcal{T}(\mathcal{P})$ from $\mathcal{T}(\tilde{\mathcal{P}})$
17:      **return** $\mathcal{T}(\mathcal{P}), \tilde{\mathcal{P}}$
18: **end function**

---

**Algorithm 2** Modeling Tree Update

---

**Require:** New problem $\mathcal{P}$, modeling tree $\mathbb{T}$
**Ensure:** Updated modeling tree

 1: **function** UPDATETREE($\mathcal{P}, \mathcal{T}$)
 2:      $\mathcal{T}(\mathcal{P}), \tilde{\mathcal{P}} \leftarrow$ TreeSearch($\mathcal{P}, \mathbb{T}$)
 3:      # Prompts Modeling with Modeling Thoughts in J.3
 4:      Model the problem $\mathcal{P}$ using the modeling thoughts and solve the model for optimal value $y$
 5:      **if** $y$ does not equal to the ground-truth objective value **then**
 6:          # Prompts of Modeling Thoughts Distillation in Appendix J.4
 7:          Distill schema (problem type,$\mathcal{C}_{\mathcal{P}}, \mathcal{T}(\mathcal{P})$) from $\mathcal{P}$
 8:          Find the children of the maximal subproblem $\tilde{\mathcal{P}}$
 9:          **for** child $\tilde{\mathcal{P}}_k$ in the children **do**
10:             # Prompt of Add New Tree Nodes in Appendix J.4
11:             **if** $\mathcal{P} \subset_{\mathcal{S}} \tilde{\mathcal{P}}_k$ **then**
12:                Insert $\mathcal{P}$ as parent of $\tilde{\mathcal{P}}_k$ and children of $\tilde{\mathcal{P}}$
13:             **else**
14:                Insert $\mathcal{P}$ as sibling of $\tilde{\mathcal{P}}_k$ and children of $\tilde{\mathcal{P}}$
15:             **end if**
16:          **end for**
17:      **end if**
18:      **return** $\mathbb{T}$
19: **end function**

---

# E More Related Works

## E.1 Machine Learning for Accelerating Mathematical Optimization Solving

Mathematical optimization finds broad applications across critical domains such as chip design [38] and optimization [31]. Driven by advances in graph models [27] and generative models [26], machine learning techniques have been widely adopted to speed up combinatorial optimization solving [5]. These efforts follow main directions: improving key modules in branch-and-bound algorithms [11, 34], and predicting high-quality solutions to warm-start solvers [24, 15, 19].

# F Proof of Proposition 4.2

**Lemma F.1** (Transitivity of $\subseteq_\mathcal{S}$)**.** *Given three problems $\mathcal{P}_1, \mathcal{P}_2$ and $\mathcal{P}_3$, if $\mathcal{P}_1 \subseteq_\mathcal{S} \mathcal{P}_2$ and $\mathcal{P}_2 \subseteq_\mathcal{S} \mathcal{P}_3$, then $\mathcal{P}_1 \subseteq_\mathcal{S} \mathcal{P}_3$.*

*Proof.* We prove by induction. If the initial tree contains only the root node (abstract OR problem). The property holds trivially. Assume the modeling tree is subproblem order-preserving after $k$ insertions. We show that the tree preserves the property when adding a new node $\mathcal{P}$. We Identify the maximum subproblem $\mathcal{P}^{(M)}$ for $\mathcal{P}$. For each child $\mathcal{P}_t^{(M)}$ of $\mathcal{P}^{(M)}$, $\mathcal{P}_t^{(M)}$ cannot be the subproblem of $\mathcal{P}$. Thus, $\mathcal{P}$ cannot be in the descendent of $\mathcal{P}_t^{(M)}$. We have the following two situations.

**Case 1 Analysis** ($\mathcal{P}$ becomes parent of $\mathcal{P}_t^{(M)}$). By insertion rule, we have $\mathcal{P}^{(M)} \subseteq_\mathcal{S} \mathcal{P}$. By assumption we have $\mathcal{P} \subseteq_\mathcal{S} \mathcal{P}_t^{(M)}$. Finally, the transitivity ensures all ancestors of $\mathcal{P}^{(M)}$ satisfy $\mathcal{P}' \subseteq_\mathcal{S} \mathcal{P}_t^{(M)}$.

**Case 2 Analysis** ($\mathcal{P}$ becomes sibling of $\mathcal{P}_t^{(M)}$). First, the operation maintains $\mathcal{P}^{(M)} \subseteq_\mathcal{S} \mathcal{P}$ and $\mathcal{P}^{(M)} \subseteq_\mathcal{S} \mathcal{P}_t^{(M)}$. We notice that no new ancestor-descendant relationships are created between $\mathcal{P}$ and siblings, and existing subproblem relations are preserved through shared parent $\mathcal{P}^{(M)}$.

Thus, the modeling tree remains subproblem order-preserving. By induction, the proposition holds for all tree updates. $\square$

# G Limitations and Failure Cases

Following OptiMUS [1], the errors mainly come from three aspects: missing or wrong constraints, incorrect modeling and coding errors. In [1], almost half of the errors come from the incorrect modeling (62.5% for ComplexOR), e.g., defining wrong binary variables in the model. We analyze the failures of OptiTree and normalize the failure rates to sum to 1. We find that most errors are from the missing constraints. OptiTree can significantly reduce the error of incorrect modeling, which mainly comes from the wrong variable definitions. The constraint errors are the main error for OptiTree. First, accurately formulating constraints requires a deep and nuanced understanding of the specific problem scenario and its domain knowledge. For highly specialized or novel OR problems, existing background knowledge may not fully capture the intricate details necessary for precise constraint definitions. The second limitation is the intrinsic **reasoning ability of the base LLM**. While OptiTree provides a structured framework and knowledge, the final synthesis depends on the LLM's inherent reasoning capabilities. This framework may struggle with problems requiring significant logical or relational reasoning. For instance, in the staff rostering problem, the model must allocate staff for 24-hour shifts to meet coverage constraints (with workers working 8-hour days). If it fails to recognize that the demand period from 1:00 to 2:00 AM is covered by workers who began their shifts at 8:00 PM, it will result in an incorrect constraint and ultimately a flawed model.

Table 11: Failure cases information

| | ComplexOR | IndustryOR | OptMATH |
|---|---|---|---|
| Incorrect modeling | 0 | 10.9 | 23.8 |
| Missing constraints | 66.7 | 82.6 | 61.9 |
| Coding errors | 33.3 | 6.5 | 14.3 |

# H  Case Analysis

## H.1  Example: Inventory Problem

We provide a detailed analysis of an example from the **IndustryOR** dataset. This example demonstrates that while **OptiTree successfully corrects fundamental structural errors** (what [1] terms "Incorrect Modeling"), it still faces challenges with constraints that require deep, context-specific **logical reasoning**.

**Problem**. A factory requires a specialized tool for $n$ planning stages. $r_j$ specialized tools are needed. At the end of this stage, all tools used within this stage must be sent for repair. There are two repair methods: one is slow repair, which is cheaper (costs $b$ per tool) but takes longer ($p = 3$ stages to return); the other is fast repair, which costs $c$ per tool and requires $q = 1$ stages to return. If the tools cannot meet the needs, new ones must be purchased, with a cost of $a$ per new tool. Determine an optimal plan to minimize the cost.

**The Errors of the Standard Output Model**    The standard output from DeepSeek-V3 produces a structurally flawed model. It tries to fulfill demand by combining new purchases with tools returning from repair, but it completely lacks any concept of inventory tools. The model given by the Standard output of DeepSeek-V3 is as follows. Let $x_j \geq 0$ be the new tools purchased at stage $j$, $s_j \geq 0$ be the tools sent to slow repair at stage $j$, and $f_j \geq 0$ be the tools sent to fast repair at stage $j$.

$$\min \quad \sum_{j=1}^{n} (ax_j + bs_j + cf_j)$$
$$\text{s.t.} \quad s_j + f_j = r_j \quad \forall j \in \{1, \dots, n\} \qquad \text{(Repair Requirement)}$$
$$x_j + s_{k-p} + f_{k-q} \geq r_j \quad \forall j \in \{1, \dots, n\} \qquad \text{(Demand Fulfillment)}$$

Following the errors described in OptiMUS [1], the model output has the error type of **Incorrect modeling**, which fails to define the **Inventory variables** $I$.

**The Improvement of OptiTree**    OptiTree correctly identifies that this problem contains an **Inventory Problem** subproblem. By applying the modeling thoughts, it generates a much-improved model that correctly introduces inventory variables $I$ and a proper inventory balance constraint. The model given by OptiTree (DeepSeek-V3) is as follows. Let $x_j \in \mathbb{Z}_+$ be the new tools purchased at stage $j$, $s_j \in \mathbb{Z}_+$ be the tools sent to slow repair at stage $j$, $f_j \in \mathbb{Z}_+$ be the tools sent to fast repair at stage $j$, and $I_j \in \mathbb{Z}_+$ be the inventory of available tools at start of stage $j$. OptiTree finds a **subproblem of the Inventory Problem** for the original problem.

$$\min \quad \sum_{j=1}^{n} (ax_j + bs_j + cf_j)$$
$$\text{s.t.} \quad I_1 = 0 \qquad \text{(Initial Inventory)}$$
$$s_j + f_j = r_j \quad \forall j \in \{1, \dots, n\} \qquad \text{(Repair Requirement)}$$
$$I_{j+1} = (I_j + x_j - r_j) + s_{j-p} + f_{j-q} \quad \forall j \in \{1, \dots, n\} \qquad \text{(Inventory Balance)}$$
$$I_j + x_j \geq r_j \quad \forall j \in \{1, \dots, n\} \qquad \text{(Demand Fulfillment)}$$

**The Errors in the Model Given by OptiTree**    Despite the structural correction, the OptiTree model still contains a subtle error that requires **common-sense logical reasoning**. The model does not

account for the finite 10-stage horizon. A human modeler would reason that it is illogical to pay for a repair that will finish after the tools are no longer needed. Slow repairs (taking $p = 3$ stages) are disabled for the last 3 stages ($j = 8, 9, 10$) since tools sent for repair at these stages wouldn't return by stage 10, and fast repairs (taking $q = 1$ stage) are disabled for the final stage ($j = 10$) for the same reason. The repair requirement uses inequality

$$s_j + f_j \leq r_j \quad \text{(Repair Requirement)}$$

rather than equality $s_j + f_j = r_j$, allowing flexibility to not repair tools when unnecessary—particularly near the end of the horizon. Thus, the final corrected model should be

$$
\begin{aligned}
\min \quad & \sum_{j=1}^{n} (a x_j + b s_j + c f_j) \\
\text{s.t.} \quad & I_1 = 0 && \text{(Initial Inventory)} \\
& s_j + f_j \leq r_j \quad \forall j \in \{1, \ldots, n\} && \text{(Repair Requirement)} \\
& I_{j+1} = (I_j + x_j - r_j) + s_{j-p} + f_{j-q} \quad \forall j \in \{1, \ldots, n\} && \text{(Inventory Balance)} \\
& I_j + x_j \geq r_j \quad \forall j \in \{1, \ldots, n\} && \text{(Demand Fulfillment)}
\end{aligned}
$$

This demonstrates a limitation where OptiTree applies a general pattern correctly but misses a nuanced, context-dependent constraint. The error stems not from a lack of OR knowledge patterns but from a gap in the base LLM's logical reasoning about the problem's specific context. We believe this limitation will diminish as the underlying reasoning capabilities of LLMs continue to improve.

# I  Broader Impacts

This paper focuses on the automatic optimization modeling in operations research. The traditional modeling process often demands significant human effort and specialized domain knowledge, making it both costly and time-consuming. Our work aims to enhance the efficiency and accuracy of this process. We introduce a novel tree-based method for organizing operations research problem data. Unlike existing approaches that treat each problem individually, our modeling tree leverages the relationships between different problems. During the modeling process, we can retrieve and utilize valuable modeling thoughts to improve overall effectiveness. Additionally, our research offers new perspectives on organizing, utilizing, and managing knowledge within the field of operations research.

# J   Prompts for LLMs

In this part, we present the meta-prompts we used in this paper.

## J.1   Schema in OptiTree

> **Schema**
>
> Problem Type: Diet Problem with Integer Constraint,
>
> Statement Thoughts: [
>     Statement Thoughts: The Diet Problem with Integer Constraint involves
>         determining the quantity of each food item that should be
>         consumed to meet nutritional requirements while minimizing cost.
>         The twist is that the quantities must be integers, representing
>         the whole number of servings or portions of each food item. The
>         problem is formulated to ensure nutritional constraints(such as
>         calories, protein, vitamins, etc.) are satisfied, and the total
>         cost is minimized.,
>     Nutritional Constraints: Ensure that the selected food items provide
>         at least the required amount of nutrients, such as calories,
>         proteins, fats, vitamins, and minerals.,
>     Cost Minimization: The total cost of the selected food items should
>         be minimized while meeting nutritional constraints.,
>     Integer Servings: The servings of food items must be integer values,
>         representing realistic portions of each food item.
> ],
>
> Modeling Thoughts: [
>     [Define Decision Variables] Define decision variables that represent
>         the integer quantities of servings for each food item.,
>     [Define Objective Function] Minimize the total cost of the selected
>         food items.,
>     [Define Nutritional Constraints] Ensure that the nutrient intake
>         meets or exceeds the minimum required levels for each nutrient.,
>     [Implement Integer Constraints] Ensure integer constraints for the
>         serving sizes of each food item in the model.,
>     [Comprehensive Verification] Check the common errors in the
>         optimization model.,
>     [Write Gurobi Code] Write the Gurobi code to solve the problem.,
>     [Gurobi Code] \n'''python\nimport json\nimport numpy as np\nimport
>         math\nimport gurobipy as gp\nfrom gurobipy import GRB\n\n# Create
>         a new model\nmodel = gp.Model('model')\n\n# define parameters\n\
>         n# define variables\n\n# define constraints\n\n# define objective
>         \n\n# Optimize the model\nmodel.optimize()\nstatus = model.
>         status\n\nobj_val = None\n# Check whether the model is infeasible,
>         has infinite solutions, or has an optimal solution\nif status ==
>         gp.GRB.INFEASIBLE:\n obj_val = \infeasible\\nelif status == gp.
>         GRB.UNBOUNDED:\n obj_val = \unbounded\\nelif status == gp.GRB.
>         OPTIMAL:\n obj_val = model.objVal\ntime = model.TimeLimit\nprint
>         (\Timecost\:,time)\nprint(\Objective Value:\, obj_val)\n''',
>     [error_tips]
>         Ensure decision variables are defined as integers.,
>         Confirm that all nutritional requirements are modeled correctly
>             and constraints are set accurately.,
>         Review that cost coefficients are correctly assigned in the
>             objective function.,

```
        gurobi_example: Example not provided since specific problem data
            is omitted; follow reasoning flow for formulation specifics.
]
```

## J.2 Prompts for Subproblem Identification

**Subproblem Identification**

```
You are a mathematical formulator working with a team of optimization
    experts. The objective is to tackle a complex optimization problem.

You are given a specific problem and its current basic problem type. You
    are also provided with a list of subtypes for this basic problem type
    .
Input problem: {input_problem}.
Its current basic problem type:{basic_type}

You are given a list of defined subtypes and the statement thoughts of
    subtypes: {statement_thought_info}.

Your task is to determine if the specific problem belongs to one of the
    given subtypes. If it does, return the subtype directly. Importantly,
     you must return the only subtype verbatim(don't return the statement
     thoughts), which is provided in the list. If it does not, return
    subtype not found.

Also, return a boolean value indicating whether the input problem belongs
     to the defined subtypes. Return only the final answer in the
    following JSON format:

```json
{{
    matching_subtype: <matching_subtype>,
    reasoning: <reasoning_process>
    belongs_to_subtypes: <boolean_value>
}}
```

- Note that I'm going to use Python JSON.loads() function to parse the
    JSON file, so please make sure the format is correct (don't add ','
    before enclosing '}}' or ']' characters.
- Generate the complete JSON file and don't omit anything.
- Use '```json' and '```' to enclose the json file.
```

## J.3 Modeling

**Modeling with Modeling Thoughts**

```
You are a mathematical formulator working with a team of optimization
    experts. The objective is to tackle a complex optimization problem.
```

```
You are an expert in problem analysis and can apply previous problem-
    solving approaches to new issues. The user will provide a specific
    task statement thoughts and a modeling thought. Your goal is to
    analyze the user's task and generate a specific solution based on the
     modeling thought. If the instantiated solution involves Python code,
     only provide the code and let the compiler handle it. If the
    solution does not involve code, provide a final answer that is easy
    to extract from the text.
It should be noted that all the Python code should be within one code
    block; the answer should not include more than one code block! And
    strictly follow the modeling thought to instantiate the Gurobi code,
    but you should also adjust the input parameter according to the user
    input!

User Input:
{user_input}
modeling thought:
{modeling thought}

Please analyze the above user task statement thoughts and modeling
    thought, and generate a specific, detailed solution. If the solution
    involves Python code, only provide the code. If not, provide a clear
    and extractable final answer.
```

**Code Correction**

```
You are an excellent Python programming master who is proficient in
    analyzing and editing Python code, and you are also good at
    understanding real-world problems. Your task is:
1. Analyze the given Python code
2. Edit the input code to make sure the edited code is correct and can
    run and solve the problem correctly.
Your response should follow the format below:
'''python
## Edited code here
'''
```

## J.4    Tree Update

**Statement Thoughts Distillation stopping at the root node**

```
You are given a specific combinatorial optimization problem.

Your task is to summarize the industrial scene type of this specific
    problem and provide a detailed statement thoughts of this type.

Importantly, you must classify the problem into more precise industrial
    scenarios, such as the Traveling Salesman Problem (TSP), facility
    location problem, Parallel Machine Scheduling, and so on. Avoid using
     broad categories such as linear programming, mixed-integer
    optimization, integer optimization, Integer Linear Programming
    Problem, and so on.

- **Specific Problem**: {specific_problem}
```

```
Please provide the following information in JSON format:

‘‘‘json
{{
    "industrial_scene_type": "<industrial_scene_type>",
    "statement thoughts_of_type":{{
    "statement thoughts": "<the more precise of statement thoughts>",
    "constraints":
    {{
      "<get the Constraint 1>": "Detailed description of constraint 1",
      "<get the Constraint 2>": "Detailed description of constraint 2"
    }}
    }}
}}
‘‘‘
Here is an output example:
{{
    ’industrial_scene_type’: ’Maximum Flow Problem’,
    ’statement thoughts_of_type’: {{
    ’statement thoughts’: ’The Maximum Flow Problem involves determining
        the highest possible flow that can be routed through a directed
        graph from a specified source node to a sink node, while adhering
         to the capacity limits of the edges. This problem is
        foundational in network flow theory and has applications in
        transportation networks, communication systems, supply chain
        logistics, and resource distribution. The solution must respect
        edge capacities, flow directionality, and conservation laws at
        intermediate nodes.’
    ’constraints’: {
        ’Directed Graph’: ’Flow can only travel in the direction
            specified by the edges in the graph.’,
        ’Capacity Constraints’: "The flow on each edge must be non-
            negative and cannot exceed the edge’s maximum capacity.",
        ’Flow Conservation’: ’For every node except the source and sink,
            the total incoming flow must equal the total outgoing flow.’
    }},
}}
}}
```

**Statement Thoughts Distillation stopping at other node**

```
You are a mathematical formulator working with a team of optimization
    experts. The objective is to tackle a complex optimization problem.

You are given a specific problem, its current basic problem type, and the
     statement thoughts of the basic problem type.

Your task is to determine the more specific subtype of the given basic
    problem type that the specific problem belongs to, and provide a more
     detailed statement thoughts of this subtype.

- **Specific Problem**: {specific_problem}
- **Current Basic Problem Type**: {current_basic_problem_type}
- **statement thoughts of Basic Problem Type**: {statement
    thoughts_of_basic_problem_type}

Please provide the following information in JSON format:
```

```json
{{
    current_basic_problem_type: {current_basic_problem_type},
    statement thoughts_of_basic_problem_type: {statement
        thoughts_of_basic_problem_type},
    formulated_subtype: <subtype>,
    statement thoughts_of_subtype:{{
    statement thoughts: <the more precise of statement thoughts>,
    constraints:
    {{
      <get the Constraint 1>: "Detailed description of constraint 1",
      "<get the Constraint 2>": "Detailed description of constraint 2"
    }}
}}
}}
```

## Modeling Thoughts Distillation based on Statement Thoughts

You are a mathematical formulator working with a team of optimization
    experts. The objective is to tackle a complex optimization problem.

Please list the steps to formulate a {problem_type} problem and use the
    Gurobi code to solve it. You need to record some errors that are easy
    to make during the formulation process. Please output a JSON format.

You are given a specific combinatorial optimization problem, its solution
    process, and the problem type, along with its statement thoughts.

- **Problem Type**: {problem_type}
- **statement thoughts of Problem Type**: {statement thoughts}
- **Specific Problem**: {specific_problem}
- **Solution step of Specific Problem**: {solution_step}

Your task is to return a modeling thought for this problem type, which
    includes the following five parts:

1. **problem_type**: The provided problem type.
2. **statement thoughts**: The provided statement thoughts of the problem
    type.
3. **reason_flow**: A detailed step-by-step reasoning process for solving
    a series of problems that belong to a problem type, according to the
    provided solution process of the specific problem.
4. **example_application**: A detailed example application that matches
    the specific problem and its solution process.
5. **increment**: a list.

Additionally, in the solution steps, the Gurobi code is included. You
    must only use the fixed Gurobi code mentioned below in the solution
    steps. This is the fixed Gurobi code ---- "### Gurobi Code:\n```
    python\nimport json\nimport numpy as np\nimport math\nimport gurobipy
    as gp\nfrom gurobipy import GRB\n\n# Create a new model\nmodel = gp.
    Model('model')\n\n# define parameters\n\n# define variables\n\n#

```
      define constraints\n\n# define objective \n\n# Optimize the model\
      nmodel.optimize()\nstatus = model.status\n\nobj_val = None\n# Check
      whether the model is infeasible, has infinite solutions, or has an
      optimal solution\nif status == gp.GRB.INFEASIBLE:\n obj_val = \"
      infeasible\"\nelif status == gp.GRB.UNBOUNDED:\n obj_val = \"
      unbounded\"\nelif status == gp.GRB.OPTIMAL:\n obj_val = model.objVal\
      ntime = model.TimeLimit\nprint(\"Timecost\":,time)\nprint(\"Objective
       Value:\", obj_val)\n‘‘‘"
Please provide the following information in JSON format:

Here is a modeling thought example:

{{
"Problem Type": "Travelling Sales Person Problem",
’statement thoughts’: {{
    ’statement thoughts’: "The Transportation Problem involves optimizing
         the shipment of goods from multiple distribution centers to
        various destinations to minimize total transportation costs while
         meeting all destination demands. In this scenario, a company
        with four distribution centers (A, B, C, D) must supply five
        destinations (1, 2, 3, 4, 5) such that each destination’s demand
        is fully satisfied. The key objective is to determine the optimal
         shipment quantities from each center to each destination that
        result in the lowest possible transportation costs. This problem
        is a linear programming model where decision variables represent
        the amount shipped from each center to each destination, subject
        to supply and demand limitations.",
    ’constraints’: {{’Supply Constraints’: ’The total quantity of goods
        transported from each distribution center to all destinations
        must not exceed the available supply at that center.’, ’Demand
        Constraints’: "The total quantity of goods received by each
        destination from all distribution centers must exactly match the
        destination’s specified demand.", ’Non-Negative Transportation’: ’
        The amount of goods transported from any distribution center to a
         destination must be non-negative (i.e., shipments cannot have
        negative quantities).’}}
    }}
"Modeling Thoughts": [
    "[Define Decision Variables] Define decision variables for edges \\(
        x_{{ij}} \\) and possibly auxiliary variables for MTZ \\( u_i \\)
        ",
    "[Define Objective Function] Sum of distances multiplied by \\( x_{{
        ij}} \\)",
    "[Define Degree Constraints] Each node entered and exited exactly
        once",
    "[Define Subtour Elimination Constraints] Subtour elimination via MTZ
         or callbacks",
    "[Comprehensive Verification] Check the common errors in the
        optimization model",
    "[Write Gurobi Code] Write the Gurobi code the solve the problem.",
    "[Gurobi Code]\n‘‘‘python\nimport json\nimport numpy as np\nimport
        math\nimport gurobipy as gp\nfrom gurobipy import GRB\n\n# Create
         a new model\nmodel = gp.Model(’model’)\n\n# define parameters\n\
        n# define variables\n\n# define constraints\n\n# define objective
         \n\n# Optimize the model\nmodel.optimize()\nstatus = model.
        status\n\nobj_val = None\n# Check whether the model is infeasible,
         has infinite solutions, or has an optimal solution\nif status ==
         gp.GRB.INFEASIBLE:\n obj_val = \"infeasible\"\nelif status == gp.
```

```
                GRB.UNBOUNDED:\n obj_val = \"unbounded\"\nelif status == gp.GRB.
                OPTIMAL:\n obj_val = model.objVal\ntime = model.TimeLimit\nprint
                (\"Timecost\":,time)\nprint(\"Objective Value:\", obj_val)\n'''",
        "[Common Errors to Avoid]\n1. **Incorrect Subtour Elimination**:
                Ensure MTZ constraints exclude the starting city and are applied
                to correct indices.\n2. **Indexing Mistakes**: Use consistent 0-
                based or 1-based indexing for cities.\n3. **Self-Loops**:
                Explicitly disable \\( x_{{ii}} \\) variables.\n4. **Bounds on
                MTZ Variables**: Set \\( u_i \\) bounds correctly (\\( 1 \\leq
                u_i \\leq n-1 \\)).\n5. **Objective Function**: Ensure distances
                are correctly paired with \\( x_{{ij}} \\) and exclude \\( i = j
                \\).\n\nBy following these steps and avoiding common pitfalls,
                you can effectively model and solve TSP using Gurobi."
],

}}

Important:
- Use plain JSON without markdown syntax
- Ensure all quotes are properly escaped
- Include all required keys: problem_type, statement thoughts,
    reason_flow, example_application,increment
```

## Add New Tree Nodes

You are a mathematical formulator working with a team of optimization
    experts. The objective is to tackle a complex optimization problem.

You are given a primary problem type and its statement thoughts. You are
    also provided with a list of other problem types, each with its
    statement thoughts.

According statement thoughts, your task is to determine which problem
    types in the list are subtypes of the given primary problem type.
    Return a list of problem types that are identified as subtypes of the
    primary problem type. To be more specific, the subtype contains the
    constraint form of the primary problem type.
Importantly, the returned subtypes must be provided in the list; if there
    is any problem type that is a subtype of the given primary problem
    type in the list, return an empty list.

- **Primary Problem Type**: \{primary_problem_type\}
- **statement thoughts of Primary Problem Type**: \{statement
    thoughts_type\}
- **List of Problem Types**: \{list_of_problem_types\}

Please provide the following information in JSON format:

```json
{{
    "primary_problem_type": "{primary_problem_type}",
    "matching_subtypes": ["<problem_type>"]

}}
```

# K Examples

To better understand OptiTree, we provide two examples of OptiTree for each of the five datasets.

## K.1 NL4Opt

---

**NL4Opt**

**Natural Language Description:**
There is 1000 mg of gold available that is needed to make long and short cables. Long cables require 10 mg of gold while short cables require 7 mg of gold. Because of their compact size, at least 5 times the number of short cables are needed than the long cables. In addition, there needs to be at least 10 long cables made. If each long cable sold results in a $12 profit and each short cable sold results in a $5 profit, how many of each type of cable should be made to maximize profit?

------------------------------------------------------------------

**Subproblem Identification Process:**

```
{
    "matching_problem_type": "Product Mix Optimization",
    "reasoning": "The problem involves determining the optimal number of
        long and short cables to produce given constraints on resource
        availability (gold) and demand (minimum number of cables), with
        the objective of maximizing profit. This aligns with the 'Product
        Mix Optimization' problem type, which deals with optimizing the
        mix of products to produce under resource constraints to maximize
        profit or minimize cost.",
    "belongs_to_problem_types": true
}

{
    "matching_subtype": "Sales and Inventory Optimization with Profit
        Maximization",
    "reasoning": "The problem involves determining the optimal number of
        long and short cables to produce to maximize profit, considering
        constraints on the available gold and the ratio of short to long
        cables. This aligns with the 'Sales and Inventory Optimization
        with Profit Maximization' subtype, which focuses on maximizing
        profit under given constraints, including storage (in this case,
        gold availability) and integer constraints on production
        quantities.",
    "belongs_to_subtypes": true
}

{
    "matching_subtype": "subtype not find",
    "reasoning": "The input problem involves maximizing profit by
        determining the number of long and short cables to produce,
        considering gold availability and production constraints. It does
        not involve inventory holding costs, selling prices, demand, or
        warehouse capacity constraints, which are key features of the '
        Single-Period Inventory Optimization with Capacity Constraints'
        subtype. Therefore, it does not match the provided subtype.",
    "belongs_to_subtypes": false
}
```

------------------------------------------------------------------

---

**Get the Modeling Thoughts:**

```
{
'Problem Type': 'Sales and Inventory Optimization with Profit
    Maximization',
'Statement Thoughts':
'This subtype of Product Mix Optimization focuses on determining the
    optimal sales and inventory quantities for different products to
    maximize total profit while considering inventory holding costs and
    storage space limitations. The problem involves balancing the trade-
    off between sales revenue and inventory costs under given
    constraints.Storage Space Constraint-The total inventory quantity of
     all products cannot exceed a specified limit (e.g., 1000 units in
    this case).Integer Constraint-Sales and inventory quantities must be
     integers due to the indivisibility of the products.',
'Modeling Thoughts':[
    '[Define Decision Variables] Define integer decision variables for
        sales and inventory quantities for each product.',
    '[Define Objective Function] Maximize total profit by considering
        sales revenue and subtracting inventory holding costs.',
    '[Define Constraints] Add constraints for storage space limitations
        and ensure non-negativity of variables.',
    '[Comprehensive Verification] Check for common errors such as
        incorrect profit calculation or constraint formulation.',
    '[Write Gurobi Code] Write the Gurobi code to solve the problem.',
    '[Gurobi Code]:', '```python', 'import json', 'import numpy as np', '
        import math', 'import gurobipy as gp', 'from gurobipy import GRB
        ', '# Create a new model', "model = gp.Model('model')", '# define
         parameters', '# define variables', 'x = model.addVar(vtype=GRB.
        INTEGER, name="x")', 'y = model.addVar(vtype=GRB.INTEGER, name="y
        ")', '# define constraints', 'model.addConstr(x + y <= 1000, name
        ="InventorySpaceConstraint")', '# define objective', 'model.
        setObjective(8*x + 7*y, GRB.MAXIMIZE)', '# Optimize the model', '
        model.optimize()', 'status = model.status', 'obj_val = None', '#
        Check whether the model is infeasible, has infinite solutions, or
         has an optimal solution', 'if status == gp.GRB.INFEASIBLE:', '
        obj_val = "infeasible"', 'elif status == gp.GRB.UNBOUNDED:', '
        obj_val = "unbounded"', 'elif status == gp.GRB.OPTIMAL:', '
        obj_val = model.objVal', 'time = model.Runtime', 'print("Timecost
        :", time)', 'print("Objective Value:", obj_val)', '```',
    '[Common Errors to Avoid]', '1. **Incorrect Profit Calculation**:
        Ensure the objective function correctly accounts for both sales
        revenue and inventory costs.', '2. **Constraint Formulation**:
        Verify that the storage space constraint is correctly formulated
        and does not exceed the specified limit.', '3. **Variable Types
        **: Ensure variables are defined as integers to comply with the
        indivisibility of products.', '4. **Non-Negativity**: Explicitly
        enforce non-negativity constraints if not implicitly included.',
        '5. **Model Status Check**: Always check the model status to
        handle infeasible or unbounded cases appropriately.'
]

}
```

------------------------------------------------------------------------

**Solver Code:**

```
import gurobipy as gp
from gurobipy import GRB

# Create a new model
model = gp.Model('CableProduction')

# Define decision variables
long_cables = model.addVar(vtype=GRB.INTEGER, name="long_cables")
short_cables = model.addVar(vtype=GRB.INTEGER, name="short_cables")

# Define objective function: maximize profit
model.setObjective(12 * long_cables + 5 * short_cables, GRB.MAXIMIZE)

# Define constraints
# Gold usage constraint: 10*long + 7*short <= 1000
model.addConstr(10 * long_cables + 7 * short_cables <= 1000, name="
    GoldConstraint")

# Ratio constraint: short >= 5*long
model.addConstr(short_cables >= 5 * long_cables, name="RatioConstraint")

# Minimum long cables constraint
model.addConstr(long_cables >= 10, name="MinLongCables")

# Optimize the model
model.optimize()

# Check and display results
if model.status == GRB.OPTIMAL:
    print(f"Optimal number of long cables: {long_cables.x}")
    print(f"Optimal number of short cables: {short_cables.x}")
    print(f"Maximum profit: ${model.objVal}")
elif model.status == GRB.INFEASIBLE:
    print("Model is infeasible")
elif model.status == GRB.UNBOUNDED:
    print("Model is unbounded")
```

## NL4Opt

**Natural Language Description:**
A patient can be hooked up to two machines to have medicine delivered, machine 1 and machine 2. Machine 1 delivers 0.5 units of medicine to the heart per minute and 0.8 units of medicine per minute to the brain. Machine 2 delivers 0.3 units of medicine per minute to the heart and 1 unit of medicine per minute to the brain. In addition however, machine 1 creates 0.3 units of waste per minute while machine 2 creates 0.5 units of waste per minute. If at most 8 units of medicine can be received by the heart and at least 4 units of medicine should be received by the brain, how many minutes should each machine be used to minimize the total amount of waste produced?

- - - - - - - - - - - - - - - - - - - - - - - - - - - - - - - - - - - - - - - - - - - - - - - - - - - - - - - - - - - - - - - - - - - - - - -

**Subproblem Identification Process:**

```
{
    "matching_problem_type": "Resource Allocation Problem",
    "reasoning": "The problem involves allocating resources (time on
        machines) to meet certain constraints (medicine delivery to heart
         and brain) while minimizing waste, which fits the definition of
        a Resource Allocation Problem.",
```

```
    "belongs_to_problem_types": true
}

{
    "matching_subtype": "Cost-Minimizing Resource Allocation with
        Capacity Constraints",
    "reasoning": "The problem involves allocating resources (machine
        usage time) to minimize total waste (cost) while meeting specific
         medicine delivery requirements (capacity constraints) for the
        heart and brain. This aligns with the subtype's focus on
        minimizing total cost under capacity constraints.",
    "belongs_to_subtypes": true
}

{
    "matching_subtype": "subtype not find",
    "reasoning": "The given problem involves minimizing waste production
         while meeting medicine delivery constraints to the heart and
        brain. It does not fit any of the provided subtypes, which focus
        on budget allocation, parallel project resource allocation,
        revenue-maximizing load management, or supplier selection and
        equipment allocation. The problem is a standard Mixed Integer
        Linear Programming (MILP) problem with constraints on medicine
        delivery and waste minimization, but it does not align with the
        specific subtypes listed.",
    "belongs_to_subtypes": false
}
```

--------------------------------------------------------------------------

**Get the Modeling Thoughts:**

```
{
'Problem Type': 'Cost-Minimizing Resource Allocation with Capacity
    Constraints',
'Statement Thoughts':
'This subtype of the Resource Allocation Problem focuses on minimizing
    the total cost while allocating resources (sensor deployments and
    data analysis teams) to meet specific requirements (data collection
    and processing) under capacity and budget constraints. The goal is
    to optimally assign sensors to plots and allocate data processing
    tasks to teams such that all data collection requirements are met,
    and the total cost is minimized.,
Data Balance Constraint:The data collection requirements for each plot
    must be matched by the data processing capacity of the assigned team,
     ensuring that all collected data is processed without exceeding
    team capacities.,
Budget Constraint:The total cost of deploying sensors and utilizing data
     analysis teams must not exceed the available budget, ensuring cost-
    effective resource allocation.',
'Modeling Thoughts': [
    '[Define Decision Variables] Define integer variables for the number
        of times each team is used (e.g., \\( x_A \\), \\( x_B \\)).',
        '[Define Objective Function] Minimize the total cost, which
        includes sensor deployment costs and data processing costs.',
    '[Define Data Balance Constraint] Ensure the total data processed
        equals the sum of data collection requirements.',
```

        ’[Define Budget Constraint] Ensure the total cost does not exceed
            the available budget.’,
        ’[Comprehensive Verification] Check for common errors such as
            incorrect variable bounds, constraint formulation, and objective
            function alignment.’,
        ’[Write Gurobi Code] Implement the model in Gurobi to solve the
            problem.’],
        ’[Gurobi Code]:‘‘‘python\nimport json\nimport numpy as np\nimport
            math\nimport gurobipy as gp\nfrom gurobipy import GRB\n\n# Create
             a new model\nmodel = gp.Model(\’model\’)\n\n# define parameters\
            n\n# define variables\n\n# define constraints\n\n# define
            objective \n\n# Optimize the model\nmodel.optimize()\nstatus =
            model.status\n\nobj_val = None\n# Check whether the model is
            infeasible, has infinite solutions, or has an optimal solution\
            nif status == gp.GRB.INFEASIBLE:\n obj_val = "infeasible"\nelif
            status == gp.GRB.UNBOUNDED:\n obj_val = "unbounded"\nelif status
            == gp.GRB.OPTIMAL:\n obj_val = model.objVal\ntime = model.
            TimeLimit\nprint("Timecost":,time)\nprint("Objective Value:",
            obj_val)\n‘‘‘’,
        ’[Common Errors to Avoid]:\n1. **Incorrect Variable Types**: Ensure
            variables are defined as integers if fractional allocations are
            not allowed.\n2. **Constraint Formulation**: Verify that
            constraints correctly represent the problem requirements (e.g.,
            data balance must be exact).\n3. **Objective Function Alignment
            **: Ensure the objective function includes all relevant costs (e.
            g., both sensor deployment and data processing costs).\n4. **
            Budget Constraint Omission**: Do not forget to include the budget
             constraint if applicable.\n5. **Indexing Errors**: Ensure all
            indices and summations are correctly implemented in the code.’
]

}

------------------------------------------------------------------------------

**Solver Code:**

```python
import gurobipy as gp
from gurobipy import GRB

# Create a new model
model = gp.Model('MedicineDelivery')

# Define decision variables: minutes each machine is used
x1 = model.addVar(lb=0, vtype=GRB.CONTINUOUS, name="Machine1_minutes")
x2 = model.addVar(lb=0, vtype=GRB.CONTINUOUS, name="Machine2_minutes")

# Set objective: minimize total waste
model.setObjective(0.3 * x1 + 0.5 * x2, GRB.MINIMIZE)

# Add constraints
# Heart constraint: at most 8 units
model.addConstr(0.5 * x1 + 0.3 * x2 <= 8, "Heart_limit")
# Brain constraint: at least 4 units
model.addConstr(0.8 * x1 + 1.0 * x2 >= 4, "Brain_minimum")

# Optimize the model
```

```
model.optimize()

# Check solution status
status = model.status
if status == GRB.OPTIMAL:
    print(f"Optimal solution found:")
    print(f"Machine 1 minutes: {x1.x}")
    print(f"Machine 2 minutes: {x2.x}")
    print(f"Total waste produced: {model.objVal}")
elif status == GRB.INFEASIBLE:
    print("Model is infeasible")
elif status == GRB.UNBOUNDED:
    print("Model is unbounded")
```

## K.2 MAMO EasyLP

### MAMO EasyLP

**Natural Language Description:**
An environmental organization is planning to invest in two projects: Project X which involves tree planting, and Project Y which focuses on waste management. The investment in each project must be a whole number due to the indivisible nature of resources. The total combined investment for both projects cannot exceed 20 units due to budget limitations. Additionally, the overall environmental impact score, calculated as twice the investment in Project X plus the investment in Project Y, should be at least 10 points to meet their objectives.Each unit of investment in Projects X and Y has an associated cost of 4 and 5 units respectively, and the organization aims to minimize this total cost while meeting all constraints.Given these conditions, what is the minimum total cost required for this scenario? Please provide your answer rounded to the nearest whole number.

-------------------------------------------------------------------

**Subproblem Identification Process:**

```
{
    "matching_problem_type": "Capital Budgeting Problem",
    "reasoning": "The problem involves selecting investments in projects
        (Project X and Project Y) with integer investment amounts,
        budget constraints, and an objective to minimize total cost while
        meeting environmental impact requirements. This aligns with the
        Capital Budgeting Problem, which deals with allocating limited
        resources among competing projects to maximize or minimize a
        certain objective under constraints.",
    "belongs_to_problem_types": true
}

{
    "matching_subtype": "subtype not find",
    "reasoning": "The input problem involves selecting investments in
        two projects with integer constraints, budget limitations, and an
        objective to minimize total cost while meeting an environmental
        impact score. The provided subtypes focus on facility location
        with service balance and capacity constraints, and multi-
        objective optimization with weighted objectives, neither of which
        align with the given problem's characteristics.",
    "belongs_to_subtypes": false
```

```
}
```

----------------------------------------------------------------

**Get the Modeling Thoughts:**

```
{
'Problem Type': 'Capital Budgeting Problem',
'Statement Thoughts':
'The Capital Budgeting Problem involves selecting a combination of
    investments or purchases to maximize or minimize an objective (e.g.,
     cost or profit) while adhering to budget constraints, operational
    requirements, and other limitations. In this case, the airline must
    choose between two aircraft models to minimize total costs while
    meeting range, quantity, and operating cost constraints. This
    problem is common in industries requiring strategic resource
    allocation, such as transportation, manufacturing, and
    infrastructure development.,
Range Requirement Constraint: The combined range of all purchased
    aircraft must meet or exceed 40,000 kilometers to fulfill
    operational needs.,
Aircraft Quantity Constraint: The total number of aircraft purchased
    must be at least 8 to ensure flight safety and operational
    redundancy.,
Operating Cost Constraint: The total hourly operating cost of all
    aircraft must not exceed $150,000 USD to maintain financial
    sustainability.'
,
'Modeling Thoughts': [
    '[Define Decision Variables] Define integer decision variables for
        the number of each type of aircraft to purchase.',
    '[Define Objective Function] Formulate the objective to minimize
        total cost, including purchase and operating costs.',
    '[Define Range Constraint] Ensure the combined range of all aircraft
         meets or exceeds the required range.',
    '[Define Quantity Constraint] Ensure the total number of aircraft
        purchased meets the minimum requirement.',
    [Define Operating Cost Constraint] Ensure the total hourly operating
         cost does not exceed the specified limit.',
    '[Comprehensive Verification] Check for common errors such as
        incorrect cost calculations or constraint formulations.',
    '[Write Gurobi Code] Implement the model in Gurobi to solve the
        problem.'],

    '[ Gurobi Code], '''python', 'import json', 'import numpy as np', '
        import math', 'import gurobipy as gp', 'from gurobipy import GRB
        ', '# Create a new model', "model = gp.Model('model')", '# define
         parameters', '# define variables', '# define constraints', '#
        define objective ', '# Optimize the model', 'model.optimize()', '
        status = model.status', 'obj_val = None', '# Check whether the
        model is infeasible, has infinite solutions, or has an optimal
        solution', 'if status == gp.GRB.INFEASIBLE:', ' obj_val = "
        infeasible"', 'elif status == gp.GRB.UNBOUNDED:', ' obj_val = "
        unbounded"', 'elif status == gp.GRB.OPTIMAL:', ' obj_val = model.
        objVal', 'time = model.TimeLimit', 'print("Timecost":,time)', '
        print("Objective Value:", obj_val)', '''',
```

```
    '[Common Errors to Avoid]', '1. **Incorrect Cost Calculation**:
        Ensure the operating cost is correctly included in the objective
        function.', '2. **Constraint Formulation**: Verify that
        constraints are correctly formulated (e.g., >= vs <=).', '3. **
        Integer Variables**: Ensure decision variables are defined as
        integers.', '4. **Units Consistency**: Ensure all costs and
        ranges are in consistent units.', '5. **Solver Status Check**:
        Always check the solver status to confirm an optimal solution was
         found.']
}

- - - - - - - - - - - - - - - - - - - - - - - - - - - - - - - - - - - - - - - - - - -
```

**Solver Code:**

```
import gurobipy as gp
from gurobipy import GRB

# Create a new model
model = gp.Model('Environmental_Projects')

# Define decision variables (integer variables for investment units)
x = model.addVar(vtype=GRB.INTEGER, name="Project_X_Investment")
y = model.addVar(vtype=GRB.INTEGER, name="Project_Y_Investment")

# Set objective function: minimize total cost (4*X + 5*Y)
model.setObjective(4*x + 5*y, GRB.MINIMIZE)

# Add constraints
# Budget constraint: X + Y <= 20
model.addConstr(x + y <= 20, "Budget_Constraint")

# Environmental impact constraint: 2X + Y >= 10
model.addConstr(2*x + y >= 10, "Environmental_Impact_Constraint")

# Non-negativity constraints
model.addConstr(x >= 0, "Non_Negativity_X")
model.addConstr(y >= 0, "Non_Negativity_Y")

# Optimize the model
model.optimize()

# Check solution status
if model.status == GRB.OPTIMAL:
    print("Minimum Total Cost:", round(model.objVal))
else:
    print("No optimal solution found")
```

**MAMO EasyLP**

**Natural Language Description:**
A tourism company is planning to allocate resources between two major services: $X$ represents local sightseeing tours and $Y$ represents adventure activities. The total number of services that can be offered in a month cannot exceed 200 due to resource constraints. The minimum number of local sightseeing tours ($X$) they need to offer is at least 50, and for adventure activities ($Y$), it

cannot be less than 30. Each service has associated operational costs, with each local sightseeing tour costing $300 and each adventure activity costing $500. The company aims to minimize the total cost while meeting all the constraints(X, Y are integers). What is the minimum possible monthly operational cost (in dollars) for the company? Round your answer to the nearest dollar.

--------------------------------------------------------------------------

**Subproblem Identification Process:**

```
{
    "matching_problem_type": "Resource Allocation Problem",
    "reasoning": "The problem involves allocating limited resources (
        total services <= 200) between two services (X and Y) with
        specific constraints (minimum requirements for X and Y) and an
        objective to minimize costs. This fits the definition of a
        Resource Allocation Problem, where the goal is to optimally
        distribute resources under given constraints.",
    "belongs_to_problem_types": true
}

{
    "matching_subtype": "Cost-Minimizing Resource Allocation with
        Capacity Constraints",
    "reasoning": "The problem involves allocating resources (local
        sightseeing tours and adventure activities) to minimize total
        operational costs while adhering to capacity constraints (total
        services <= 200, minimum services for X >= 50 and Y >= 30). This
        matches the subtype 'Cost-Minimizing Resource Allocation with
        Capacity Constraints', which focuses on minimizing costs under
        capacity and other constraints.",
    "belongs_to_subtypes": true
}

{
    "matching_subtype": "Integer Linear Programming for Budget
        Allocation",
    "reasoning": "The problem involves allocating resources (local
        sightseeing tours and adventure activities) with integer
        constraints to minimize total operational costs while satisfying
        various linear constraints (total services, minimum requirements)
        . This matches the description of 'Integer Linear Programming for
         Budget Allocation' where the goal is to minimize total cost
        under integer constraints and linear constraints.",
    "belongs_to_subtypes": true
}

{
    "matching_subtype": "Integer Linear Programming for Resource
        Allocation with Multiple Constraints",
    "reasoning": "The input problem involves allocating resources (local
         sightseeing tours and adventure activities) to minimize total
        cost while meeting specific constraints (minimum number of
        services, total service limit). The decision variables (X and Y)
        must be integers, similar to the subtype which involves integer
        allocations to meet constraints and minimize costs.",
    "belongs_to_subtypes": true
}
```

```
{
    "matching_subtype": "Integer Linear Programming for Resource
        Allocation with Proportional and Range Constraints",
    "reasoning": "The input problem involves allocating resources
        between two services (local sightseeing tours and adventure
        activities) with integer constraints, aiming to minimize the
        total cost while satisfying combined resource limits and range
        constraints (minimum number of services for each type). This
        matches the description of the subtype 'Integer Linear
        Programming for Resource Allocation with Proportional and Range
        Constraints', which involves allocating resources between
        two services with integer constraints and similar types of
        constraints.",
    "belongs_to_subtypes": true
}

{
    "matching_subtype": "Integer Linear Programming for Service
        Allocation with Demand and Staffing Constraints",
    "reasoning": "The input problem involves allocating resources (
        services) between two types (local sightseeing tours and
        adventure activities) with integer constraints. The goal is to
        minimize total cost while satisfying minimum demand requirements
        (at least 50 local tours and 30 adventure activities) and a
        resource limitation (total services cannot exceed 200). This
        matches the subtype 'Integer Linear Programming for Service
        Allocation with Demand and Staffing Constraints', which also
        involves allocating services with integer constraints, minimizing
         cost, and satisfying demand and resource constraints.",
    "belongs_to_subtypes": true
}

{
    "matching_subtype": "Integer Linear Programming for Staff Allocation
         with Service Capacity and Cost Constraints",
    "reasoning": "The input problem involves allocating integer
        resources (services) between two types (local sightseeing tours
        and adventure activities) with the goal of minimizing total
        operational costs while meeting minimum service requirements and
        adhering to a total service limit. This closely matches the
        subtype description, which involves allocating integer staff
        members between two service types to minimize costs while meeting
         demand and staffing constraints. Both problems involve integer
        decision variables, cost minimization, and similar constraint
        structures.",
    "belongs_to_subtypes": true
}
```

------------------------------------------------------------------------

**Get the Modeling Thoughts:**

```
{
"Problem Type": "Integer Linear Programming for Staff Allocation with
    Service Capacity and Cost Constraints",
"Statement Thoughts":
```

"This problem involves allocating integer numbers of staff members
    between two types of after-sales services (phone support and on-site
     service) with the goal of minimizing total employment costs while
    meeting daily service demand requirements and adhering to a total
    staffing limit. The decision variables represent the number of staff
     allocated to each service type, and all allocations must be
    integers.,
Service Demand Constraint: Each phone support staff member can handle 50
     phone consultations and 20 remote technical support sessions per
    day, while each on-site service staff member can provide 10 on-site
    repairs and 5 equipment replacements per day. The total service
    capacity must meet or exceed the daily demand for each service type.,

Staffing Constraint: The total number of phone support and on-site
    service staff members cannot exceed 100."
,
"Modeling Thoughts": [
    "[Define Decision Variables] Define integer decision variables for
        the number of phone support staff members (x) and on-site service
         staff members (y).",
    "[Define Objective Function] Minimize the total employment cost: 200
        x + 300y.",
    "[Define Service Demand Constraints] Ensure the total service
        capacity meets or exceeds the daily demand for each service type
        .",
    "[Define Staffing Constraint] Ensure the total number of staff
        members does not exceed 100.",
    "[Comprehensive Verification] Check for common errors such as
        incorrect constraint formulation, non-integer solutions, and
        infeasible demand requirements.",
    "[Write Gurobi Code] Write the Gurobi code to solve the problem."
],
    '[ Gurobi Code], '''`python', 'import json', 'import numpy as np', '
        import math', 'import gurobipy as gp', 'from gurobipy import GRB
        ', '# Create a new model', "model = gp.Model('model')", '#
        define parameters', '# define variables', '# define constraints
        ', '# define objective ', '# Optimize the model', 'model.
        optimize()', 'status = model.status', 'obj_val = None', '# Check
         whether the model is infeasible, has infinite solutions, or has
         an optimal solution', 'if status == gp.GRB.INFEASIBLE:', '
        obj_val = "infeasible"', 'elif status == gp.GRB.UNBOUNDED:', '
        obj_val = "unbounded"', 'elif status == gp.GRB.OPTIMAL:', '
        obj_val = model.objVal', 'time = model.TimeLimit', 'print("
        Timecost":,time)', 'print("Objective Value:", obj_val)', '''`',
    "[Common Errors to Avoid:]",
        "1. **Incorrect Demand Constraints**: Ensure the service capacity
             constraints are correctly formulated .",
        "2. **Non-Integer Solutions**: Ensure variables are defined as
            integers (vtype=GRB.INTEGER).",
        "3. **Infeasible Demand**: Check if the demand values are too
            high to be met with the staffing limit.",
        "4. **Objective Function**: Ensure the cost coefficients are
            correctly applied .",
        "5. **Indexing Mistakes**: Ensure all constraints use the correct
             variables."
    ]

}

```
----------------------------------------------------------------------
```
**Solver Code:**

```python
import gurobipy as gp
from gurobipy import GRB

# Create a new model
model = gp.Model('TourismResourceAllocation')

# Define variables
X = model.addVar(vtype=GRB.INTEGER, name="X") # Local sightseeing tours
Y = model.addVar(vtype=GRB.INTEGER, name="Y") # Adventure activities

# Define constraints
model.addConstr(X + Y <= 200, name="TotalServicesLimit")
model.addConstr(X >= 50, name="MinLocalTours")
model.addConstr(Y >= 30, name="MinAdventureActivities")

# Define objective: minimize total cost (300X + 500Y)
model.setObjective(300 * X + 500 * Y, sense=GRB.MINIMIZE)

# Optimize the model
model.optimize()

# Check solution status and output results
if model.status == GRB.OPTIMAL:
    print("Minimum monthly operational cost: $", round(model.objVal))
else:
    print("No optimal solution found")
```

## K.3   MAMO ComplexLP

**MAMO ComplexLP**

**Natural Language Description:**
Natural Languange description: Imagine you're a college student aiming to balance your diet and budget. You have identified nine different food items from your local grocery store that you can include in your menu: Chicken, Rice, Apples, Steak, Lentils, Fish, Tofu, Cheese, and Bread. Each of these foods provides varying amounts of protein, carbohydrates, and calories, and each comes with its own price. Here is the detailed nutritional content and cost for each food item: - Chicken: Gives you 15 grams of protein, 18 grams of carbohydrates, and 300 calories for $4. - Rice: Offers 1 gram of protein, 25 grams of carbohydrates, and 267 calories for $2. - Apples: Provide 1 gram of protein, 21 grams of carbohydrates, and 266 calories for $5. - Steak: Contains 6 grams of protein, 3 grams of carbohydrates, and 119 calories for a higher cost of $10. - Lentils: These give 3 grams of protein, 7 grams of carbohydrates, and 166 calories for just$2. - Fish: Delivers 17 grams of protein, 13 grams of carbohydrates, and 129 calories for $8. - Tofu: Offers a substantial 18 grams of protein, 27 grams of carbohydrates, and 216 calories for $10. - Cheese: Gives 12 grams of protein, 17 grams of carbohydrates, and 76 calories for $9. - Bread: Provides 2 grams of protein, a massive 30 grams of carbohydrates, and 258 calories for $4. Your daily dietary goal is to consume at least 90 grams of protein, 105 grams of carbohydrates, and 1805 calories. Your challenge is to figure out how to meet these nutritional requirements from the food options mentioned above while spending the least amount of money. So, what is the least amount of money you need to spend to meet your daily dietary requirements? Please note that the response should be a single answer, asking for only the optimal value.

```
----------------------------------------------------------------------
```

**Subproblem Identification Process:**

```
{
    "matching_problem_type": "Diet Problem",
    "reasoning": "The given problem involves selecting a mix of food
        items that meet specific nutritional requirements while
        minimizing cost, which is characteristic of the Diet Problem, a
        standard form of linear programming introduced by George Stigler.
        The aim is to decide quantities of food to meet nutritional
        constraints within constraints, similar to the Diet Problem
        listed.",
    "belongs_to_problem_types": true
}

{
    "matching_subtype": "Diet Problem with Integer Constraint",
    "reasoning": "The given problem involves determining the quantity of
        each food item to meet nutritional requirements while minimizing
        cost. The quantities must be integers, representing whole
        servings of each food item. This aligns with the 'Diet Problem
        with Integer Constraint' subtype, which includes nutritional
        constraints, cost minimization, and integer servings.",
    "belongs_to_subtypes": true
}
```

------------------------------------------------------------------------

**Get the Modeling Thoughts:**

```
{
"Problem Type": "Diet Problem with Integer Constraint",
"Statement Thoughts":
 "The Diet Problem with Integer Constraint involves determining the
    quantity of each food item that should be consumed to meet
    nutritional requirements while minimizing cost. The twist is that
    the quantities must be integers, representing the whole number of
    servings or portions of each food item. The problem is formulated
    to ensure nutritional constraints (such as calories, protein,
    vitamins, etc.) are satisfied, and the total cost is minimized.,
Nutritional Constraints: Ensure that the selected food items provide at
    least the required amount of nutrients, such as calories, proteins,
    fats, vitamins, and minerals.,
Cost Minimization: The total cost of the selected food items should be
    minimized while meeting nutritional constraints.,
Integer Servings: The servings of food items must be integer values,
    representing realistic portions of each food item.",
"Modeling Thoughts": [
    "[Define Decision Variables] Define decision variables that
        represent the integer quantities of servings for each food item
        .",
    "[Define Objective Function] Minimize the total cost of the selected
         food items.",
    "[Define Nutritional Constraints] Ensure that the nutrient intake
        meets or exceeds the minimum required levels for each nutrient.",
    "[Implement Integer Constraints] Ensure integer constraints for the
        serving sizes of each food item in the model.",
```

```
        "[Comprehensive Verification] Check the common errors in the
            optimization model.",
        "[Write Gurobi Code] Write the Gurobi code to solve the problem.",

        "[Gurobi Code]:\n‘‘‘python\nimport json\nimport numpy as np\nimport
            math\nimport gurobipy as gp\nfrom gurobipy import GRB\n\n# Create
             a new model\nmodel = gp.Model(’model’)\n\n# define parameters\n\
            n# define variables\n\n# define constraints\n\n# define objective
             \n\n# Optimize the model\nmodel.optimize()\nstatus = model.
            status\n\nobj_val = None\n# Check whether the model is infeasible
            , has infinite solutions, or has an optimal solution\nif status
            == gp.GRB.INFEASIBLE:\n obj_val = \"infeasible\"\nelif status ==
            gp.GRB.UNBOUNDED:\n obj_val = \"unbounded\"\nelif status == gp.
            GRB.OPTIMAL:\n obj_val = model.objVal\ntime = model.TimeLimit\
            nprint(\"Timecost\":,time)\nprint(\"Objective Value:\", obj_val)\
            n‘‘‘"
    ,
    "[error_tips]":
        "Ensure decision variables are defined as integers.",
        "Confirm that all nutritional requirements are modeled correctly
            and constraints are set accurately.",
        "Review that cost coefficients are correctly assigned in the
            objective function.",
        "gurobi_example": "Example not provided since specific problem
            data is omitted; follow reasoning flow for formulation
            specifics."
]
}
```

--------------------------------------------------------------------------

**Solver Code:**

```
import gurobipy as gp
from gurobipy import GRB

# Create a new model
model = gp.Model(’DietProblem’)

# Define the food items and their nutritional content and cost
foods = [’Chicken’, ’Rice’, ’Apples’, ’Steak’, ’Lentils’, ’Fish’, ’Tofu
    ’, ’Cheese’, ’Bread’]
protein = [15, 1, 1, 6, 3, 17, 18, 12, 2]
carbs = [18, 25, 21, 3, 7, 13, 27, 17, 30]
calories = [300, 267, 266, 119, 166, 129, 216, 76, 258]
cost = [4, 2, 5, 10, 2, 8, 10, 9, 4]

# Define the nutritional requirements
required_protein = 90
required_carbs = 105
required_calories = 1805

# Define decision variables for the servings of each food item
x = model.addVars(foods, vtype=GRB.INTEGER, name="x")

# Define the objective function: Minimize the total cost
```

```
model.setObjective(gp.quicksum(cost[i] * x[foods[i]] for i in range(len(
    foods))), GRB.MINIMIZE)

# Define nutritional constraints
model.addConstr(gp.quicksum(protein[i] * x[foods[i]] for i in range(len(
    foods))) >= required_protein, "ProteinConstraint")
model.addConstr(gp.quicksum(carbs[i] * x[foods[i]] for i in range(len(
    foods))) >= required_carbs, "CarbsConstraint")
model.addConstr(gp.quicksum(calories[i] * x[foods[i]] for i in range(len
    (foods))) >= required_calories, "CaloriesConstraint")

# Optimize the model
model.optimize()

# Extract the optimal value
if model.status == GRB.OPTIMAL:
    print("Objective Value:", model.objVal)
else:
    print("No optimal solution found.")
```

## MAMO ComplexLP

**Natural Language Description:**
Consider a courier company that needs to deliver packages to five distinct cities, denoted as E, F, G, H, and I. The courier can start from any city, but they must visit each city only once and then return to the starting point. The aim is to find a route that would minimize the total delivery cost. The cost might include factors like distance, fuel expenses, or traffic conditions. Here's an outline of the delivery cost between these cities: The cost to deliver from City E to F is 50 units, to G is 48 units, to H is 99 units, and to I is 91 units. From City F, it costs 50 units to deliver to E, 57 units to deliver to G, 84 units to H, and 72 units to I. For City G, the delivery costs are 48 units to E, 57 units to F, 46 units to H, and 86 units to I. If the package starts from City H, it costs 99 units to deliver to E, 84 units to F, 46 units to G, and 29 units to I. Lastly, from City I, it costs 91 units to deliver to E, 72 units to F, 86 units to G, and 29 units to H. What is the least total delivery cost for the courier to visit each city exactly once and then return to the starting point?

--------------------------------------------------------------------------

**Subproblem Identification Process:**

```
{
    "matching_problem_type": "TSP Problem",
    "reasoning": "The specific problem described involves finding the
        least cost route for a courier to visit each city once before
        returning to the starting point, which aligns with the Travelling
         Salesman Problem (TSP). The objective of minimizing the total
        delivery cost fits the criteria for a TSP, one of the defined
        problem types.",
    "belongs_to_problem_types": true
}
```

--------------------------------------------------------------------------

**Get the Modeling Thoughts:**

```
{
```

```
"Problem Type": "TSP Problem",
"Statement Thoughts":
"The Traveling Salesman Problem (TSP) involves finding the shortest
    possible route that visits each city exactly once and returns to the
    original city. The problem is represented as a matrix of distances
    between cities. It is an NP-hard problem in combinatorial
    optimization, important for logistics, planning, and routing.,
Tour Constraints: Each city must be visited exactly once, forming a
    single continuous tour.",
Subtour Elimination: Constraints are necessary to prevent the formation
    of disconnected subtours (mini-tours not connected to the main tour)
    .",
"Modeling Thoughts": [
    "[Define Decision Variables] Define binary decision variables for
        each edge \\( x_{ij} \\) indicating if the edge is used in the
        tour.",
    "[Define Objective Function] Minimize the sum of distances for
        chosen edges.",
    "[Define Degree Constraints] Ensure each city is entered and exited
        exactly once.",
    "[Define Subtour Elimination Constraints] Implement constraints,
        such as MTZ constraints, to enforce single tour formation.",
    "[Comprehensive Verification] Check the model for common errors such
        as incorrect indexing or missing constraints.",
    "[Write Gurobi Code] Implement the model using Gurobi to solve the
        TSP."

    '[ Gurobi Code], '''python', 'import json', 'import numpy as np', '
        import math', 'import gurobipy as gp', 'from gurobipy import GRB
        ', '# Create a new model', "model = gp.Model('model')", '#
        define parameters', '# define variables', '# define constraints
        ', '# define objective ', '# Optimize the model', 'model.
        optimize()', 'status = model.status', 'obj_val = None', '# Check
         whether the model is infeasible, has infinite solutions, or has
         an optimal solution', 'if status == gp.GRB.INFEASIBLE:', '
        obj_val = "infeasible"', 'elif status == gp.GRB.UNBOUNDED:', '
        obj_val = "unbounded"', 'elif status == gp.GRB.OPTIMAL:', '
        obj_val = model.objVal', 'time = model.TimeLimit', 'print("
        Timecost:",time)', 'print("Objective Value:", obj_val)', '''',
    "[Common Errors to Avoid]: \n1. Incorrect Subtour Elimination:
        Ensure MTZ constraints exclude the starting city and are applied
        to correct indices. \n2. Indexing Mistakes: Use consistent 0-
        based or 1-based indexing for cities. \n3. Self-Loops: Explicitly
         disable \\( x_{ii} \\) variables. \n4. Bounds on MTZ Variables:
        Set \\( u_i \\) bounds correctly (\\( 1 \\leq u_i \\leq n-1 \\)).
         \n5. Objective Function: Ensure distances are correctly paired
        with \\( x_{ij} \\) and exclude \\( i = j \\)."
    ]
}
```

------------------------------------------------------------------

**Solver Code:**

```
import gurobipy as gp
from gurobipy import GRB
```

```
# Number of cities
n = 5

# Distance matrix
dist = [
    [0, 50, 48, 99, 91], # From City E
    [50, 0, 57, 84, 72], # From City F
    [48, 57, 0, 46, 86], # From City G
    [99, 84, 46, 0, 29], # From City H
    [91, 72, 86, 29, 0] # From City I
]

# Create model
m = gp.Model('TSP')

# Variables: x[i,j] = 1 if edge i->j is used
x = m.addVars(n, n, vtype=GRB.BINARY, name='x')
for i in range(n):
    x[i,i].UB = 0 # Disable self-loops

# Assignment constraints
for i in range(n):
    m.addConstr(gp.quicksum(x[i,j] for j in range(n) if j != i) == 1,
        name=f"out_{i}")
    m.addConstr(gp.quicksum(x[j,i] for j in range(n) if j != i) == 1,
        name=f"in_{i}")

# MTZ variables and constraints for subtour elimination
u = m.addVars(range(1, n), lb=1, ub=n-1, vtype=GRB.CONTINUOUS, name='u')

for i in range(1, n):
    for j in range(1, n):
        if i != j:
            m.addConstr(u[i] - u[j] + n * x[i,j] <= n - 1, name=f"mtz_{i}
                _{j}")

# Objective function: Minimize the total delivery cost
obj = gp.quicksum(x[i,j] * dist[i][j] for i in range(n) for j in range(n
    ) if i != j)
m.setObjective(obj, GRB.MINIMIZE)

# Solve the model
m.optimize()

# Extract and print the optimal tour and its cost
if m.status == GRB.OPTIMAL:
    print('Optimal tour:')
    current = 0
    tour = [current]
    visited = set([current])
    while len(visited) < n:
        for j in range(n):
            if j != current and x[current,j].X > 0.5:
                tour.append(j)
                current = j
                visited.add(current)
                break
    tour.append(0) # Return to start
```

```
    print(' -> '.join(map(str, tour)))
    print(f'Total delivery cost: {m.ObjVal}')
else:
    print('No solution found.')
```

## K.4 IndustryOR

**IndustryOR**

**Natural Language Description:**
The number of salespeople required at a 24-hour convenience store in different time periods is as follows: 2:00-6:00 - 10 people, 6:00-10:00 - 15 people, 10:00-14:00 - 25 people, 14:00-18:00 - 20 people, 18:00-22:00 - 18 people, 22:00-2:00 - 12 people. Salespeople start their shifts at 2:00, 6:00, 10:00, 14:00, 18:00, and 22:00, working continuously for 8 hours. Determine the minimum number of salespeople needed to meet the requirements.

--------------------------------------------------------------------------------

**Subproblem Identification Process:**

```
{
    "matching_problem_type": "Staff Rostering Problem",
    "reasoning": "The problem involves determining the minimum number of
        salespeople needed to meet varying demand across different time
        periods, with specific shift start times and durations. This
        aligns with the 'Staff Rostering Problem', which deals with
        scheduling staff to meet demand while minimizing costs or the
        number of staff.",
    "belongs_to_problem_types": true
}
```

--------------------------------------------------------------------------------

**Get the Modeling Thoughts:**

```
{
"Problem Type": "Staff Rostering Problem",
"Statement Thoughts": {
 "Assign shifts to employees over a scheduling period while satisfying
     labor regulations, employee availability, and operational coverage
     requirements. This problem is common in industries such as
     healthcare, retail, and hospitality, where shift-based workforces
     are essential. Key features include handling multiple shift types (
     e.g., morning, evening, night), accommodating employee preferences
     and time-off requests, ensuring adequate staffing levels, and
     complying with legal and contractual obligations. The goal is to
     create a feasible and equitable schedule that balances
     organizational needs with employee well-being.,
Labor Regulations Compliance: Shifts must comply with labor laws,
    including maximum consecutive working days, mandatory rest periods
    between shifts, and weekly working hour limits.,
Employee Availability:"Employees cannot be assigned to shifts during
    their declared unavailability (e.g., time-off, preferred days off)
    or overlapping with existing assignments.",
```

Operational Coverage Requirements: Each shift must meet a minimum
    staffing level, which may vary by shift type, day, or role (e.g.,
    peak hours require more employees).,
Shift Sequence Restrictions: Employees must have sufficient rest between
    consecutive shifts (e.g., a minimum gap of 12 hours after a night
    shift).,
Contractual Work Hours: Assignments must respect employment contracts,
    including part-time/full-time distinctions and maximum allowable
    hours per pay period.",
Fair Shift Distribution: Shifts must be distributed equitably among
    employees to avoid overburdening individuals, considering
    preferences, seniority, and historical assignments.",
Skill and Role Matching: Employees must possess required certifications
    or qualifications for specialized shifts (e.g., a licensed
    pharmacist for medication dispensing shifts).",
"Modeling Thoughts": [
    "[Define Decision Variables] Create binary variables for employee-
        shift-day assignments",
    "[Define Objective Function] Minimize understaffing/overstaffing
        penalties or maximize employee preference satisfaction",
    "[Define Coverage Constraints] Ensure minimum required staff per
        shift",
    "[Define Availability Constraints] Respect employee unavailability
        days",
    "[Define Legal Constraints] Enforce maximum consecutive shifts and
        minimum rest periods",
    "[Define Pattern Constraints] Prevent prohibited shift sequences",
    "[Comprehensive Verification] Validate constraint logic and variable
         indexing",
    "[Write Gurobi Code] Implement model using Gurobi API"
    "[ Gurobi Code], '''python', 'import json', 'import numpy as np', '
        import math', 'import gurobipy as gp', 'from gurobipy import GRB
        ', '# Create a new model', "model = gp.Model('model')", '# define
         parameters', '# define variables', '# define constraints', '#
        define objective ', '# Optimize the model', 'model.optimize()', '
        status = model.status', 'obj_val = None', '# Check whether the
        model is infeasible, has infinite solutions, or has an optimal
        solution', 'if status == gp.GRB.INFEASIBLE:', ' obj_val = "
        infeasible"', 'elif status == gp.GRB.UNBOUNDED:', ' obj_val = "
        unbounded"', 'elif status == gp.GRB.OPTIMAL:', ' obj_val = model.
        objVal', 'time = model.TimeLimit', 'print("Timecost":,time)', '
        print("Objective Value:", obj_val)', '''''",
    "[Common Errors to Avoid:]",
        "1. **Incorrect Indexing**: Mixing up employee/day/shift
            dimensions in variables",
        "2. **Missed Constraints**: Forgetting to enforce minimum rest
            periods between shifts",
        "3. **Overlooking Part-Time Rules**: Not accounting for maximum
            weekly hours",
        "4. **Inefficient Modeling**: Creating excessive variables for
            simple patterns",
        "5. **Data Mismatches**: Alignment between availability data and
            variable indices",
        "6. **Objective Imbalance**: Not properly scaling understaffing/
            overstaffing penalties"
    ]
}

--------------------------------------------------------------------

**Solver Code:**

```
import gurobipy as gp
from gurobipy import GRB

# Create a new model
model = gp.Model('ShiftScheduling')

# Time periods and their requirements
time_periods = ['2:00-6:00', '6:00-10:00', '10:00-14:00', '14:00-18:00',
     '18:00-22:00', '22:00-2:00']
requirements = [10, 15, 25, 20, 18, 12]
shift_starts = [0, 1, 2, 3, 4, 5] # 0:2:00, 1:6:00, etc.

# Decision variables: number of people starting at each shift
x = model.addVars(6, vtype=GRB.INTEGER, name="x")

# Objective: minimize total number of employees
model.setObjective(gp.quicksum(x[i] for i in range(6)), GRB.MINIMIZE)

# Coverage constraints
# Each time period must be covered by people who started in the current
    or previous 3 shifts
# (since each shift covers 4 time periods)
model.addConstr(x[0] + x[5] >= requirements[0], "Coverage_0")
model.addConstr(x[0] + x[1] >= requirements[1], "Coverage_1")
model.addConstr(x[1] + x[2] >= requirements[2], "Coverage_2")
model.addConstr(x[2] + x[3] >= requirements[3], "Coverage_3")
model.addConstr(x[3] + x[4] >= requirements[4], "Coverage_4")
model.addConstr(x[4] + x[5] >= requirements[5], "Coverage_5")

# Optimize the model
model.optimize()

# Output the solution
if model.status == GRB.OPTIMAL:
    print(f"Minimum number of salespeople needed: {int(model.objVal)}")
    print("Shift assignments:")
    for i in range(6):
        print(f"Start at {time_periods[i].split('-')[0]}: {int(x[i].x)}
            people")
else:
    print("No optimal solution found")
```

**IndustryOR**

**Natural Language Description:**
A wealthy noble passed away, leaving the following inheritance:
- A painting by Caillebotte: $25000
- A bust of Diocletian: $5000
- A Yuan dynasty Chinese vase: $20000
- A 911 Porsche: $40000
- Three diamonds: each $12000
- A Louis XV sofa: $3000
- Two very precious Jack Russell racing dogs: each $3000 (will stipulates they must not be

separated)
- A sculpture from 200 AD: $10000
- A sailing boat: $15000
- A Harley Davidson motorcycle: $10000
- A piece of furniture once belonging to Cavour: $13000
which must be shared between two sons.  How to formulate a mathematical program and solve it using COPTPY to minimize the difference in value between the two parts?

------------------------------------------------------------------------

**Subproblem Identification Process:**

```
{
    "matching_problem_type": "Knapsack Problem",
    "reasoning": "The problem involves dividing a set of items (
        inheritance) into two parts (sons) with the objective of
        minimizing the difference in value between the two parts. This is
         a classic example of the Knapsack Problem, specifically a
        partition problem which is a variant of the Knapsack Problem
        where the goal is to divide items into two subsets with equal or
        as equal as possible sums.",
    "belongs_to_problem_types": true
}

{
    "matching_subtype": "subtype not find",
    "reasoning": "The input problem is about dividing inheritance items
        between two sons to minimize the difference in value, which is a
        form of the Partition Problem, a special case of the Knapsack
        Problem. The provided subtype 'Integer Linear Programming (ILP)
        with Knapsack Constraints' is about selecting optimal numbers of
        AI representatives to meet demand constraints, which is not
        relevant to the inheritance division problem.",
    "belongs_to_subtypes": false
}
```

------------------------------------------------------------------------

**Get the Modeling Thoughts:**

```
{
"Problem Type": "Knapsack Problem",
"Statement Thoughts":
"The city's planning department aims to maximize the total capacity
    increase of 4 major traffic intersections (A, B, C, D) by selecting
    expansion projects under a given budget. Each intersection has
    specific expansion options with associated costs and capacity
    increases. The problem is modeled as a 0-1 Knapsack Problem, where
    each expansion is treated as an item with a weight (cost) and value
    (capacity increase), and the goal is to select a subset of items to
    maximize the total value without exceeding the budget.,
Budget Limitation: The total cost of all selected expansions must not
    exceed the allocated budget.",
Binary Selection: Each expansion project for an intersection is either
    fully selected or not selected; partial or fractional expansions are
     not allowed,
```

Non-negative Costs and Capacities: The cost and capacity increase
    associated with each expansion must be non-negative values.,
Mutual Exclusivity per Intersection: At most one expansion project can
    be selected per intersection (if multiple options exist for a single
    intersection).",
"Modeling Thoughts": [
    "[Define Decision Variables] Integer variables for the number of
        expansions for each intersection.",
    "[Define Objective Function] Maximize the total capacity increase:
        200x_A + 300x_B + 400x_C + 500x_D.",
    "[Define Constraints] Budget constraint: 100x_A + 150x_B + 200x_C +
        250x_D \u2264 500. Total capacity requirement: 200x_A + 300x_B +
        400x_C + 500x_D \u2265 1000.",
    "[Comprehensive Verification] Check variable bounds, constraint
        coefficients, and objective direction.",
    "[Write Gurobi Code] Implement model using Gurobi's Python API."
    "### Gurobi Code:\n'''python\nimport json\nimport numpy as np\
        nimport math\nimport gurobipy as gp\nfrom gurobipy import GRB\n\n
        # Create a new model\nmodel = gp.Model('model')\n\n# define
        parameters\n\n# define variables\n\n# define constraints\n\n#
        define objective \n\n# Optimize the model\nmodel.optimize()\
        nstatus = model.status\n\nobj_val = None\n# Check whether the
        model is infeasible, has infinite solutions, or has an optimal
        solution\nif status == gp.GRB.INFEASIBLE:\n obj_val = \"
        infeasible\"\nelif status == gp.GRB.UNBOUNDED:\n obj_val = \"
        unbounded\"\nelif status == gp.GRB.OPTIMAL:\n obj_val = model.
        objVal\ntime = model.TimeLimit\nprint(\"Timecost\":,time)\nprint
        (\"Objective Value:\", obj_val)\n'''",
    "### Common Errors to Avoid:\n1. **Incorrect Variable Type**: Using
        CONTINUOUS instead of INTEGER for the number of expansions.\n2.
        **Missing Constraints**: Forgetting to add the budget constraint
        or the capacity requirement.\n3. **Objective Direction**:
        Minimizing instead of maximizing the total capacity increase.\n4.
         **Constraint Formulation**: Incorrectly formulating the budget
        or capacity constraints.\n5. **Non-negativity**: Forgetting to
        set lower bounds on variables."
    ]

}

--------------------------------------------------------------------------------
**Solver Code:**

```python
import gurobipy as gp
from gurobipy import GRB

# Create a new model
model = gp.Model('inheritance_division')

# Items and their values (must keep dogs together)
items = [
    ("Painting", 25000),
    ("Bust", 5000),
    ("Vase", 20000),
    ("Porsche", 40000),
    ("Diamond1", 12000),
```

```
        ("Diamond2", 12000),
        ("Diamond3", 12000),
        ("Sofa", 3000),
        ("Dogs", 6000), # Combined value since they must stay together
        ("Sculpture", 10000),
        ("Boat", 15000),
        ("Motorcycle", 10000),
        ("Furniture", 13000)
]

# Binary variables for each item (1 if assigned to son1, 0 if to son2)
x = model.addVars(len(items), vtype=GRB.BINARY, name="assign")

# Difference variable to minimize
diff = model.addVar(lb=0, vtype=GRB.CONTINUOUS, name="difference")

# Objective: minimize the difference
model.setObjective(diff, GRB.MINIMIZE)

# Calculate total values for each son
son1_value = gp.quicksum(x[i] * items[i][1] for i in range(len(items)))
son2_value = gp.quicksum((1 - x[i]) * items[i][1] for i in range(len(
    items)))

# Constraint: difference is >= absolute difference between son values
model.addConstr(diff >= son1_value - son2_value)
model.addConstr(diff >= son2_value - son1_value)

# Optimize the model
model.optimize()

# Check solution status
if model.status == GRB.OPTIMAL:
    print("Optimal solution found")
    print(f"Minimum difference: ${model.objVal:.2f}")

    print("\nSon 1 receives:")
    son1_total = 0
    for i in range(len(items)):
        if x[i].X > 0.5:
            print(f"- {items[i][0]}: ${items[i][1]}")
            son1_total += items[i][1]

    print("\nSon 2 receives:")
    son2_total = 0
    for i in range(len(items)):
        if x[i].X < 0.5:
            print(f"- {items[i][0]}: ${items[i][1]}")
            son2_total += items[i][1]

    print(f"\nSon 1 total: ${son1_total}")
    print(f"Son 2 total: ${son2_total}")
elif model.status == GRB.INFEASIBLE:
    print("Model is infeasible")
elif model.status == GRB.UNBOUNDED:
    print("Model is unbounded")
```

## K.5 ComplexOR

### ComplexOR

**Natural Language Description:**
This problem involves optimizing the number of raw rolls cut from a stock roll to meet specific width orders using various cutting patterns. This is a cutting stock problem. Given a roll of width RollWidth and a set of NumWidths widths Widths to be cut. Each width $i$ has a certain number of orders Orders_i. There are NumPatterns patterns and each pattern $j$ has a certain number of rolls of each width $i$ NumRollsWidthPattern$\{i, j\}$. The problem aims to minimize the total number of raw rolls cut. It is constrained that for each width $i$, the total number of rolls cut meets the total orders. The decision to be made concerns how to decide the number of rolls cut using each pattern $j$. The number of widths to be cut is 5. The total number of patterns is 2. The set of widths to be cut from the roll is [2, 3, 5]. The width of the raw roll is 10. The number of orders for each width is [4, 2, 2]. The number of rolls of each width $i$ in each pattern $j$ is [[1, 2, 0], [0, 0, 1]]. Number of rolls cut for each width is non-negative. Total number of rolls cut for each width $i$ must meet or exceed Orders_i. Number of rolls cut using each pattern $j$ must be non-negative. Minimize the total number of raw rolls cut to meet the orders.

-------------------------------------------------------------

**Subproblem Identification Process:**

```
{
    "matching_problem_type": "Cutting Stock Problem",
    "reasoning": "The specific problem described involves optimizing the
        cutting of raw rolls to meet specific width orders using various
        cutting patterns. This is a classic example of a Cutting Stock
       Problem, which is explicitly listed in the provided problem types
        . The objective and constraints of minimizing raw material usage
        while satisfying order requirements align directly with the
        characteristics of the Cutting Stock Problem.",
    "belongs_to_problem_types": true
}
```

-------------------------------------------------------------

**Get the Modeling Thoughts:**

```
{
"Problem Type": "Cutting Stock Problem",
"Statement Thoughts":
 "The Cutting Stock Problem is a combinatorial optimization problem
    commonly encountered in manufacturing and logistics. It involves
    determining the optimal way to cut raw material rolls of fixed
    length into smaller pieces of specified lengths and quantities
    required by customers, with the objective of minimizing the total
    number of rolls used and thereby reducing waste. The challenge lies
     in efficiently allocating the cuts to satisfy all customer demands
     while adhering to the physical limitations of the raw material
    rolls,
Roll Capacity: Each raw material roll has a fixed length, and the total
    length of all pieces cut from a single roll must not exceed this
    length.,
Demand Satisfaction: The total number of each type of piece produced
    across all rolls must exactly match the customer's specified demand
    for that type.,
```

```
Integrality Constraint: The number of pieces of each type cut from a
    roll and the number of rolls used must be non-negative integers, as
    partial pieces or partial rolls are not feasible.",
"Modeling Thoughts": [
    "[Define Decision Variables] Define integer variables for the number
        of times each cutting pattern is used.",
    "[Define Objective Function] Minimize the total number of raw
        material rolls used.",
    "[Define Demand Constraints] Ensure each customer's demand is met by
        the sum of pieces from all patterns.",
    "[Define Pattern Validity] Ensure each cutting pattern does not
        exceed the raw material length.",
    "[Comprehensive Verification] Check for valid patterns, correct
        demand calculations, and integer constraints.",
    "[Write Gurobi Code] Implement the model using Gurobi with column
        generation or predefined patterns."
    "### Gurobi Code:\n```python\nimport json\nimport numpy as np\
        nimport math\nimport gurobipy as gp\nfrom gurobipy import GRB\n\n
        # Create a new model\nmodel = gp.Model('model')\n\n# define
        parameters\n\n# define variables\n\n# define constraints\n\n#
        define objective \n\n# Optimize the model\nmodel.optimize()\
        nstatus = model.status\n\nobj_val = None\n# Check whether the
        model is infeasible, has infinite solutions, or has an optimal
        solution\nif status == gp.GRB.INFEASIBLE:\n obj_val = \"
        infeasible\"\nelif status == gp.GRB.UNBOUNDED:\n obj_val = \"
        unbounded\"\nelif status == gp.GRB.OPTIMAL:\n obj_val = model.
        objVal\ntime = model.TimeLimit\nprint(\"Timecost\":,time)\nprint
        (\"Objective Value:\", obj_val)\n```",
    "### Common Errors to Avoid:\n1. **Invalid Patterns**: Ensure each
        pattern's total length does not exceed the raw material length.\
        n2. **Non-Integer Variables**: Use integer variables for the
        number of rolls.\n3. **Incorrect Demand Calculation**: Verify
        that each pattern's contribution to demand is correctly
        multiplied by the number of rolls.\n4. **Insufficient Patterns**:
         Predefined patterns may not cover optimal solutions; consider
        column generation for better results.\n5. **Indexing Mistakes**:
        Align pattern indices correctly with sizes in constraints."
    ]

}
```

----------------------------------------------------------------

**Solver Code:**

```
import gurobipy as gp
from gurobipy import GRB

# Define data
roll_width = 10
widths = [2, 3, 5]
orders = [4, 2, 2]
num_patterns = 2
patterns = [[1, 2, 0], [0, 0, 1]]
num_widths = len(widths)

# Create a new model
```

```
model = gp.Model('CuttingStock')

# Define variables
# Integer variable y_j representing the number of times pattern j is
    used
y = model.addVars(num_patterns, vtype=GRB.INTEGER, name="y")

# Set objective
# Minimize the total number of raw rolls used
model.setObjective(gp.quicksum(y[j] for j in range(num_patterns)), GRB.
    MINIMIZE)

# Define constraints
# Demand satisfaction constraint
for i in range(num_widths):
    model.addConstr(
        gp.quicksum(patterns[j][i] * y[j] for j in range(num_patterns))
            >= orders[i],
        name=f"demand_{i}"
    )

# Pattern validity is inherent in the input and problem formulation

# Optimize the model
model.optimize()

# Check the solution status and print the result
if model.status == GRB.OPTIMAL:
    print("Optimal Total Number Of Raw Rolls Cut:", model.objVal)
    for j in range(num_patterns):
        print(f"Number of times pattern {j} is used: {y[j].x}")
elif model.status == GRB.INFEASIBLE:
    print("Model is infeasible")
elif model.status == GRB.UNBOUNDED:
    print("Model is unbounded")
else:
    print("Optimization was stopped with status", model.status)
```

## ComplexOR

**Natural Language Description:**
Capacitated facility location problems focus on determining the optimal placement of a certain number of facilities to serve a set number of customers in a way that minimizes the total cost, considering fixed costs, capacities, customer demands, and transport costs. Capacitated facility location problems deal with locating NumberOfFacilities facilities to serve NumberOfCustomers customers, at minimum total cost. Considering potential facility locations and customer zones as fixed points in a network, each facility has a fixed FacilityFixedCost and a FacilityCapacity. Furthermore, there exists a CustomerDemand for each customer zone, and a FacilityToCustomerTransportCost representing the cost of transport between facilities and customer zones. The number of potential facilities that can be established is 10. The number of customer zones to be served is 20. The fixed cost associated with establishing a facility is [8517, 5068, 9433, 6127, 6033, 5966, 7762, 9406, 6602, 7040]. The cost of transporting goods from each facility to each customer zone is [[80, 94, 44, 51, 190, 44, 129, 178, 129, 91, 172, 119, 177, 150, 90, 51, 53, 97, 184, 87], [139, 33, 104, 135, 50, 176, 97, 121, 47, 29, 186, 163, 149, 108, 156, 169, 100, 160, 153, 85], [153, 36, 18, 170, 18, 181, 178, 68, 171, 106, 159, 110, 21, 106, 91, 29, 144, 140, 155, 116], [103, 59, 78, 125, 14, 11, 152, 95, 76, 173, 36, 148, 75, 132, 59, 153, 113, 74, 185, 71], [193, 186, 130, 145, 114, 150, 33, 154, 20, 75, 103, 30,

137, 131, 167, 32, 53, 150, 176, 166], [159, 130, 156, 65, 36, 59, 199, 124, 104, 72, 180, 73, 43, 152, 143, 90, 161, 65, 172, 141], [173, 121, 110, 127, 22, 159, 195, 137, 47, 10, 87, 11, 154, 66, 126, 60, 152, 54, 20, 25], [181, 34, 186, 152, 109, 195, 133, 198, 30, 65, 69, 19, 109, 143, 108, 196, 59, 133, 10, 123], [82, 113, 147, 21, 88, 24, 38, 16, 70, 122, 148, 192, 116, 108, 18, 20, 143, 18, 116, 142], [176, 170, 87, 91, 195, 183, 124, 89, 72, 97, 89, 23, 45, 196, 97, 27, 83, 81, 171, 148]]. The capacity of each facility, which limits the amount of goods the facility can handle or produce is [301, 291, 453, 472, 492, 375, 493, 297, 331, 246]. The demand of each customer zone that must be met by the facilities is [117, 86, 69, 53, 110, 74, 136, 140, 126, 79, 54, 86, 114, 76, 136, 73, 144, 51, 53, 120]. Each facility has a maximum capacity of FacilityCapacity. Total number of facilities to be located is NumberOfFacilities. Each customer zone is served by at least one facility and its demand must be met. Total output for each facility cannot exceed its capacity. The number of facilities and customers is fixed and non-negative. Minimize the total cost of locating facilities and transporting goods to customer zones.

------------------------------------------------------------------------

**Subproblem Identification Process:**

```
{
    "matching_problem_type": "Facility Location-Allocation Problem",
    "reasoning": "The specific problem described is a capacitated
        facility location problem, which involves determining the optimal
         placement of facilities to serve customers while minimizing
        costs. This aligns with the 'Facility Location-Allocation Problem
        ' from the provided list, as it involves decisions about where to
         locate facilities and how to allocate resources to meet customer
        demands.",
    "belongs_to_problem_types": true
}

{
    "matching_subtype": "Capacitated Facility Location Problem",
    "reasoning": "The input problem matches the description of the '
        Capacitated Facility Location Problem' as it involves minimizing
        the total costs including fixed costs and transportation costs
        while considering the capacities of facilities and customer
        demands. It meets the constraints of Demand Satisfaction,
        Facility Capacity Constraint, Binary Facility Opening, Non-
        Negative Allocation, and City Demand Allocation, which align
        closely with those described under the 'Capacitated Facility
        Location Problem'.",
    "belongs_to_subtypes": true
}

{
    "matching_subtype": "Capacitated Facility Location Problem with
        Distances",
    "reasoning": "The input problem involves determining the optimal
        placement of facilities to serve customer zones while minimizing
        total costs, including fixed facility costs and transportation
        costs between facilities and customer zones. This aligns with the
         description of the 'Capacitated Facility Location Problem with
        Distances' subtype, which focuses on minimizing transportation
        costs proportional to distances between facilities and regions,
        while respecting capacity constraints.",
    "belongs_to_subtypes": true
}
```

------------------------------------------------------------

**Get the Modeling Thoughts:**

{
"Problem Type": "Capacitated Facility Location Problem with Distances",
"Statement Thoughts":
"The Capacitated Facility Location Problem with Distances is a
    specialized version of the Facility Location-Allocation Problem
    where each facility has a limited capacity for service, and the
    allocation of customers to facilities involves transportation costs
    proportional to the distance between them. The goal is to determine
    the optimal set of facilities to open and the allocation of customer
     demands to these facilities in a way that minimizes the total cost,
     which includes the construction costs of facilities and the
    transportation costs associated with the distance, while respecting
    capacity constraints.,
Demand Satisfaction with Distance Costs: Each region's population demand
     must be fully met with the consideration of minimizing
    transportation costs, which are proportional to the distance between
     facilities and regions.",
Facility Capacity with Population Limits: The sum of the populations
    allocated to a facility must not exceed its maximum response
    capacity.,
Facility Activation: Facilities must either be open or closed. Only open
     facilities incur construction costs and can be used for serving
    regions.,
Exclusive Assignment: Each region must be fully served by one open
    facility to prevent splitting of demand across multiple facilities.",

"Moldeing Thoughts": [
    "[Define Decision Variables] Define binary decision variables to
        determine if a facility is opened and continuous variables to
        represent the population served by facilities.",
    "[Define Objective Function] Minimize total cost by summing
        construction costs and distance-based response time costs.",
    "[Define Demand Constraints] Each region's population demand must be
         fully met.",
    "[Define Capacity Constraints] Ensure each facility does not exceed
        its capacity, using binary variables to enforce activation.",
    "[Comprehensive Verification] Check common errors in the
        optimization model, such as incorrect indexing or capacity
        constraints.",
    "[Write Gurobi Code] Write the Gurobi code to solve the problem."
    "[Gurobi Code]:\n'''python\nimport json\nimport numpy as np\nimport
        math\nimport gurobipy as gp\nfrom gurobipy import GRB\n\n# Create
         a new model\nmodel = gp.Model('model')\n\n# define parameters\n\
        n# define variables\n\n# define constraints\n\n# define objective
        \n\n# Optimize the model\nmodel.optimize()\nstatus = model.status
        \n\nobj_val = None\n# Check whether the model is infeasible, has
        infinite solutions, or has an optimal solution\nif status == gp.
        GRB.INFEASIBLE:\n obj_val = 'infeasible'\nelif status == gp.GRB.
        UNBOUNDED:\n obj_val = 'unbounded'\nelif status == gp.GRB.OPTIMAL
        :\n obj_val = model.objVal\ntime = model.TimeLimit\nprint('
        Timecost:', time)\nprint('Objective Value:', obj_val)\n'''",
    "[ommon Errors to Avoid]:\n1. **Incorrect Constraints**: Ensure
        population demand constraints align with assigned populations.\n2
        . **Capacity Constraints Misapplication**: Apply constraints

```
        correctly with binary variables enforcing activation.\n3. **
        Parameter Indexing**: Accurately align index references for
        facilities, distances, and regions.\n4. **Objective Calculation
        Mistakes**: Properly associate distances with population served
        by facilities.\n5. **Binary Activation Variables**: Ensure binary
         variables correctly regulate facility activation."
    ]

}

--------------------------------------------------------------------------
```

**Solver Code:**

```python
import gurobipy as gp
from gurobipy import GRB

# Parameters
num_facilities = 10
num_customers = 20
facility_fixed_cost = [8517, 5068, 9433, 6127, 6033, 5966, 7762, 9406,
    6602, 7040]
transport_cost = [
    [80, 94, 44, 51, 190, 44, 129, 178, 129, 91, 172, 119, 177, 150, 90,
        51, 53, 97, 184, 87],
    [139, 33, 104, 135, 50, 176, 97, 121, 47, 29, 186, 163, 149, 108,
        156, 169, 100, 160, 153, 85],
    [153, 36, 18, 170, 18, 181, 178, 68, 171, 106, 159, 110, 21, 106, 91,
        29, 144, 140, 155, 116],
    [103, 59, 78, 125, 14, 11, 152, 95, 76, 173, 36, 148, 75, 132, 59,
        153, 113, 74, 185, 71],
    [193, 186, 130, 145, 114, 150, 33, 154, 20, 75, 103, 30, 137, 131,
        167, 32, 53, 150, 176, 166],
    [159, 130, 156, 65, 36, 59, 199, 124, 104, 72, 180, 73, 43, 152, 143,
        90, 161, 65, 172, 141],
    [173, 121, 110, 127, 22, 159, 195, 137, 47, 10, 87, 11, 154, 66, 126,
        60, 152, 54, 20, 25],
    [181, 34, 186, 152, 109, 195, 133, 198, 30, 65, 69, 19, 109, 143,
        108, 196, 59, 133, 10, 123],
    [82, 113, 147, 21, 88, 24, 38, 16, 70, 122, 148, 192, 116, 108, 18,
        20, 143, 18, 116, 142],
    [176, 170, 87, 91, 195, 183, 124, 89, 72, 97, 89, 23, 45, 196, 97,
        27, 83, 81, 171, 148]
]
facility_capacity = [301, 291, 453, 472, 492, 375, 493, 297, 331, 246]
customer_demand = [117, 86, 69, 53, 110, 74, 136, 140, 126, 79, 54, 86,
    114, 76, 136, 73, 144, 51, 53, 120]

# Create a new model
model = gp.Model('Capacitated Facility Location')

# Decision Variables
x = model.addVars(num_facilities, num_customers, vtype=GRB.CONTINUOUS,
    name='x')
y = model.addVars(num_facilities, vtype=GRB.BINARY, name='y')

# Objective Function: Minimize total cost
```

```python
model.setObjective(
    gp.quicksum(facility_fixed_cost[i] * y[i] for i in range(
        num_facilities)) +
    gp.quicksum(transport_cost[i][j] * x[i, j] for i in range(
        num_facilities) for j in range(num_customers)),
    GRB.MINIMIZE
)

# Constraints
# Demand Satisfaction
for j in range(num_customers):
    model.addConstr(gp.quicksum(x[i, j] for i in range(num_facilities))
        == customer_demand[j])

# Capacity Limits
for i in range(num_facilities):
    model.addConstr(gp.quicksum(x[i, j] for j in range(num_customers))
        <= facility_capacity[i] * y[i])

# Non-negativity
for i in range(num_facilities):
    for j in range(num_customers):
        model.addConstr(x[i, j] >= 0)

# Optimize the model
model.optimize()

# Output
if model.status == GRB.OPTIMAL:
    print('Optimal Objective Value:', model.objVal)
    selected_facilities = [i for i in range(num_facilities) if y[i].x >
        0.5]
    allocation = {(i, j): x[i, j].x for i in range(num_facilities) for j
        in range(num_customers) if x[i, j].x > 0}
    print('Selected Facilities:', selected_facilities)
    print('Allocation:', allocation)
else:
    print('No optimal solution found.')
```

