# OpenReview forum: "OptiTree: Hierarchical Thoughts Generation with Tree Search for LLM Optimization Modeling"
_NeurIPS.cc/2025/Conference — NeurIPS 2025 poster_

### Official Review · Reviewer_9LBo · 2025-06-24

**Clarity:** 4
**Significance:** 3
**Originality:** 3
**Rating:** 4
**Confidence:** 4

**Summary:**

OptiTree proposes a tree-search framework that adaptively breaks down complex operations-research problems into a sequence of simpler subproblems, navigating a “modeling tree” whose nodes store distilled hierarchical modeling thoughts for large-language-model guidance. During inference, the system selects the most relevant subproblems, assembles their thoughts into a unified modeling plan, and continually updates the tree to preserve subproblem order while capturing new decomposition patterns. Experiments on five optimization-modeling benchmarks show OptiTree lifts solving accuracy by over 10 percentage points versus state-of-the-art prompt and fine-tuned baselines.

**Questions:**

* Can you include results when OptiTree is applied on top of reasoning LLMs (e.g., GPT-o1, DeepSeek-R1) and fine-tuned LLMs (e.g., ORLM, LLM-OPT). This would make the empirical case even more compelling.
* What is the setup for comparing Efficiency? Because runtime is dominated by tree breadth/depth and LLM calls, does OptiTree’s speed-up stem mainly from exploring a smaller search tree (fewer iterations), or from other engineering choices? Is bigO time complexity possible to provide?  How does OptiTree scale for complex problems?
* Since high “variable-definition failure” rates motivate the method, can you quantify how much OptiTree lowers this specific error relative to baselines—ideally with a before-and-after breakdown.

**Ethical Concerns:**

["NO or VERY MINOR ethics concerns only"]

**Final Justification:**

The rebuttal resolved my main concerns: added metrics beyond objective value, clarified efficiency setup, and linked variable-definition errors to decomposition with quantified improvements. Additional experiments (robustness, diverse LLM backbones, error breakdown) strengthen the technical contribution and generalizability. Remaining issues are minor and do not affect overall merit. Given these clarifications, I maintain my positive score.

**Limitations:**

yes

**Quality:**

3

**Strengths And Weaknesses:**

# Strengths

* Introduces a modeling-thought tree that stores and retrieves subproblem reasoning snippets, adding a clear structural advance over standard chain-of-thought prompting.
* Yields double-digit accuracy gains and nearly perfect code-execution rates while cutting inference latency versus prior prompt pipelines.
* The paper motivates its design with diagnostic analyses, intuitive diagrams, and thorough ablations that clarify each component’s contribution.
* The method generalize well across all benchmark when the tree is constructed with different dataset.

# Weaknesses

* Focuses on exact objective matches (accuracy). Can provide deeper insight with measuring near-optimality, solver tolerance, or robustness—key factors in real-world optimization.
* The link between diagnosed variable-definition errors and the chosen decomposition remedy improvements are not convincingly demonstrated.
* Setup for efficiency comparison is unclear.

---

> ### Author Rebuttal · Authors · 2025-07-31
>
> Thank you for your valuable comments. We sincerely hope that we could properly address your concerns.
>
> > W1: Focuses on exact objective matches. Deeper insight with measuring near-optimality, solver tolerance, or robustness.
>
> We provide additional results on modeling accuracy beyond just the objective value. However, many benchmarks lack labeled models in their datasets. Therefore, we conducted experiments on a subset of the OptMATH training set (100 problems) that includes annotated models. Since our method and the baselines were not trained on this dataset, the comparison is both safe and fair.
>
> - **Optimality and Tolerance**. We present the solver execution time and code pass rate in Table 1. The code pass rate indicates the ratio of code without bugs. In the existing optimization modeling datasets, we solve the optimization models to optimal solutions within seconds, so we do not set solver tolerances for these experiments.
> - **Robustness**. We ask the LLM to randomly rewrite the problem and test performance on these rewritten versions. The rewriting process alters 10% of the sentences by changing their order and rephrasing another 10% while maintaining their meaning. Results in Table 2 demonstrate the model's robustness to perturbations in the problem.
> - For **deeper insights** into the optimization modeling, we provide more metrics on the correctness of the optimization models. To further address your concerns, we find the datasets with annotated models and provide the proportion that LLM-generated models match with the ground-truth models in some statistical information, such as the number of variables, binary variables, integer variables, and constraints of the optimization models.  As shown in Table 1, OptiTree achieves the highest matching ratio with the ground-truth model, underscoring its reliability.
>
> **Table 1**: More metrics.
>
> |OptMATH|Var Num|Bin Num|Int Num|Cons Num|ACC|Solver Time|Code Pass|
> |-|-|-|-|-|-|-|-|
> |DeepSeek-R1|79.0|85.0|67.0|46.0|66.0|0.0047|97.0|
> |OpenAI-o1|83.0|87.0|**93.0**|50.0|68.0|0.0042|98.0|
> |ORLM|43.0|52.0|44.0|35.0|37.0|0.0040|92.0|
> |LLMOPT|56.0|65.0|57.0|37.0|41.0|0.0040|93.0|
> |CoE (GPT-4o)|67.0|78.0|85.0|46.0|58.0|0.0041|93.0|
> |OptiMUS (GPT-4o)|84.0|79.0|76.0|43.0|61.0|0.0043|96.0|
> |MCTS (GPT-4o)|75.0|80.0|81.0|46.0|63.0|0.0045|95.0|
> |**Ours (GPT-4o)**|**90.0**|**92.0**|83.0|**68.0**|**71.0**|0.0040|**98.0**|
>
> **Table 2**: Rubustness of OptiTree.
>
> ||Nl4Opt|MAMO ComplexLP|IndustryOR|
> |-|-|-|-|
> |OptiTree|96.2|81.0|48.0|
> |OptiTree (Perturbed)|95.8|80.1|46.0|
>
>
> > Q1: Results when OptiTree is applied on top of reasoning LLMs
>
> We conducted experiments to build OptiTree on various LLMs. The results are shown in Table 3, demonstrating that OptiTree generalizes well across different LLMs and significantly improves their performance.
>
> **Table 3**: Results on more LLMs.
>
> ||NL4Opt|MAMO EasyLP|MAMO ComplexLP|ComplexOR|IndustryOR|
> |-|-|-|-|-|-|
> |DeepSeek-R1|86.1|79.5|57.3|68.4|38.0|
> |Ours (DeepSeek-R1)|97.9|96.9|82.5|84.2|57.0|
> |OpenAI-o1|87.1|87.6|54.5|73.6|40.0|
> |Ours (OpenAI-o1)|96.5|95.1|83.9|84.2|53.0|
> |ORLM|85.7|82.3|37.4|63.2|38.0|
> |Ours (ORLM)|93.7|88.8|72.5|79.0|44.0|
> |LLMOPT|93.0|97.0|68.0|72.7|46.0|
> |Ours (LLMOPT)|93.6|96.2|78.7|84.2|48.0|
>
> > W3&Q2: Setup for efficiency comparison is unclear.
>
> The efficiency comparison setup involves measuring the average time, in seconds, for each method to solve a single problem from the various benchmark datasets.
>
> - The result in Table 2 of the main text represents the inference time required to generate the final optimization model for each new problem. We (1) use tree search to find an appropriate problem and (2) perform optimization modeling, combining the modeling thoughts from the problem and we solve the model. The total time includes both the tree search and modeling phases.
> - The total runtime for each prompt-based baseline (like CoE, OptiMUS, and MCTS) and for the OptiTree pipeline was recorded for every problem attempted. These times were then averaged across each dataset.
> - We know you may wonder about the solver execution time for the solver code of each problem. This time is included in the modeling time. As shown in Table 1, the solver time is under a second, which can be considered negligible.
>
> > Q2: Reason for OptiTree to speed up. Time complexity analysis. How does OptiTree scale for complex problems?
>
> - **The high efficiency of OptiTree**.The speed of OptiTree, especially compared to other prompt-based methods, can be attributed to two main factors.
>
>   - OptiTree explores a **much smaller search space**. Existing methods involve iterative searches through variables, constraints, and objective functions, which can exponentially enlarge the search space as problem complexity increases. In contrast, OptiTree searches predefined candidate subproblems, requiring only the identification of the most suitable subproblem within a finite set, leading to greater efficiency.
>   - **OptiTree has fewer iterations in the workflow**. Existing methods often involve repeated calls to certain agents in a multi-agent system for optimization modeling. For example, for complex problems, existing methods may call the terminology interpretation repeatedly. These methods use a management agent to control the workflow automatically. However, the agent's hallucination can lead to suboptimal decision chains. In contrast, OptiTree avoids this multi-agent system, resulting in a more streamlined and efficient workflow.
>
> - We try to estimate the **time complexity** of OptiTree. Given a problem $\mathcal{P}$, we analyze its two main components: Tree Search and Modeling.
>
>   - The complexity of tree search is influenced by the depth of the search path $H$ and the number of children at each node for comparison $NumChild_h$ in each step $h$.
>
>   -  The modeling time is influenced by the complexity of the original problem description $Com(\mathcal{P})$ and the length of the modeling prompt. The promplt length can be roughly esitimated by $l(\mathcal{P})+l(\mathcal{T}(\mathcal{P}))+C$, depending on the length of problem description $l(\mathcal{P})$ and the modeling thoughts $l(\mathcal{T}(\mathcal{P}))$. The notation $C$ is the fixed length of the prompt template.
>
>   - We suppose further that the time for LLM generation is $\alpha\cdot Com\cdot l$, where $\alpha$ is a constant, $Com$ is the reasoning complexity ratio measuring the reasoning complexity of the task, and $l$ is the prompt length.  For subproblem identification, the reasoning complexity is low and set to 1 for simplicity. Suppose further that $l(child)$ is the text length for each child in the tree. Thus, the total time estimation is given by
>
>     $\alpha\sum_{h=1}^H NumChild_h\cdot l(child)+\alpha\cdot Com(\mathcal{P})\cdot(l(\mathcal{P})+l(\mathcal{T}(\mathcal{P}))+C)$
>
> - **The scalability of OptiTree**. OptiTree is designed to scale for problems with longer question lengths or higher complexities. For longer question lengths, we avoid implementing a complex multi-agent workflow that calls agents repeatedly for specific tasks. Instead, we search within a finite subproblem space, significantly reducing the number of requests to the LLM. As shown in Table 2 of the main text, the search process is efficient.
>
> > W2&Q3: The link between diagnosed variable-definition errors and the chosen decomposition remedy. Quantify how much OptiTree lowers this specific error.
>
> We explain the link between them.
>
> - **Variable Definition Challenge**. Following the paper of OptiMUS, the errors mainly come from three aspects: missing or wrong constraints, incorrect modeling and coding errors. **The incorrect models mostly come from the wrong variable definition**, e.g., defining the wrong binary variables in the model. Almost half of the errors of OptiMUS come from the incorrect modeling (62.5% for ComplexOR). Our initial analysis in **Motivation 1** pinpoints that this is also the main error for CoT. For the complex OR problems, directly defining variables is a challenging task for LLMs. This observation motivated us to enhance the variable definition process using step-wise guidance.
> - **Our approach is addressing the incorrect modeling from the perspective of variable definition**. We do so in the three aspects.
>   - **Step-by-Step Decomposition**. Instead of defining complex variables for the entire problem at once, OptiTree decomposes the task into modeling a series of simpler subproblems. Modeling a subproblem and correctly defining its variables are less challenging tasks for an LLM. Our experiment in **Motivation 3** confirms this, showing a significant accuracy improvement when providing the LLM with a ground-truth model of a subproblem first.
>   - **Enhancement with Modeling Thoughts**. The process is further enhanced by retrieving modeling thoughts for the identified subproblem. As shown in the schema examples, these thoughts provide explicit, step-by-step guidance, starting with how to **define variables**. This provides a strong, reliable template that the LLM can adapt.
>
>
> - We present the percentage of all kinds of mistakes for methods in Table 4. OptiTree reduces the **errors of variable definitions**.
>
> **Table 4**: Failure cases information.
>
> ||ComplexOR|||IndustryOR|||
> |-|-|-|-|-|-|-|
> ||CoT|OptiMUS|OptiTree|CoT|OptiMUS|OptiTree|
> |Wrong Variables/In correct modeling|54.5|62.5|0|59.1|60.6|10.9|
> |Missing constraints|18.1|12.6|66.7|24.0|18.2|82.6|
> |Coding errors|27.4|24.9|33.3|16.9|21.2|6.5|

---

### Official Review · Reviewer_vacN · 2025-06-24

**Clarity:** 1
**Significance:** 3
**Originality:** 3
**Rating:** 4
**Confidence:** 3

**Summary:**

This paper develops a method to improve LLM-based optimization modeling through a modeling tree that organizes a wide range of OR problems based on their hierarchical problem taxonomy and complexity. Then, they recurrently search the tree to identify a series of simpler subproblems and synthesize the global modeling thoughts. The authors perform thorough experiments to show the performance benefits of their method.

**Questions:**

1. While the authors see improvements using GPT-4o and deepseek-v3 to build the tree, can your method further improve (1) stronger models like the reasoning models GPT-o1/3 and deepseek-R1? (2) weaker models like qwen and llama, if they use these models to build the tree and construct solutions?
2. I wonder if the authors proposed pipeline is sensitive to different dataset used to build the tree? Also, if the pipeline is sensitive of different seeds (do you get drastically different trees if you run the code with different seeds)?
3. Main paper, Line 221: if $P \subseteq P_k^{(M)}$: is this a typo, should it be the other way around $P_k^{(M)} \subseteq P$?
4. Figure 2: can the authors provide a few examples of the errors in IndustryOR and how their methods help the performance?
5. Can the authors provide the entire tree built from the algorithm and a minimal script to reproduce the result so that we can examine the meaningfulness of the tree and the algorithm?

**Ethical Concerns:**

["NO or VERY MINOR ethics concerns only"]

**Final Justification:**

Overall, I find this paper interesting and the results reported promising, but I feel like this paper (in the current form) lacks clarity and reproducibility due to the missing details, typos, and lack of open-source code in the submission stage. I increased the score from 3 to 4 to acknowledge the authors' effort in the rebuttal stage, but due to the reproducibility concerns, I personally feel like this paper would be significant stronger if it undergoes another run of revision and resubmission.

**Limitations:**

The authors discussed limitations in Sec F of the Appendix.

**Quality:**

2

**Strengths And Weaknesses:**

**Strengths:**

1. The high level concepts proposed in the paper make sense and the performance substantially improves from existing methods, especially on the MAMO complex dataset where they raise the performance from 56.8% (previous methods) to 81.5% (their methods).
2. The authors provide a few ablation studies to justify their methods empirically.
3. Their motivated observations in Sec. 3 seem insightful.

**Weaknesses:**

 1. The pseudocode provided in the appendix seems unclear and insufficient to reproduce the entire experiments. There are many things that I find confusing, and the authors should revise the pseudocode to provide more clarity:
     - Page 19: you should explicitly refer to the exact name of the prompt you are using on each line. Just a few examples I find confusing:
        1. I'm confused about which prompt is used to “synthesize $Tau(P)$ from $Tau(P(M))$” (Algorithm 1, Line 21) - your modeling prompts on page 25-27 does not seem to require the $Tau(P(M))$ information, so how do you generate modeling thoughts by combining information from the maximal subproblem’s modeling thought?
        2. Algorithm 2, line 3: “modeling fails” - please make it more clear where is this executed and what does this mean.
        3. Algorithm 2, line 4: What prompt is used to “distill schema (problem type, CP, $Tau(P)$) from P”?
        4. Algorithm 2, Line 5: $P(M) ← MaximalSubproblem(P, T)$: is this $P(M)$ the same $P(M)$ that you get from Algorithm 1, Line 20?
    - Page 23: Modeling Thoughts Distillation: there seems to be two prompts (separated by “You are given a specific combinatorial optimization problem.” on page 24. what’s the difference?
    - Page 23: Code Correction: how is this prompt used? How many code correct iterations do you do and when do you do it? How much performance benefit do you get from code correction? It seems like the authors did not mention this part at all in the main paper or in the pseudocode.
    - What’s the difference between the Subproblem Identification prompt (Page 22) and Add New Tree Nodes prompt (Page 27)? Is it just in Page 22 you output an extra similarity score, but for Page 27 you don’t.
    - Page 24: The “constraints” information in the statement thoughts outputs - are they used anywhere? Do you only use <the more precise of statement thoughts> as the “statement thoughts” for your subproblem matching, or you use the whole json for the matching?

As a side remark (reproducibility): I strongly encourage the authors to provide the code during the rebuttal period. It seems like this field has issue with code opensourcing, publishing papers without code or delay the opensourcing detrimentally hurt the field’s progress, especially the paper heavily depends on prompting off-the-shelf LLMs.

2. The authors rely on off-the-shelf LLMs for their entire algorithm, which raises the question of robustness of their method (see question 2 for additional questions). The authors should also have a discussion regarding when their tree modeling fails, given that the reliance on the off-the-shelf LLMs.
3. The authors should report the performance on recent benchmarks including Optmath and Optibench.

---

> ### Author Rebuttal · Authors · 2025-07-31
>
> Thank you for your valuable comments. We sincerely hope that we could properly address your concerns.
>
> > W1: The pseudo-code and prompts
>
> - We first explain the pseudocode.
>
>   - **Global thoughts synthesis process**. We apologize for not including this prompt in the Appendix. We generate modeling thoughts by integrating information from the maximal subproblem’s modeling thought and the current problem description.
>
>     > This is an OR problem with the following problem description
>     >
>     > {problem description}
>     >
>     > The problem contains a subproblem of {subproblem name}. The modeling thoughts to model this subproblem are as follows. Please refer to the thoughts and list the modeling thoughts for the original OR problems.
>     >
>     > {subproblem modeling thoughts}
>     >
>
>   - **Modeling fails** means that the final objective value of the generated optimization model does not equal the ground-truth value for that problem. In the tree update, if the generated answer for the current problem is incorrect, the system determines that the current modeling tree is insufficient for that problem type and updates the current problem into the tree.
>
>   - **The distillation prompts** are listed in Appendix H.4. The first block of prompts is used for statement thoughts distillation, and the second block of prompts is used for modeling thoughts and schema distillation.
>
>   - Yes. The **maximal subproblem** is the same as Algorithm 1.
>
> - Two prompts for **modeling thoughts distillation**. We apologize for mislabeling. These blocks of prompts are used for statement thought distillation.
>
>   - The first prompt is used for situations when we have to find subproblems in the previous search. It asks the LLM to take the specific problem and its current basic subproblem type as input and then determine a more precise subproblem.
>   - The second prompt is used for situations when the search process stops at the root node. Thus, we cannot refer to any prior subproblem for thought construction.
>
> - **Code correction**. We will revise the manuscript to include it for clarity.
>
>   - The **code correction module** is a standard component of the prompt-based optimization modeling methods, such as CoE and OptiMUS. It is applied after the solver codes generated by the LLM raise errors. The prompt instructs the LLM to analyze the bugs in the code. We use **one iteration** of this correction step.
>   - **The code correction is not a core contribution and a trick in our work**. We find in practice that the trigger of the code correction module is less than 3%. This implies that the successful execution rate for the initial codes is high, and the benefits of the correction prompt are not significant.
>
>
> - The **Subproblem Identification Prompt** is used during the **tree search** to find the single best subproblem to proceed with at each level. It returns the single best matching subproblem and a JSON object containing a similarity score.
>
>   **Add New Tree Nodes Prompt** is used during the **tree update** phase to correctly position a new node within the tree's hierarchy. The LLM uses the prompt and identify all parent-child relationships between a new problem and its potential siblings. It returns a list of all problem types that are identified as subtypes of the primary one; it does not output a similarity score.
>
> - Yes, the "constraints" information is used, and the entire JSON object is used for the subproblem matching.
>
>
> > W2: When their tree modeling fails.
>
> Following the paper of OptiMUS, the errors mainly come from three aspects: missing or wrong constraints, incorrect modeling and coding errors. Almost half of the errors of OptiMUS come from the incorrect modeling (62.5% for ComplexOR), e.g., defining the wrong binary variables in the model. The results for OptiTree are presented in Table 1. OptiTree reduces the errors of incorrect models and the constraint errors are the main error for OptiTree.
>
> - First, accurately formulating constraints requires a deep and nuanced **understanding of the specific problem scenario and its domain knowledge**. For highly specialized or novel OR problems, existing background knowledge may not fully capture the intricate details necessary for precise constraint definitions.
> - The second limitation is the intrinsic **reasoning ability of the base LLM**. This framework may struggle with problems requiring significant logical or relational reasoning. For instance, in the Staff Rostering problem, the model must effectively allocate staff for 24-hour shifts to meet coverage constraints (the workers work for 8 hours a day). If it fails to recognize that the demand period from 1:00 to 2:00 AM is covered by workers who began their shifts at 8:00 PM, it will result in an incorrect constraint and ultimately a flawed model.
>
> **Table 1**: Failure cases information.
>
> ||ComplexOR|IndustryOR|OptMATH|
> |-|-|-|-|
> |Incorrect modeling|0|10.9|23.8|
> |Missing constraints|66.7|82.6|61.9|
> |Coding errors|33.3|6.5|14.3|
>
> > W3: Performance on recent benchmarks
>
> We include the fine-tuned baselines, Evo-Step and OptMATH. We also include two new benchmarks, the OptiBench and OptMATH datasets for testing. The fine-tuned models for Evo-Step and OptMATH have not yet been released, so results are drawn from the original papers. The results demonstrate the superior performance of OptiTree over the baselines across benchmarks.
>
> **Table 2**: The comparisons with more datasets and baselines.
>
> ||NL4Opt|MAMO EasyLP|MAMO ComplexLP|ComplexOR|IndustryOR|OptiBench|OptMATH|
> |-|-|-|-|-|-|-|-|
> |DeepSeek-R1|86.1|79.5|57.3|68.4|38.0|70.2|33.1|
> |OpenAI-o1|87.1|87.6|54.5|73.6|40.0|71.5|34.9|
> |ORLM|85.7|82.3|37.4|63.2|38.0|51.1|2.6|
> |LLMOPT|93.0|**97.0**|68.0|72.7|46.0|66.4|40.0|
> |Evo-Step|84.5|85.3|61.6|-|36.4|-|-|
> |OptMATH|95.9|89.9|54.1|-|31.0|66.1|34.7|
> |CoE (GPT-4o)|76.4|85.7|46.4|68.4|34.0|43.2|18.6|
> |OptiMUS (GPT-4o)|82.0|85.1|47.3|79.0|34.0|45.8|20.2|
> |MCTS (GPT-4o)|90.3|87.4|56.8|68.4|42.0|64.0|37.3|
> |**Ours (GPT-4o)**|**96.2**|95.6|**81.0**|**84.2**|**48.0**|**71.9**|**45.8**|
> |CoE (DeepSeek-V3)|79.2|85.9|43.1|63.2|33.0|55.2|24.1|
> |OptiMUS (DeepSeek-V3)|80.6|87.1|45.2|79.0|36.0|58.8|32.5|
> |MCTS (DeepSeek-V3)|89.6|88.0|51.6|79.0|46.0|67.9|38.6|
> |**Ours (DeepSeek-V3)**|**98.3**|**96.9**|**81.5**|**84.2**|**54.0**|**74.7**|**52.4**|
>
> > Q1: Improvement of (1) stronger models like reasoning models, (2) weaker models like Qwen and llama.
>
> We conduct experiments to build OptiTree on (2) stronger models, including GPT-o1 and DeepSeek-R1; (2) weaker models, including Qwen-2.5 7B and Llama3-8B. The results in Table 3 demonstrate that OptiTree can significantly improve the performance of the LLMs.
>
> **Table 3**: Improvement on more models.
>
> ||NL4Opt|MAMO EasyLP|MAMO ComplexLP|ComplexOR|IndustryOR|
> |-|-|-|-|-|-|
> |DeepSeek-R1|86.1|79.5|57.3|68.4|38.0|
> |Ours (DeepSeek-R1)|97.9|96.9|82.5|84.2|57.0|
> |OpenAI-o1|87.1|87.6|54.5|73.6|40.0|
> |Ours (OpenAI-o1)|96.5|95.1|83.9|84.2|53.0|
> |Qwen2.5 14B|79.4|79.6|45.0|57.9|31.0|
> |Ours (Qwen2.5 14B)|88.2|89.6|78.7|73.6|37.0|
> |Llama3.1-8B|40.5|71.2|39.8|63.1|24.0|
> |Ours (Llama3.1-8B)|55.0|75.9|64.4|73.7|28.0|
>
> > Q2: Sensitivity to different datasets used to build the tree and different seeds.
>
> - First, we show that the pipeline is not sensitive to the different datasets used to build the tree. We conduct the following experiments. (1) We randomly select 400 different problems in the OR-Instruct dataset for tree construction. (2) We randomly select 100 problems for tree construction. (3) We select problems from the NL4Opt and MAMO ComplexLP datasets for tree construction. We found that OptiTree maintains stable performance across different datasets.
>
> - Second, we run the code with different seeds for tree construction and testing.  We find that OptiTree achieves a stable performance across the random seeds.
>
>
> Table 4: Sensitivity analysis.
>
> ||MAMO ComplexLP|ComplexOR|IndustryOR|
> |-|-|-|-|
> |Randomly Selected 1|78.7|79.0|46.0|
> |Randomly Selected 2|82.0|84.2|48.0|
> |Randomly Selected 3|79.6|84.2|48.0|
> |Fewer Problems 1|81.5|84.2|48.0|
> |Fewer Problems 2|80.5|84.2|46.0|
> |NL4Opt (all)|74.9|73.7|44.0|
> |MAMO Complex (20 problems)+NL4Opt (all)|84.4|79.0|46.0|
> |Tree Construction Seed 1|83.9|84.2|48.0|
> |Tree Construction Seed 2|78.7|84.2|46.0|
> |Tree Construction Seed 3|82.9|79.0|48.0|
>
> > Q3: Line 221: Is this a typo?
>
> $P$ is a subproblem of $P_k^{(M)}$, and thus $P$ is the parent as the deeper node represents a more complex problem, i.e., $P \subseteq P_k^{(M)}$.
>
> > Q4: Figure 2: Examples of the in IndustryOR
>
> We provide a simplified example due to the space limit.
>
> - Problem: A store plans to purchase and sell the goods for the first quarter of next year. The warehouse capacity is $M$ units, and the initial goods in stock are $M_0$ units. Determine the units to be purchased and sold each month to maximize total profit, with purchase and sales prices for the $i^{th}$ month denoted as $p_i$ and $s_i$.
>
> - Analysis. OptiTree can reduce the errors in incorrect models.
>
>   - **Without OptiTree**. Define $x_i$ as the units purchased at the beginning of each month and $y_i$ as the units sold each month.
>
>     $max_{x,y}\sum_i s_iy_i-p_ix_i$
>
>     $x_0-y_0+M_0\le M$
>
>     $x_i-y_i\le M,i>0$, inventory constraints
>
>   - **OptiTree** identifies this problem as an inventory problem, with $I_i$ representing the inventory at each month.
>
>     $max_{x,y}\sum_i s_iy_i-p_ix_i$
>
>     $I_0=M_0$
>
>     $I_i=I_{i-1}+x_i-y_i$
>
>     $I_{i-1}+x_i\le M$, inventory constraints
>
>     $y_i\le I_{i-1}+x_i$, sale constraints
>
>
> > Q5: The code for the paper
>
> Thank you for your interest in our work and your valuable suggestions. However, **we are sorry that authors cannot include links to external pages (including anonymous repositories) during the rebuttal period**. We will publicly release all code and final modeling trees shortly.

---

> ### Author Response · Authors · 2025-08-04
>
> Dear Reviewer vacN,
>
> Thank you very much for taking the time to read our rebuttal and for promptly acknowledging it. We truly appreciate your engagement with our work during the review process.
>
> To further address your concerns, we provide the revised pseudo-code following your suggestions as follows.
>
> ----
>
> Algorithm 1 Tree Search for Subproblem Decomposition
>
> ----
>
> - Require: Target problem $\mathcal{P}$, modeling tree $\mathbb{T}$
>
> - Ensure: Hierarchical modeling thoughts $\mathcal{T} (\mathcal{P})$, and the maximal subproblem $\tilde{\mathcal{P}}$
>
> - **function** TreeSearch($\mathcal{P}$, $\mathbb{T}$ ):
>
>   - Initialize current node $\mathcal{N}\leftarrow$ The root of the modeling tree $\mathbb{T}$
>   - \# **prompts block 1 and 2 in Appendix H.4**
>
>   - Extract statement thoughts $\mathcal{C}_\mathcal{P}$
>   - \# Identified subproblems
>   - **while** $\mathcal{N}$ has children **do**
>     - Get the children of $\mathcal{N}$
>     - \# **prompts in Appendix H.2**
>     - Identify the subproblem of $\mathcal{P}$ among the children with similarity scores
>     - **If** $\mathcal{P}$ has a subproblem among the children **then**
>       - $\mathcal{N}\leftarrow$ Subproblem with the max similarity score
>       - The maximal subproblem $\tilde{\mathcal{P}}\leftarrow \mathcal{N}$
>     - **else**
>       - $\mathcal{N}\leftarrow$ None
>     - **end if**
>   - **end while**
>
> - Synthesize $\mathcal{T}(\mathcal{P})$ from $\mathcal{T}(\tilde{\mathcal{P}})$
>
> - **return** $\mathcal{T}(\mathcal{P})$, $\tilde{\mathcal{P}}$
>
> - **end function**
>
> ----
>
> ----
>
>  Algorithm 2 Modeling Tree Update
>
> ----
>
> - Require: Target problem $\mathcal{P}$, modeling tree $\mathbb{T}$
> - Ensure: Updated Modeling Tree
> - **function** TreeUpdate($\mathcal{P}$, $\mathbb{T}$ ):
>
>   - $\mathcal{T}(\mathcal{P})$, $\tilde{\mathcal{P}}\leftarrow$ TreeSearch($\mathcal{P}$, $\mathbb{T}$)
>   - \# **prompts in Appendix H.3**
>   - Model the problem $\mathcal{P}$ using the modeling thoughts and solve the model for optimal value $y$
>   - **If** $y$ does not equal to the ground-truth objective value then
>     - \# **prompts block 1 and 2 in Appendix H.4**
>     - Distill schema (problem type,$\mathcal{C}_\mathcal{P}$,$\mathcal{T} (\mathcal{P})$) from $\mathcal{P}$
>     - Find the children of the maximal subproblem $\tilde{\mathcal{P}}$
>     - **for** child $\tilde{\mathcal{P}}_k$ in the children **do**
>       - \# **prompts block 3 in Appendix H.4**
>       - **if** $\mathcal{P}\subset_{\mathcal{S}}\tilde{\mathcal{P}}_k$ **then**
>         - Insert $\mathcal{P}$ as parent of $\tilde{\mathcal{P}}_k$ and children of $\tilde{\mathcal{P}}$
>       - **else**
>         - Insert $\mathcal{P}$ as sibling of $\tilde{\mathcal{P}}_k$ and children of $\tilde{\mathcal{P}}$
>       - **end if**
>     - **end for**
>   - **end if**
> - **return** $\mathbb{T}$
> - **end function**
>
> ----
>
>
> If you have any further thoughts, questions, or suggestions---either regarding our current response or the broader direction of the work---we would be genuinely grateful to hear them. We highly value your perspective and would welcome any opportunity for continued discussion or clarification.
>
> Best,
>
> Authors

---

> > ### Author Response · Authors · 2025-08-07
> >
> > Dear Reviewer vacN,
> >
> > We would like to extend our sincere gratitude for the time and effort you have devoted to reviewing our submission. Your positive feedback, insightful comments, and constructive suggestions have been invaluable to us, guiding us in improving the quality of our work!
> >
> > We eagerly await your feedback to understand if our responses have adequately addressed all your concerns. If so, we would deeply appreciate it if you could raise your score. If not, we are eager to address any additional queries you might have, which will enable us to enhance our work further.
> >
> > Once again, thank you for your guidance and support.
> >
> > Best,
> >
> > Authors

---

> > > ### Comment · Reviewer_vacN · 2025-08-07
> > >
> > > Thank you for the revised pseudocode and clarification provided in the rebuttal. Overall, I find this paper interesting and the results reported promising, but I feel like this paper (in the current form) lacks clarity and reproducibility due to the missing details, typos, and lack of open-source code in the submission stage. As the authors make significant effort in improving the clarity of the paper, I'm ok with increasing the score to 4, borderline accept. I understand that it is the neurips policy that does not allow the authors providing the code at rebuttal stage, but I personally feel like it might be much better if this paper undergoes another run of revision and resubmission, so that the reviewers can have more information (e.g. open-source code) to rigorously evaluate the reproducibility of this work. If the paper get accepted this round, I strongly urge the authors to fully open source the code and improve the writing of the paper based on the points discusses in the rebuttal stage.

---

> > > > ### Author Response · Authors · 2025-08-08
> > > >
> > > > Dear Reviewer vacN,
> > > >
> > > > Thank you for your thorough review and for engaging so thoughtfully with our rebuttal. We sincerely appreciate your time, your constructive feedback, and your consideration in raising your score.
> > > >
> > > > We understand and share your concerns regarding the clarity and reproducibility of the work, especially without the open-source code available for review.
> > > >
> > > > If the paper is accepted, we promise to:
> > > >
> > > > - **Fully open-source our work.** We will release the complete, well-documented codebase, the final modeling trees used in our experiments, and clear instructions to reproduce all key results as soon as possible.
> > > > - **Thoroughly revise the manuscript.** We will perform a comprehensive revision to incorporate all the clarifications discussed during the rebuttal stage, improving the overall writing and ensuring all methodological details are present to make the work as accessible as possible.
> > > >   - We will provide more detailed explanations of our methods in the main text and correct any typos based on your valuable suggestions. Additionally, we will include revised pseudocode to improve readability and clarity.
> > > >   - All experiments discussed during the rebuttal will be added to the paper, including results on additional baselines, benchmarks, base models, and sensitivity analyses.
> > > >   - We will include a section analyzing both successful and unsuccessful cases of the proposed method, utilizing examples for clarity.
> > > >   - We will reorganize the prompts in the appendix to ensure they are easy to follow.
> > > >
> > > > Thank you once again for your valuable guidance. Your feedback has been instrumental in helping us improve the paper, and we are committed to elevating it to the highest standard.
> > > >
> > > > Best,
> > > >
> > > > Authors

---

### Official Review · Reviewer_MotJ · 2025-07-02

**Clarity:** 2
**Significance:** 3
**Originality:** 2
**Rating:** 4
**Confidence:** 4

**Summary:**

OptiTree is a prompt-based framework that improves LLM-driven optimization modeling by recursively decomposing a natural-language problem into a chain of simpler subproblems. It constructs an offline modeling tree that organizes OR problem types in a taxonomy; each node stores high-level statement thoughts (key features) and modeling thoughts (variable, constraint, and objective templates). At inference time, an LLM follows the tree from the root to the deepest matching node, assembles the retrieved thoughts into one prompt, and generates solver-ready code.

**Questions:**

1. The paper briefly mentions the limitations of OptiTree. Could the authors elaborate on these limitations in more detail? Specifically, analyzing scenarios where OptiTree may underperform and discussing the potential reasons would provide a more balanced perspective.

For other questions, please refer to the weaknesses.

**Ethical Concerns:**

["NO or VERY MINOR ethics concerns only"]

**Final Justification:**

The rebuttal meaningfully addresses my initial concerns, which significantly strengthens the paper’s clarity and empirical grounding. The authors provide a concrete, fully automated pipeline for constructing and updating the modeling tree. I have raised my score by one.

**Limitations:**

The paper briefly mentions the limitations of OptiTree. Could the authors elaborate on these limitations in more detail? Specifically, analyzing scenarios where OptiTree may underperform and discussing the potential reasons would provide a more balanced perspective.

**Paper Formatting Concerns:**

None.

**Quality:**

2

**Strengths And Weaknesses:**

Strengths:
1.  Focuses on a long-standing bottleneck in operations research by turning LLMs into reliable optimization-modeling assistants.
2. Constructs a Modeling Tree that stores both “statement thoughts” and “modeling thoughts”, turning prompt retrieval into a structured, hierarchical memory for LLMs.
2. Proposes an incremental insertion algorithm that allows the tree to grow continuously without retraining.

Weaknesses:
1.  The paper lacks a clear explanation of how to construct the modeling tree. After reading it, key details remain unclear, such as how to add tree nodes or evaluate whether a problem should be integrated. Is this process fully automated?
2.  How is the greatest search depth determined during tree traversal?
3.  Although the paper shows that the Modeling Tree as a whole improves solution quality, it never measures how performance evolves when thoughts are aggregated level-by-level along the search path. Without such a vertical study, it is hard to verify that deeper, more-specific nodes truly add incremental value (rather than the final node alone doing all the work) and to understand the cost–benefit trade-off between prompt length and accuracy.
4. In the experimental section, it would be beneficial to include comparisons with the latest methods, such as Evo-Step [15] and OptMATH [16].

---

> ### Author Rebuttal · Authors · 2025-07-31
>
> Thank you for your valuable comments. We sincerely hope that we could properly address your concerns.
>
> > W1: Explanation of constructing the modeling tree
>
> The modeling tree construction is **fully automated**. To construct a modeling tree, we need a dataset (OR-Instruct) with various problems containing the problem description and the ground-truth optimization models.
>
> - First, we explain the **tree search process** to find an appropriate subproblem.
>   - OptiTree uses an LLM to distill the description of the target problem into a set of statement thoughts, which summarize the problem's key features and requirements into a schema. The statement thoughts are used for subproblem identification (**Appendix H.1 for an example**).
>
>   - The search begins at the **root node** of the modeling tree. Optitree evaluates the children of the current node to find the best match for the target problem. The search halts when it reaches a node where none of its children qualify as a subproblem for the target problem. The last subproblem found in this process is called the **maximum subproblem**, and its modeling thoughts are used to guide the final solution.
>
>     The simplified prompt for subproblem identification in tree search.
>
>     > Input problem: {input_problem_statement_thoughts}.
>     >
>     > You are given a list of defined subtypes and the statement thoughts of subproblems: {subproblems_statement_thoughts}.
>     >
>     > Your task is to determine if the specific problem belongs to one of the given subtypes. Return only the final answer in the following JSON format:
>     >
>     > ‘‘‘json{
>     >
>     > ​	matching_subtype: "",
>     >
>     > ​	reasons: "",
>     >
>     > ​	similarity: [0,1,2]
>     >
>     > }‘‘‘
>
>     The simplified prompt for modeling using modeling thoughts.
>
>     > You are a mathematical formulator to formulate the given OR problem as an optimization model and generate Gurobi code to solve the model.
>     >
>     > {problem description}
>     >
>     > The problem has a subproblem of {subproblem name}. During the modeling process, we can refer to the subproblem model: {subproblem model}
>
> - Second, we explain the evaluation process **whether a problem should be updated into the tree**. For each problem in a given dataset, we perform a tree search to find the maximum subproblem and use the modeling thoughts to enhance the modeling of the original problem. We solve the generated optimization model and obtain the final answer of the objective value. **If the final answer matches the ground-truth answer**, no update is necessary, indicating that the tree can already address this type of problem. **Otherwise**, the unsuccessful problem is used to expand the tree.
>
>
> - Third, we explain the **adding process of the tree nodes**.
>
>   - **Distill a Schema**: First, a new modeling schema (including problem type, statement thoughts, and modeling thoughts) is automatically distilled for the failed problem. The prompt is as follows.
>
>      > Your task is to summarize the type of this specific problem and provide a detailed statement thoughts of this type. You must classify the problem into more precise scenarios, such as the Traveling Salesman Problem (TSP), the facility location problem, and so on.
>      >
>      > Specific Problem: {problem description}
>      >
>      > Here is an output format example ...
>
>   - **Position Among Siblings**: The new node is placed as a child of the maximum subproblem. The system then uses an LLM to verify the relationship between the new problem and the existing children of the maximum subproblem.
>
>      - If P is a subproblem of an existing child, P is inserted between the parent and that child.
>      - Otherwise, P is inserted as a new direct child, becoming a sibling to the others.
>
>      > You are given a primary problem type and its statement thoughts.
>      >
>      > {problem_statement_thoughts}
>      >
>      > You are also provided with a list of other problems with their statement thoughts. Your task is to determine which problem types in the list are subtypes of the given primary problem type.
>      >
>      > {list_of_problem_types_and_statement_thoughts}
>
> > W2: What is the greatest search depth?
>
> We report the greatest search depth across different datasets. For the harder datasets, the search depth tends to increase, indicating that more complex problems feature more sophisticated subproblem structures.
>
> **Table 1**: The greatest search depth.
>
> ||NL4Opt|MAMO EasyLP|MAMO ComplexLP|ComplexOR|IndustryOR|
> |-|-|-|-|-|-|
> |Greatest Search Depth|4|4|10|9|10|
>
> > W3: Performance evolution when thoughts are aggregated along the search path/ cost–benefit trade-off between prompt length and accuracy.
>
> - We have conducted the related experiments in Section 5.4, Impact of the Search Depth. We compared the performance of OptiTree with an unrestricted search depth against variants with maximum depths of 1 or 3. The results in Table 3 demonstrate that performance improves with deeper searches.
> - We analyze the prompt length for the modeling step, which includes the fixed prompt template, the problem length, and the length of the modeling thoughts. Table 2 shows that the prompt length does not significantly increase as the search goes deeper.
> - Given that we only use the thoughts from the final node, the cost-benefit trade-off is not about a growing prompt length, but rather about **search time vs. accuracy**. The search time for different search depths is presented as follows. Table 3 shows that the cost is a small component of the overall efficient process.
>
> **Table 2**: The prompt length for the problems (characters).
>
> ||NL4Opt|MAMO EasyLP|MAMO ComplexLP|ComplexOR|IndustryOR|
> |-|-|-|-|-|-|
> |Depth=1|4201|5294|5837|6302|5018|
> |Depth=2|4143|5382|5795|6512|4923|
> |Depth=3|4340|5434|5893|6659|5195|
>
> **Table 3**: The comparisons of trade-off between search depth (search time) and the performance.
>
> |||NL4Opt|MAMO EasyLP|MAMO ComplexLP|ComplexOR|IndustryOR|
> |-|-|-|-|-|-|-|
> |Depth=1|Time|10.3|6.1|10.1|23.0|14.2|
> ||ACC|93.8|93.3|75.4|68.4|40.0|
> |Depth=3|Time|12.0|8.5|12.5|26.5|15.6|
> ||ACC|96.2|95.6|80.1|79.0|44.0|
> |No limit Depth|Time|13.9|9.3|13.3|31.0|19.9|
> ||ACC|96.2|95.6|81.0|84.2|48.0|
>
> > W4: Comparisons with the latest methods, such as Evo-Step and OptMATH
>
> We include Evo-Step and OptMATH in our baselines. The Evo-Stop and OptMATH are fine-tuned models. We also include two new benchmarks, the OptiBench and OptMATH datasets for testing. As the fine-tuned models for Evo-Step and OptMATH have not yet been released, results are drawn from the original papers. The results illustrate the superior performance of OptiTree compared to these baselines across benchmarks.
>
> **Table 4**: The comparisons with more datasets and baselines.
>
> ||NL4Opt|MAMO EasyLP|MAMO ComplexLP|ComplexOR|IndustryOR|OptiBench|OptMATH|
> |-|-|-|-|-|-|-|-|
> |DeepSeek-R1|86.1|79.5|57.3|68.4|38.0|70.2|33.1|
> |OpenAI-o1|87.1|87.6|54.5|73.6|40.0|71.5|34.9|
> |ORLM|85.7|82.3|37.4|63.2|38.0|51.1|2.6|
> |LLMOPT|93.0|**97.0**|68.0|72.7|46.0|66.4|40.0|
> |Evo-Step|84.5|85.3|61.6|-|36.4|-|-|
> |OptMATH|95.9|89.9|54.1|-|31.0|66.1|34.7|
> |CoE (GPT-4o)|76.4|85.7|46.4|68.4|34.0|43.2|18.6|
> |OptiMUS (GPT-4o)|82.0|85.1|47.3|79.0|34.0|45.8|20.2|
> |MCTS (GPT-4o)|90.3|87.4|56.8|68.4|42.0|64.0|37.3|
> |**Ours (GPT-4o)**|**96.2**|95.6|**81.0**|**84.2**|**48.0**|**71.9**|**45.8**|
> |CoE (DeepSeek-V3)|79.2|85.9|43.1|63.2|33.0|55.2|24.1|
> |OptiMUS (DeepSeek-V3)|80.6|87.1|45.2|79.0|36.0|58.8|32.5|
> |MCTS (DeepSeek-V3)|89.6|88.0|51.6|79.0|46.0|67.9|38.6|
> |**Ours (DeepSeek-V3)**|**98.3**|**96.9**|**81.5**|**84.2**|**54.0**|**74.7**|**52.4**|
>
> > Q1: Analyzing scenarios where OptiTree may underperform and discussing the potential reasons.
>
> - **Failure cases analysis**. Following [4], the errors mainly come from three aspects: missing or wrong constraints, incorrect modeling and coding errors. In [4], almost half of the errors come from the incorrect modeling (62.5% for ComplexOR), e.g., defining wrong binary variables in the model. We analyze the failures of OptiTree and normalize the failure rates to sum to 1. We find that most errors are from the missing constraints. **OptiTree can significantly reduce the error of incorrect modeling, which mainly comes from the wrong variable definitions**.
> - **Limitations**. The constraint errors are the main error for OptiTree.
>   - First, accurately formulating constraints requires a deep and nuanced **understanding of the specific problem scenario and its domain knowledge**. For highly specialized or novel OR problems, existing background knowledge may not fully capture the intricate details necessary for precise constraint definitions.
>   - The second limitation is the intrinsic **reasoning ability of the base LLM**. While OptiTree provides a structured framework and knowledge, the final synthesis depends on the LLM's inherent reasoning capabilities. This framework may struggle with problems requiring significant logical or relational reasoning. For instance, in the staff rostering problem, the model must allocate staff for 24-hour shifts to meet coverage constraints (with workers working 8-hour days). If it fails to recognize that the demand period from 1:00 to 2:00 AM is covered by workers who began their shifts at 8:00 PM, it will result in an incorrect constraint and ultimately a flawed model.
>
> **Table 5**: Failure cases information.
>
> ||ComplexOR|IndustryOR|OptMATH|
> |-|-|-|-|
> |Incorrect modeling|0|10.9|23.8|
> |Missing constraints|66.7|82.6|61.9|
> |Coding errors|33.3|6.5|14.3|
>
> [1] Evo-Step: Evolutionary Generation and Stepwise Validation for Optimizing LLMs in OR.
>
> [2] OptMATH: A Scalable Bidirectional Data Synthesis Framework for Optimization Modeling.
>
> [3] OptiBench Meets ReSocratic: Measure and Improve LLMs for Optimization Modeling.
>
> [4] OptiMUS: Scalable Optimization Modeling with (MI)LP Solvers and Large Language Models.

---

> > ### Author Response · Authors · 2025-08-04
> > **Further Explanation of OptiTree for your Better Understanding (Part 1)**
> >
> > Dear Reviewer MotJ,
> >
> > To further address your concerns. We provide (1) a failure case of OptiTree to see when it underperforms, (2) a case to demonstrate how OptiTree works using a modeling tree, and (3) the pseudo-code for your better understanding of our work.
> >
> > - We provide an example where OptiTree fails to understand the constraints in the problems and thus demonstrate the critical role of stronger common-sense reasoning.
> >
> >   > Both chorine and water softener need to be added to a pool. One unit of chlorine takes 1 minute to be effective while one unit of water softener takes 2 minutes to be effective. Because too much chlorine can burn your eyes, there has to at most half as much chlorine as water softener in the pool. If there has to be at least 200 units of chlorine in the water and a total of 500 units of chemicals in the water, how many units of each should be added to minimize the total time it takes for the pool to be ready? (Note: chlorine must be added before water softener or vice versa).
> >
> >   Let $x\_c \in \mathbb{R}\_+$ be the amount of chlorine (units), and $x\_w \in \mathbb{R}\_+$ be the amount of water softener (units). The output model of OptiTree is
> >
> >   $$ \begin{aligned} & \min t \\\\ \text{s.t.} \quad & x_c + x_w = 500 \\\\ & x_c \leq 0.5x_w \\\\& t \geq x_c \\\\ & t \geq 2x_w \\\\ & x_c \geq 200 \\\\ & x_w \geq 0 \end{aligned} $$
> >
> >   The optimization model fails because its constraints are mathematically infeasible and violate real-world requirements. The core issue stems from the rigid equality constraint $x_c+x_w=500$ (total chemicals must equal 500 units), which conflicts with the safety constraint $x_c\le0.5x_w$ (chlorine cannot exceed half the water softener amount) when combined with the minimum chlorine requirement $x_c\ge200$.
> >
> >   The error is due to a **misunderstanding of the original problem**. While OptiTree provides a structured framework and knowledge, the final synthesis depends on the LLM's inherent reasoning capabilities. **This framework may struggle with problems requiring significant logical or relational reasoning. We believe that improving common sense reasoning or logical reasoning ability in the base model can effectively address this challenge**.

---

> > > ### Author Response · Authors · 2025-08-04
> > > **Further Explanation of OptiTree for your Better Understanding (Part 2)**
> > >
> > > - We also provide a further example to demonstrate our method for handling general problems. The example is a shortest path problem in a network flow, where the network is a directed graph with inflow and outflow for each node.
> > >
> > >   > There are two special nodes marked as S (likely the start) and T (likely the target or terminal). The other nodes are numbered from 2 to 7. Edges connect these nodes and each edge is labeled with a number indicating its weight.
> > >   >
> > >   >
> > >   >
> > >   > Node S is connected to nodes 2 and 3 with edge weights of 1 and 4, respectively.
> > >   >
> > >   > Node 2 is connected to node S with a weight of 1, to node 4 with a weight of 3, and to node 3 with a weight of 2.
> > >   >
> > >   > Node 3 is connected to node S with a weight of 4, to node 2 with a weight of 2, to node 5 with a weight of 1, and to node 6 with a weight of 7.
> > >   >
> > >   > Node 4 is connected to node 2 with a weight of 3 and to node 6 with a weight of 2.
> > >   >
> > >   > Node 5 is connected to node 3 with a weight of 1 and to node T with a weight of 5.
> > >   >
> > >   > Node 6 is connected to nodes 3, 4, and 7 with edge weights of 7, 2, and 3, respectively.
> > >   >
> > >   > Node 7 is connected to nodes 6 and T with edge weights of 3 and 1, respectively.
> > >   >
> > >   > Node T is connected to nodes 5 and 7 with edge weights of 5 and 1, respectively.
> > >   >
> > >   >
> > >   >
> > >   > Considering the weight as distance, find the shortest distance from S to T. Provide your answer rounded to the nearest meter."
> > >
> > >   We identify the subproblem of the shortest path problem. Let $\mathcal{N} = \\{S, 2, 3, 4, 5, 6, 7, T\\}$ be the set of nodes, $\mathcal{E}$  be the set of edges (where $(i,j,c)$ represents edge from $i$ to $j$ with cost $c$). We define the decision variables
> > >   $$
> > >   x_{ij} \in \\{0,1\\} \quad \forall (i,j,c) \in \mathcal{E},
> > >   $$
> > >
> > >   where $x_{ij} = 1$ if edge $i \rightarrow j$ is included in the path, and $x_{ij} = 0$ otherwise
> > >
> > >   The model given by the standard output of DeepSeek-V3 is
> > >
> > >   > $$\begin{aligned}
> > >   > & \min \sum_{(i,j,c) \in \mathcal{E}} c \cdot x_{ij} \\\\
> > >   > \text{s.t.} \quad & \sum_{(S,j,c) \in \mathcal{E}} x_{Sj} = 1 \\\\
> > >   > & \sum_{(i,T,c) \in \mathcal{E}} x_{iT} = 1 \\\\
> > >   > & \sum_{(i,n,c) \in \mathcal{E}} x_{in} = \sum_{(n,j,c) \in \mathcal{E}} x_{nj} \quad \forall n \in \mathcal{N} \setminus \\{S,T\\} \\\\
> > >   > & x_{ij} \in \{0,1\} \quad \forall (i,j,c) \in \mathcal{E}
> > >   > \end{aligned}$$
> > >
> > >   The results given by our method are as follows. OptiTree identifies this problem as a shortest path problem in a network flow and retrieves the corresponding modeling thoughts. The key difference between the two models lies in how they handle flow constraints at the source (`S`) and sink (`T`) nodes. The standard output model assumes no edges enter the source or leave the sink, enforcing simple constraints where the source must have exactly one outgoing edge ($\sum_jx_{Sj} = 1$) and the sink must have exactly one incoming edge ($\sum_ix_{iT} = 1$). In contrast, the model from OptiTree uses a **more generalized flow balance approach from the modeling thoughts of the same kind of problems**: it ensures the source has a net outflow of 1 ($\sum x_{Sj} − \sum x_{iS} = 1$) and the sink has a net inflow of 1 ($\sum x_{iT} − \sum x_{Tj} = 1$).
> > >
> > >   > $$\begin{aligned}
> > >   > & \min \sum_{(i,j,c) \in \mathcal{E}} c \cdot x_{ij} \\\\
> > >   > \text{s.t.} \quad & \sum_{(S,j,c) \in \mathcal{E}} x_{Sj} - \sum_{(i,S,c) \in \mathcal{E}} x_{iS} = 1 \quad && \text{(Source)} \\\\
> > >   > & \sum_{(i,T,c) \in \mathcal{E}} x_{iT} - \sum_{(T,j,c) \in \mathcal{E}} x_{Tj} = 1 \quad && \text{(Sink)} \\\\
> > >   > & \sum_{(i,n,c) \in \mathcal{E}} x_{in} = \sum_{(n,j,c) \in \mathcal{E}} x_{nj} \quad \forall n \in \mathcal{N} \setminus \\{S,T\\} && \text{(Flow conservation)} \\\\
> > >   > & x_{ij} \in \{0,1\} \quad \forall (i,j,c) \in \mathcal{E}
> > >   > \end{aligned}$$

---

> > > > ### Author Response · Authors · 2025-08-04
> > > > **Further Explanation of OptiTree for your Better Understanding (Part 3)**
> > > >
> > > > To further explain the tree construction process for you, we provide the pseudo-code as follows. If you have any further thoughts, questions, or suggestions---either regarding our current response or the broader direction of the work---we would be genuinely grateful to hear them. We highly value your perspective and would welcome any opportunity for continued discussion or clarification.
> > > >
> > > > ----
> > > >
> > > > Algorithm 1 Tree Search for Subproblem Decomposition
> > > >
> > > > ----
> > > >
> > > > - Require: Target problem $\mathcal{P}$, modeling tree $\mathbb{T}$
> > > >
> > > > - Ensure: Hierarchical modeling thoughts $\mathcal{T} (\mathcal{P})$, and the maximal subproblem $\tilde{\mathcal{P}}$
> > > >
> > > > - **function** TreeSearch($\mathcal{P}$, $\mathbb{T}$ ):
> > > >
> > > >   - Initialize current node $\mathcal{N}\leftarrow$ The root of the modeling tree $\mathbb{T}$
> > > >   - \# **prompts block 1 and 2 in Appendix H.4**
> > > >
> > > >   - Extract statement thoughts $\mathcal{C}_\mathcal{P}$
> > > >   - \# Identified subproblems
> > > >   - **while** $\mathcal{N}$ has children **do**
> > > >     - Get the children of $\mathcal{N}$
> > > >     - \# **prompts in Appendix H.2**
> > > >     - Identify the subproblem of $\mathcal{P}$ among the children with similarity scores
> > > >     - **If** $\mathcal{P}$ has a subproblem among the children **then**
> > > >       - $\mathcal{N}\leftarrow$ Subproblem with the max similarity score
> > > >       - The maximal subproblem $\tilde{\mathcal{P}}\leftarrow \mathcal{N}$
> > > >     - **else**
> > > >       - $\mathcal{N}\leftarrow$ None
> > > >     - **end if**
> > > >   - **end while**
> > > >
> > > > - Synthesize $\mathcal{T}(\mathcal{P})$ from $\mathcal{T}(\tilde{\mathcal{P}})$
> > > >
> > > > - **return** $\mathcal{T}(\mathcal{P})$, $\tilde{\mathcal{P}}$
> > > >
> > > > - **end function**
> > > >
> > > > ----
> > > >
> > > > ----
> > > >
> > > >  Algorithm 2 Modeling Tree Update
> > > >
> > > > ----
> > > >
> > > > - Require: Target problem $\mathcal{P}$, modeling tree $\mathbb{T}$
> > > > - Ensure: Updated Modeling Tree
> > > > - **function** TreeUpdate($\mathcal{P}$, $\mathbb{T}$ ):
> > > >
> > > >   - $\mathcal{T}(\mathcal{P})$, $\tilde{\mathcal{P}}\leftarrow$ TreeSearch($\mathcal{P}$, $\mathbb{T}$)
> > > >   - \# **prompts in Appendix H.3**
> > > >   - Model the problem $\mathcal{P}$ using the modeling thoughts and solve the model for optimal value $y$
> > > >   - **If** $y$ does not equal to the ground-truth objective value then
> > > >     - \# **prompts block 1 and 2 in Appendix H.4**
> > > >     - Distill schema (problem type,$\mathcal{C}_\mathcal{P}$,$\mathcal{T} (\mathcal{P})$) from $\mathcal{P}$
> > > >     - Find the children of the maximal subproblem $\tilde{\mathcal{P}}$
> > > >     - **for** child $\tilde{\mathcal{P}}_k$ in the children **do**
> > > >       - \# **prompts block 3 in Appendix H.4**
> > > >       - **if** $\mathcal{P}\subset_{\mathcal{S}}\tilde{\mathcal{P}}_k$ **then**
> > > >         - Insert $\mathcal{P}$ as parent of $\tilde{\mathcal{P}}_k$ and children of $\tilde{\mathcal{P}}$
> > > >       - **else**
> > > >         - Insert $\mathcal{P}$ as sibling of $\tilde{\mathcal{P}}_k$ and children of $\tilde{\mathcal{P}}$
> > > >       - **end if**
> > > >     - **end for**
> > > >   - **end if**
> > > > - **return** $\mathbb{T}$
> > > > - **end function**

---

> ### Comment · Area_Chair_v4v6 · 2025-08-06
>
> Dear reviewer,
>
> Could you please read the authors' response, and accomplish the mandatory acknowledgement a.s.a.p.?
>
> AC

---

> ### Author Response · Authors · 2025-08-08
> **Further discussions on the improvement and limitations of OptiTree**
>
> Dear Reviewer MotJ,
>
> You noted that the limitations of our method could be discussed more thoroughly. To address this, we provide a detailed analysis of an example from the **IndustryOR** dataset.
>
> This example demonstrates that while **OptiTree successfully corrects fundamental structural errors** (what [1] terms "Incorrect Modeling"), it still faces challenges with constraints that require deep, context-specific **logical reasoning**.
>
> > **Problem**. A  factory requires a specialized tool for $n=10$ planning stages.  $r_j$ specialized tools are needed. At the end of this stage, all tools used within this stage must be sent for repair. There are two repair methods: one is slow repair, which is cheaper (costs $b$ per tool) but takes longer ($p=3$ stages to return); the other is fast repair, which costs $c$ per tool and requires $q=1$ stages to return. If the tools cannot meet the needs, new ones must be purchased, with a cost of $a$ per new tool. Determine an optimal plan to minimize the cost.
>
> - **The Errors of the Standard Output Model**.  The standard output from DeepSeek-V3 produces a structurally flawed model. It tries to fulfill demand by combining new purchases with tools returning from repair, but it completely lacks any concept of inventory tools.
>
>   > **The model given by the Standard output of DeepSeek-V3**. Let $x_j \geq 0$ be the new tools purchased at stage $j$, $s_j \geq 0$ be the tools sent to slow repair at stage $j$, and $f_j \geq 0$ be the tools sent to fast repair at stage $j$.
>   >
>   > $$\begin{aligned}
>   > \min \quad &\sum_{j=1}^{n} \left(a x_j+b s_j+c f_j\right)  &\\\\
>   > &s_j+f_j= r_j\quad \forall j \in \\{1,\dots,n\\} &\text{(Repair Requirement)} \\\\
>   > &x_j+s_{k-p}+f_{k-q} \geq r_j \quad \forall j \in \\{1,\dots,n\\} &\text{(Demand Fulfillment)}
>   > \end{aligned}$$
>
>   Following the errors described in OptiMUS [1], the model output has the error type of **Incorrect modeling**, which fails to define the **Inventory variables $I$**.
>
> - **The Improvement of OptiTree**. OptiTree correctly identifies that this problem contains an **Inventory Problem** subproblem. By applying the modeling thoughts, it generates a much-improved model that correctly introduces inventory variables $I$ and a proper inventory balance constraint.
>
>   > **The model given by OptiTree (DeepSeek-V3)**. Let $x_j \in \mathbb{Z}\_+$ be the new tools purchased at stage $j$, $s_j \in \mathbb{Z}\_+$ be the tools sent to slow repair at stage $j$, $f_j \in \mathbb{Z}\_+$ be the tools sent to fast repair at stage $j$, and $I_j \in \mathbb{Z}\_+$ be the inventory of available tools at start of stage $j$. OptiTree finds a **subproblem of the Inventory Problem** for the original problem.
>   >
>   > $$\begin{aligned}
>   > \min \quad &\sum_{j=1}^{n} \left(a x_j+b s_j+c f_j\right)  &\\\\&I_1=0&\text{(Initial Inventory)}\\\\&s_j+f_j=r_j \quad \forall j \in \\{1,\dots,n\\} &\text{(Repair Requirement)} \\\\
>   > &I_{j+1}=(I_j+x_j - r_j)+s_{j-p}+f_{j-q}\quad \forall j \in \\{1,\dots,n\\} &\text{(Inventory Balance)} \\\\&I_j+x_j \geq r_j \quad \forall j \in \\{1,\dots,n\\}  &\text{(Demand Fulfillment)}
>   > \end{aligned}$$
>
> - **The Errors in the Model Given by OptiTree**. Despite the structural correction, the OptiTree model still contains a subtle error that requires **common-sense logical reasoning**. The model does not account for the finite 10-stage horizon. **A human modeler would reason that it is illogical to pay for a repair that will finish after the tools are no longer needed**.
>
>   - Slow repairs (taking $p=3$ stages) are disabled for the last 3 stages ($j=8,9,10$) since tools sent for repair at these stages wouldn't return by stage 10, and fast repairs (taking $q=1$ stage) are disabled for the final stage ($j=10$) for the same reason. The repair requirement uses inequality
>
>     $$s_j+f_j \le r_j \quad\text{(Repair Requirement)}$$
>
>     rather than equality $s_j+f_j=r_j$, allowing flexibility to not repair tools when unnecessary---particularly near the end of the horizon. Thus, the final corrected model should be
>
>     $$\begin{aligned}
>     \min \quad &\sum_{j=1}^{n} \left(a x_j+b s_j+c f_j\right)  &\\\\&I_1=0&\text{(Initial Inventory)}\\\\&s_j+f_j \le r_j \quad \forall j \in \\{1,\dots,n\\} &\text{(Repair Requirement)} \\\\
>     &I_{j+1}=(I_j+x_j - r_j)+s_{j-p}+f_{j-q}\quad \forall j \in \\{1,\dots,n\\} &\text{(Inventory Balance)} \\\\&I_j+x_j \geq r_j \quad \forall j \in \\{1,\dots,n\\}  &\text{(Demand Fulfillment)}
>     \end{aligned}$$
>
>   - This demonstrates a limitation where OptiTree applies a general pattern correctly but misses a nuanced, context-dependent constraint. The error stems from a lack of OR knowledge patterns and a gap in the base LLM's logical reasoning about the problem's specific context. We believe this limitation will diminish as the underlying reasoning capabilities of LLMs continue to improve.
>
> [1] OptiMUS: Scalable Optimization Modeling with (MI)LP Solvers and Large Language Models.
>
> Best,
>
> Authors

---

> ### Author Response · Authors · 2025-08-09
>
> Dear Reviewer MotJ,
>
> We sincerely thank you for your time and efforts during the rebuttal process. We are writing to gently remind you that the author-reviewer discussion period will end in less than 10 hours. We have responded to your further comments and eagerly await your feedback, and we sincerely hope that our response has properly addressed your concerns.
>
> We eagerly await your feedback to understand if our responses have adequately addressed all your concerns. If so, we would deeply appreciate it if you could raise your score. We sincerely thank you once more for your insightful comments and kind support.
>
> Best,
>
> Authors

---

### Official Review · Reviewer_JMeQ · 2025-07-03

**Clarity:** 2
**Significance:** 2
**Originality:** 3
**Rating:** 4
**Confidence:** 3

**Summary:**

This paper presents OptiTree, a novel approach for improving LLM-based optimization modeling through hierarchical problem decomposition. The work addresses the challenge of automating operations research modeling, where complex problems described in natural language must be converted into mathematical optimization models. The core innovation lies in organizing OR problems into a tree structure where complex problems are decomposed into simpler subproblems, enabling more effective modeling through adaptive rather than fixed-step approaches.

**Questions:**

In table 2, I am confused by how the compute times for tree search and modeling are calculated. My understanding is that modeling should be a 1-time cost when the tree is originally built. Is this reflected in the table?

**Ethical Concerns:**

["NO or VERY MINOR ethics concerns only"]

**Final Justification:**

I am raising my score by 1 since the rebuttal helps address some of my questions. Overall, the paper is interesting but I think the methodology may be difficult to generalize to ML more broadly. There are also some concerns with clarity and reproducibility which has also been flagged by the other reviewers.

**Limitations:**

Limitations could be discussed more thoroughly.

**Paper Formatting Concerns:**

No, besides labels in appendix figures are hard to read.

**Quality:**

2

**Strengths And Weaknesses:**

The paper demonstrates good results with comprehensive experiments across five datasets, showing consistent 10%+ improvements on challenging benchmarks compared to both prompt-based methods and fine-tuned models. The methodology is rigorous, including proper comparisons to state-of-the-art reasoning LLMs like OpenAI-o1 and DeepSeek-R1, along with thorough ablation studies examining tree search, modeling thoughts, search depth, and statement thoughts components.

However, the paper suffers from clarity issues and has some experimental concerns. The motivating observations lack crucial details - it's unclear which specific dataset was used for the three motivational experiments, what constitutes the "50 standard OR problems," or what it means to "provide ground-truth models for that subproblem." Technical details are often relegated to appendices, making it difficult to understand how statement thoughts are extracted or how the similarity function SimLLM operates. The evaluation has notable limitations, relying solely on solving accuracy (whether optimal objectives match ground truth) without analyzing partial credit.

The scalability and practical deployment aspects raise additional concerns. Tree construction seems to require human curation for new domains, and it's unclear how this approach scales or what happens when problems don't fit existing tree structures. The paper lacks analysis of tree maintenance costs over time and provides no thorough error analysis of failure cases. From a significance standpoint,
it seems like the evaluation is limited to standard OR problem types and it may be difficult for this approach to generalize to ML more broadly.

---

> ### Author Rebuttal · Authors · 2025-07-31
>
> Thank you for your valuable comments. We sincerely hope that we could properly address your concerns.
>
> > W1: Details of motivating observations
>
> - We use the **IndustryOR** dataset for three experiments. The "**50 standard OR problems**" refer to a collection of foundational OR problems in the OR textbook [1], including classical problems such as the standard TSP, Knapsack, set packing, assignment, and facility location problems.
>
> - **Providing the ground-truth model**. (1) We first collect the optimization models in the textbook [1] as a ground-truth model for the 50 standard problems. (2) For a problem in IndustryOR to formulate, we use LLM to identify whether the problem contains subproblems in the 50 standard OR problems. (3) If a subproblem is identified, we augment the LLM's prompt for optimization modeling by integrating the relevant ground-truth model of the subproblem. The simplified prompt is as follows.
>
>   > You are a mathematical formulator to formulate the given OR problem as an optimization model and generate Gurobi code to solve the model.
>   >
>   > {problem description}
>   >
>   > The problem has a subproblem of {subproblem name}. During the modeling process, we can refer to the subproblem model: {subproblem model}
>
> > W1: Details of statement thoughts extraction and similarity function SimLLM.
>
> The extraction of statement thoughts and the similarity function SimLLM are both processes handled by **a LLM using specific prompts**.
>
> - **Statement thoughts** are concise, high-level statements that summarize the key features and requirements of the problem. They are generated to create a clear and comprehensible format for subproblem identification. We provide an example here (Appendix H.1).
>
>   > Diet Problem with Integer Constraint
>   >
>   > - The Diet Problem with Integer Constraint involves determining the quantity of each food item to meet nutritional requirements while minimizing cost.
>   > - Nutritional Constraints: The selected food items provide at least the required amount of nutrients.
>   > - Cost Minimization: The total cost of the selected food items should be minimized.
>   > - Integer Servings: Food items must be integer values.
>
>   An example prompt to obtain statement thoughts is:
>
>   >  Your task is to summarize the type of this specific problem and provide a detailed statement thoughts of this type. You must classify the problem into more precise scenarios, such as the Traveling Salesman Problem (TSP), the facility location problem, and so on.
>   >
>   >  Specific Problem: {problem description}
>   >
>   >  Here is an output format example ...
>
> - **SimLLM** provides a similarity score directly from the LLM, used during the tree search to measure how closely a candidate subproblem’s statement thoughts align with those of the target problem. This involves a detailed prompt asking the LLM to compare a list of statement thoughts and return the most similar one in JSON format. The output includes a discrete score of 0, 1, or 2, where 2 indicates high similarity. SimLLM is used to select the best-matching subproblem at each step of the tree search.
>
>   > Input problem: {input_problem_statement_thoughts}.
>   >
>   > You are given a list of defined subtypes and the statement thoughts of subproblems: {subproblems_statement_thoughts}.
>   >
>   > Your task is to determine if the specific problem belongs to one of the given subtypes. Return in the following JSON format:
>   >
>   > json{
>   >
>   > ​	matching_subtype: "",
>   >
>   > ​	reasons: "",
>   >
>   > ​	similarity: [0,1,2]
>   >
>   > }
>
> > W1: Evaluation with partial credit.
>
> The objective metric is standard in the LLM-based optimization modeling literature [2, 3], and in subsequent works ORLM, LLMOPT, OptMATH, OptiBench. However, defining a robust "partial credit" metric is a significant challenge.
>
> - Most existing benchmarks **only contain labeled optimal value** and **do not contain annotated ground-truth optimization model**, making the partial credit metric difficult to design.
> - Even with ground-truth models, simple text-based or structural comparisons can be unreliable. A model may have minor syntactic differences from the ground truth but be semantically equivalent. Conversely, a model might be nearly identical yet contain a critical error, such as a misplaced inequality sign.
> - To further address your concerns, we find that the training set in OptMATH has annotated models. We use 100 problems for evaluation. As our method and the baselines were not trained on this dataset, the comparison is safe and fair. We present the percentages of LLM-generated models that align with the ground-truth models based on statistical information, including the number of variables, binary variables, integer variables, constraints and objective values. Note that all the statistical information is just a reference, as the **mismatch of variables and constraints does not imply the model is incorrect**. Notably, OptiTree exhibits the highest matching ratio with the ground-truth model, underscoring its reliability.
>
> **Table 1**: Evaluation of partial credit.
>
> |OptMATH|Num Var|Num Bin|Num Int|Num Cons|Obj|
> |-|-|-|-|-|-|
> |DeepSeek-R1|79.0|85.0|67.0|46.0|66.0|
> |OpenAI-o1|83.0|87.0|**93.0**|50.0|68.0|
> |ORLM|43.0|52.0|44.0|35.0|37.0|
> |LLMOPT|56.0|65.0|57.0|37.0|41.0|
> |CoE|67.0|78.0|85.0|46.0|58.0|
> |OptiMUS|84.0|79.0|76.0|43.0|61.0|
> |MCTS|75.0|80.0|81.0|46.0|63.0|
> |Ours|**90.0**|**92.0**|83.0|**68.0**|**71.0**|
>
> > W2: Tree construction seems to require human curation, and it's unclear whether the scalability and what happens when problems don't fit existing tree structures.
>
> - We want to clarify that the tree construction and updating process is **fully automated and does not require human curation** for new domains. We construct the modeling tree using 400 randomly selected problems in the OR-Instruct dataset (training dataset for ORLM). We run across problems in the dataset to update the tree and use the single tree across the testing benchmarks.
> - Scalability. We use **just 400 problems** from the OR-Instruct dataset for tree construction and test across a wide range of different problem domains. The superior performance demonstrates the scalability.
> - If the tree search process cannot find any relevant subproblems for a given task (i.e., the search halts at the root node), **OptiTree simply provides no modeling thoughts**. In this case, the framework defaults to the LLM's baseline modeling capability without the enhancement from OptiTree.
>
> > W2 & Q: Analysis of tree maintenance costs over time and the results in Table 2 in the main text.
>
> - Your understanding of the **1-time cost for tree construction is right**. However, this part of the time is not included in Table 2 in the main text. The result in Table 2 represents the inference time required to generate the final optimization model for each new problem. We (1) use tree search to find an appropriate problem and (2) perform optimization modeling, combining the modeling thoughts from the problem. The two parts of time (Tree search and modeling) refer to these two steps.
> - We provide the tree maintenance costs. We also provide the sum of tree construction and inference time (CPU time) as follows (in **hours**, and the reported **Average Time in seconds**). Compared to the inference time of the baselines or the time cost for fine-tuning LLMs, our method is the most efficient.
>
> **Table 2**: The comparisons of the inference time.
>
> ||Tree Construction/Fine-tune Time (h)|NL4Opt (h)|MAMO EasyLP (h)|MAMO ComplexLP (h)|ComplexOR (h)|IndustryOR (h)|Total Time (h)|Average Time (In Seconds)|
> |-|-|-|-|-|-|-|-|-|
> |14B LLM|>80|-|-|-|-|-|-|-|
> |CoE|-|2.98|7.71|4.50|0.42|2.27|17.88|50.64|
> |OptiMUS|-|2.11|4.08|3.11|0.36|1.60|11.26|31.89|
> |MCTS|-|8.28|20.06|6.53|1.00|3.46|39.33|111.39|
> |Optitree|3.00|1.11|1.68|0.78|0.16|0.55|7.28|**20.62**|
>
> > W2:  generalize to ML more broadly
>
> While our work centers on optimization modeling within OR, the core principle of our approach—**leveraging a hierarchical knowledge structure to guide an LLM in decomposing complex problems into simpler, solvable subproblems and retrieving contextual hints for each**—represents a generalizable reasoning paradigm. We argue that **hierarchical knowledge retrieval with tree search** offers a versatile framework that could inspire advancements in other domains requiring multi-step, complex reasoning.
>
> > W2: Error cases and limitations
>
> Following [2], the errors mainly arise from three aspects: missing or wrong constraints, incorrect modeling and coding errors. In [2], almost half of the errors come from the incorrect modeling (62.5% for ComplexOR), e.g., defining wrong binary variables in the model. We analyze the failures of OptiTree and normalize the failure rates to sum to 1. The results are presented in Table 3. We find that most errors are from the missing constraints, and the ratio of incorrect modeling is low. We have the following conclusions. (1) **OptiTree can significantly reduce the error of incorrect modeling, primarily caused by wrong variable definitions**. The results coincide with those in Table 3, where the number of variables is closest to the ground-truth model. (2) We can also see the **limitations of OptiTree**. While it substantially mitigates incorrect modeling through decomposition and modeling thoughts, it still relies on the base model's capacity to understand and interpret problems. The requisite OR knowledge for problem comprehension remains a limitation.
>
> **Table 3**: Failure cases information.
>
> ||ComplexOR|IndustryOR|OptMATH|
> |-|-|-|-|
> |Incorrect modeling|0|10.9|23.8|
> |Missing constraints|66.7|82.6|61.9|
> |Coding errors|33.3|6.5|14.3|
>
> [1] Optimization in Operations Research
>
> [2] OptiMUS: Scalable Optimization Modeling with (MI)LP Solvers and Large Language Models
>
> [3] Chain-of-Experts: When LLMs Meet Complex Operations Research Problems

---

> ### Author Response · Authors · 2025-08-08
>
> Dear Reviewer JMeQ,
>
> Thank you for your feedback and for acknowledging that our proposed revisions will improve the paper's clarity. We will ensure that all the details discussed during our discussions are fully incorporated into the final manuscript.
>
> Please let us know if any points remain unclear or if you have any further questions. We eagerly await your feedback to understand if our responses have adequately addressed all your concerns. If so, we would deeply appreciate it if you could raise your score.
>
> Thank you again for your time and constructive suggestions!
>
> Best,
>
> Authors

---

> > ### Author Response · Authors · 2025-08-08
> > **Further discussions on the improvement and limitations of OptiTree**
> >
> > Dear Reviewer JMeQ,
> >
> > We try to discuss the limitations of OptiTree more thoroughly to fully address your concerns. We provide a detailed analysis of an example from the **IndustryOR** dataset.
> >
> > This example demonstrates that while **OptiTree successfully corrects fundamental structural errors** (what [1] terms "Incorrect Modeling"), it still faces challenges with constraints that require deep, context-specific **logical reasoning**.
> >
> > > **Problem**. A  factory requires a specialized tool for $n=10$ planning stages.  $r_j$ specialized tools are needed. At the end of this stage, all tools used within this stage must be sent for repair. There are two repair methods: one is slow repair, which is cheaper (costs $b$ per tool) but takes longer ($p=3$ stages to return); the other is fast repair, which costs $c$ per tool and requires $q=1$ stages to return. If the tools cannot meet the needs, new ones must be purchased, with a cost of $a$ per new tool. Determine an optimal plan to minimize the cost.
> >
> > - **The Errors of the Standard Output Model**.  The standard output from DeepSeek-V3 produces a structurally flawed model. It tries to fulfill demand by combining new purchases with tools returning from repair, but it completely lacks any concept of inventory tools.
> >
> >   > **The model given by the Standard output of DeepSeek-V3**. Let $x_j \geq 0$ be the new tools purchased at stage $j$, $s_j \geq 0$ be the tools sent to slow repair at stage $j$, and $f_j \geq 0$ be the tools sent to fast repair at stage $j$.
> >   >
> >   > $$\begin{aligned}
> >   > \min \quad &\sum_{j=1}^{n} \left(a x_j+b s_j+c f_j\right)  &\\\\
> >   > &s_j+f_j= r_j\quad \forall j \in \\{1,\dots,n\\} &\text{(Repair Requirement)} \\\\
> >   > &x_j+s_{k-p}+f_{k-q} \geq r_j \quad \forall j \in \\{1,\dots,n\\} &\text{(Demand Fulfillment)}
> >   > \end{aligned}$$
> >
> >   Following the errors described in OptiMUS [1], the model output has the error type of **Incorrect modeling**, which fails to define the **Inventory variables $I$**.
> >
> > - **The Improvement of OptiTree**. OptiTree correctly identifies that this problem contains an **Inventory Problem** subproblem. By applying the modeling thoughts, it generates a much-improved model that correctly introduces inventory variables $I$ and a proper inventory balance constraint.
> >
> >   > **The model given by OptiTree (DeepSeek-V3)**. Let $x_j \in \mathbb{Z}\_+$ be the new tools purchased at stage $j$, $s_j \in \mathbb{Z}\_+$ be the tools sent to slow repair at stage $j$, $f_j \in \mathbb{Z}\_+$ be the tools sent to fast repair at stage $j$, and $I_j \in \mathbb{Z}\_+$ be the inventory of available tools at start of stage $j$. OptiTree finds a **subproblem of the Inventory Problem** for the original problem.
> >   >
> >   > $$\begin{aligned}
> >   > \min \quad &\sum_{j=1}^{n} \left(a x_j+b s_j+c f_j\right)  &\\\\&I_1=0&\text{(Initial Inventory)}\\\\&s_j+f_j=r_j \quad \forall j \in \\{1,\dots,n\\} &\text{(Repair Requirement)} \\\\
> >   > &I_{j+1}=(I_j+x_j - r_j)+s_{j-p}+f_{j-q}\quad \forall j \in \\{1,\dots,n\\} &\text{(Inventory Balance)} \\\\&I_j+x_j \geq r_j \quad \forall j \in \\{1,\dots,n\\}  &\text{(Demand Fulfillment)}
> >   > \end{aligned}$$
> >
> > - **The Errors in the Model Given by OptiTree**. Despite the structural correction, the OptiTree model still contains a subtle error that requires **common-sense logical reasoning**. The model does not account for the finite 10-stage horizon. **A human modeler would reason that it is illogical to pay for a repair that will finish after the tools are no longer needed**.
> >
> >   - Slow repairs (taking $p=3$ stages) are disabled for the last 3 stages ($j=8,9,10$) since tools sent for repair at these stages wouldn't return by stage 10, and fast repairs (taking $q=1$ stage) are disabled for the final stage ($j=10$) for the same reason. The repair requirement uses inequality
> >
> >     $$s_j+f_j \le r_j \quad\text{(Repair Requirement)}$$
> >
> >     rather than equality $s_j+f_j=r_j$, allowing flexibility to not repair tools when unnecessary---particularly near the end of the horizon. Thus, the final corrected model should be
> >
> >     $$\begin{aligned}
> >     \min \quad &\sum_{j=1}^{n} \left(a x_j+b s_j+c f_j\right)  &\\\\&I_1=0&\text{(Initial Inventory)}\\\\&s_j+f_j \le r_j \quad \forall j \in \\{1,\dots,n\\} &\text{(Repair Requirement)} \\\\
> >     &I_{j+1}=(I_j+x_j - r_j)+s_{j-p}+f_{j-q}\quad \forall j \in \\{1,\dots,n\\} &\text{(Inventory Balance)} \\\\&I_j+x_j \geq r_j \quad \forall j \in \\{1,\dots,n\\}  &\text{(Demand Fulfillment)}
> >     \end{aligned}$$
> >
> >   - This demonstrates a limitation where OptiTree applies a general pattern correctly but misses a nuanced, context-dependent constraint. The error stems from a lack of OR knowledge patterns and a gap in the base LLM's logical reasoning about the problem's specific context. We believe this limitation will diminish as the underlying reasoning capabilities of LLMs continue to improve.
> >
> > [1] OptiMUS: Scalable Optimization Modeling with (MI)LP Solvers and Large Language Models.
> >
> > Best,
> >
> > Authors

---

> > ### Author Response · Authors · 2025-08-09
> >
> > Dear Reviewer JMeQ,
> >
> > We sincerely thank you for your time and efforts during the rebuttal process. We are writing to gently remind you that the author-reviewer discussion period will end in less than 10 hours. We have responded to your further comments and eagerly await your feedback, and we sincerely hope that our response has properly addressed your concerns.
> >
> > We eagerly await your feedback to understand if our responses have adequately addressed all your concerns. If so, we would deeply appreciate it if you could raise your score. We sincerely thank you once more for your insightful comments and kind support.
> >
> > Best,
> >
> > Authors

---

### Note · Authors · 2025-08-15

Dear Area Chair and Reviewers,

Thank you for the thorough and constructive review process. Your feedback has been invaluable in strengthening our paper.

Based on the reviewers' feedback, the paper's key strengths are as follows.

- Reviewers consistently praised the **novelty and concept of the Modeling Tree** as a significant contribution.

  - The method has **"clear structural advance over standard chain-of-thought prompting"** (`Reviewer 9LBo`).

  - The paper focuses on a **"long-standing bottleneck in OR"** and proposes an approach with **"structured, hierarchical memory for LLMs"** and **"high level concepts... make sense"** (`Reviewer MotJ`).

- The method demonstrated strong **improvements in experiments**.

  - OptiTree achieves **"consistent 10%+ improvements on challenging benchmarks"** (`Reviewers JMeQ and vacN`).

  - OptiTree achieves **"cutting inference latency"** and **"nearly perfect code-execution rates"**(`Reviewer 9LBo`).

- The quality of the paper's structure and analysis was also commended.

  - The methodology is **"rigorous"** and supported by **"thorough ablation studies"** (`Reviewers JMeQ and vacN`). The paper motivates its design with **"diagnostic analyses, intuitive diagrams, and thorough ablations"** (`Reviewer 9LBo`).

  - The **"motivated observations"** were considered **"insightful"** (`Reviewer vacN`).

In response, the authors provided a comprehensive rebuttal that went beyond clarification by conducting extensive new experiments and giving detailed explanations to address every major point.

- We provide detailed analysis of the successful and failure cases. We also offer more details for the pseudo-codes, prompts, and the procedure of the algorithms (`for all reviewers`).
- Adding the latest SOTA baselines (Evo-Step, OptMATH), testing on new benchmarks (OptiBench, OptMATH), and different LLMs (`in our response to Reviewers MotJ, vacN, and 9LBo`).
- Performing a quantitative error analysis for more insights (`for all reviewers`).
- Conducting a "partial credit" structural analysis for further evaluation (`in our response to Reviewers JMeQ and 9LBo`).
- Providing a sensitivity analysis on the tree construction process (`in our response to Reviewer vacN`).

We are fully committed to incorporating all new results and clarifications into a significantly improved final manuscript. We will also **fully open-source our code** to ensure complete reproducibility.

Thank you for your consideration.

Best,

Authors

---

### Decision · Program_Chairs · 2025-09-17

**Decision:**

Accept (poster)

**Comment:**

This paper studies the modeling of optimization problems from natural language descriptions. It proposes to progressively model an optimization problem by modeling subproblems, in order to improve the success rate. The reviewers all had ackowledge the novelty of this paper and lean to acceptance. However, it should be noticed that this work introduces a heavy dependence on the subproblem space. The proposed approach is expected to perform well when the subproblem space covers the target problem space. The experiments conducted in this paper fall into this category. Meanwhile, when sovling general optimization problems that are far from the subproblem space, the approach's effort of modeling subproblems could only introduce confusing context for the LLM. The authors should discuss the affect of the coverage of the subproblem space and conduct experiments.